# TIME-SERIES OPEN-SET RECOGNITION WITH ADAPTIVE LOCAL OUTLIER SYNTHESIS AND EXPOSURE

## ABSTRACT

Open-set recognition (OSR) in time-series data presents a significant challenge due to the need to detect and reject unknown classes while maintaining robust classification of known classes. To address this, we introduce Adaptive Localized latent outlier Synthesis and Exposure for Time-series (ALSET), designed to operate within the latent space of any representation learning backbone, leveraging learned embeddings to enhance OSR performance for time-series. ALSET is an outlier exposure mechanism that generates outliers by modeling the empty space around the samples from known classes using multiple Gaussians and sampling from them. It constructs local Gaussian distributions centered on known samples in the latent space, with learnable, per-dimension standard deviations, which are estimated by a neighborhood estimator. These distributions expand during training through a feedback loop between the classifier and the neighborhood estimator, which learns the structure of these Gaussians while preventing overlap with known samples. For evaluation, we adapt several state-of-the-art OSR techniques, originally designed for computer vision, to the time-series domain for the first time, establishing a comprehensive baseline for this underexplored area. Extensive experiments on UCR, UEA, and HAR benchmarks demonstrate that ALSET consistently surpasses these baselines, achieving state-of-the-art OSR performance while preserving known-class F1. The code will be made publicly available upon acceptance.

## 1 INTRODUCTION

Time-series data are crucial across a wide range of domains, including healthcare Wang et al. (2024); Bhatti et al. (2024); Shome et al. (2024), finance Cheng et al. (2022), and environmental science Wang et al. (2022); Gruca et al.. Accordingly, effective time-series representation learning has received considerable attention for downstream applications such as classification Sarkar et al. (2024), forecasting Cao et al. (b); Jin et al. (2024); Grover et al. (2024), and anomaly detection Liu & Chen (2024). In particular, classification of time-series data has emerged as a fundamental task due to its broad applicability in areas such as disease diagnosis Shome et al. (2024), fraud detection Cheng et al. (2022), and weather classification for climate analysis Ham et al. (2019).

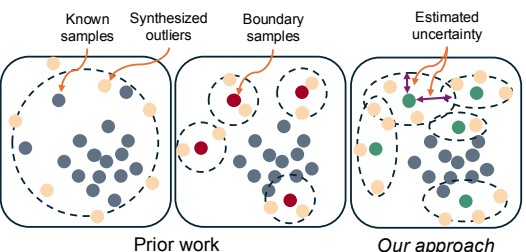

Figure 1: Comparison of synthetic outlier generation methods. (Left) Global parametric methods, such as VOS, rely on Gaussian assumptions, producing outliers in low-likelihood regions but failing to capture local data complexity. (Center) NPOS improves flexibility by identifying boundary samples via $k$NN, but fixed standard deviations limit adaptation to anisotropic regions. (Right) Our proposed method introduces adaptive, per-dimension uncertainty, generating diverse outliers that better capture the true structure of the unknown space.

Despite these advancements in time-series classification, machine learning solutions generally operate under a '*closed-set*' assumption, i.e., that the training data and test data share the same label set Salehi et al.. However, real-world environments often present unpredictable scenarios where models encounter unseen classes that are not necessarily included in the original training data.

This has resulted in a challenging new area of research called Open Set Recognition (OSR) Geng et al. (2020); Wu et al. (2024), which allows models to accurately classify known classes while also detecting and rejecting unknown ones. While OSR in domains such as computer vision has seen exciting advancements over the past few years Saito et al. (2021); Vaze et al. (2022); Cen et al. (2023), it remains largely unexplored in the context of time-series classification.

OSR methods generally operate by identifying potential unknown samples using prediction confidence scores, such as softmax probabilities Hendrycks & Gimpel (2016); Cen et al. (2023) and energy scores Liu et al. (2020); Vaze et al. (2022). Since models in OSR are trained only on samples from the known classes, they may exhibit overconfidence when encountering samples from unknown classes Bai et al. (2021). This overconfidence occurs because traditional classification models, particularly those using softmax-based cross-entropy loss, are optimized to assign high probabilities to one of the known classes for any input, including inputs from unknown distributions.

This limitation has led to the development of techniques such as Outlier Exposure (OE) Hendrycks et al. (2018), which explicitly incorporates samples from unknown classes, referred to as *outliers*, during classifier training to enhance the model's ability to distinguish between known and unknown classes. By doing so, OE reduces the model's confidence when encountering unknown samples while maintaining its confidence on known classes.

To obtain samples with unknown classes to use as outliers in OE methods, auxiliary datasets Chen et al. (2021); Du et al. (2022a); Ming et al. (2022b) have been used in prior works. Although auxiliary datasets may provide some insight into unknown classes, their fixed distribution of samples and labels limits their ability to capture the full diversity of the OSR space. To address these limitations, recent advancements have focused on *synthetic* outlier generation Du et al. (2022b); Tao et al. (2023), which aims to generate the synthetic samples within the low-density regions (the empty space around boundary samples that surround the samples from known classes).

Among the existing outlier synthesizing methods in the vision domain, VOS Du et al. (2022b) assumes class-conditional Gaussian distributions for known classes and generates synthetic outliers by sampling from low-likelihood regions. However, this approach relies on the assumption of a Gaussian structure, which often fails to represent complex data. Another technique, NPOS Tao et al. (2023), improves on this by identifying boundary samples using k-nearest neighbor density estimation and generating synthetic outliers around these points, where the outliers are sampled from Gaussian distributions centered around the boundary samples with a fixed standard deviation. Although more flexible than VOS, NPOS still faces limitations. For instance, its reliance on $k$NN for boundary sample detection introduces challenges, as $k$NN struggles to effectively capture boundaries in data distributions with mixed sparse and dense regions. Furthermore, its fixed standard deviation fails to capture the variations in low-density regions, which differ across dimensions and samples. These limitations lead us to the following question: *How can we capture local low-density regions around each sample, accounting for variations across dimensions, to generate diverse synthetic outliers that effectively represent the OSR space?*

To address this question, we propose **A**daptive **L**ocalized latent outlier **S**ynthesis and **E**xposure for **T**ime-series Open-set Recognition (ALSET). ALSET models local low-density regions around each sample from known classes by learning an adaptive, per-dimension standard deviation to parameterize a Gaussian centered around the sample, as depicted in Figure 1. To achieve this, an estimator generates a standard deviation for each input sample, which is used to parameterize a local Gaussian centered around that data point in the latent space. Synthetic outliers are then sampled from this Gaussian and exposed to the model. The classifier is trained using both the input sample with its ground-truth label and the generated outlier with a uniform label distribution over known classes. Our optimization process encourages the standard deviation to increase, allowing the generation of a broader range of diverse outliers. However, if the standard deviation becomes excessively large and the generated outliers overlap with known samples, the classifier's confidence on the known samples decreases. Since the classifier is optimized to maintain high confidence on known samples, this feedback penalizes the standard deviation encoder, preventing further overlap. Furthermore, the principle of OE ensures that the model reduces its confidence on synthetic outliers, creating a dynamic that drives the generation of outliers outside the known regions. As a result, our approach adaptively captures the heterogeneous structure of empty regions, which addresses the limitations of existing methods. ALSET operates directly in the latent space, where temporal features, periodic structures, and complex dependencies have already been captured by time-series encoders, allow-

ing our method to remain compatible with any time-series representation encoder. To establish a comprehensive baseline, we adapt several state-of-the-art OSR methods, originally developed for vision, to the time-series domain, and evaluate our approach on three widely used public benchmarks, namely UCR Dau et al. (2018), UEA Bagnall et al. (2018), and HAR Asuncion et al. (2007). Our method achieves state-of-the-art performance over various OSR methods using several encoders. Detailed ablation studies demonstrate the impact of each component of our method. We will make our implementation public upon publication of the paper.

We summarize our contributions as follows. (**1**) We propose ALSET for OSR in time-series data. Our framework adaptively models low-density regions around samples from known classes using learned per-dimension standard deviations to parameterize local Gaussians, enabling precise and adaptive synthetic outlier generation for the unique distribution of a given known space. (**2**) ALSET integrates the outlier generation step with classifier training by a feedback mechanism where the classifier penalizes excessive standard deviation growth if synthetic outliers overlap with known samples. This dynamic ensures the generation of diverse outliers that effectively represent the OSR space. (**3**) To the best of our knowledge, ALSET is the first learning method that explicitly targets OSR in time-series data. ALSET can be constructed on top of any pre-trained time-series encoders without modifying the encoder. (**4**) Extensive experiments on diverse time-series datasets demonstrate that ALSET achieves state-of-the-art performance, significantly improving unknown class detection while maintaining high accuracy for known classes. To enable reproducibility and contribute to the area, the code will be made publicly available upon acceptance

## 2 RELATED WORK

**Open-set recognition.** Early OSR methods transform the detection of unknowns into a binary classification problem by thresholding confidence scores. Baseline approaches include maximum softmax probability (MSP) Hendrycks & Gimpel (2016), energy-based scoring Liu et al. (2020), and distance-based metrics such as Mahalanobis distance Lee et al. (2018b) and nearest-neighbor statistics Sun et al. (2022). Ensemble-based uncertainty estimates have also been explored Lakshminarayanan et al. (2017). To improve calibration, several inference-time interventions have been proposed, including ODIN Liang et al. (2017) and ReAct Sun et al. (2021). More recent approaches such as ASH Djurisic et al. (2022) and Scale Xu et al. (2023) simplify model activations to enhance unknown detection with minimal overhead.

**Outlier exposure.** OE-based methods improve unknown detection by introducing auxiliary outliers during training. Hendrycks et al. Hendrycks et al. (2018) first proposed using external datasets for OE; subsequent work has focused on selecting more informative samples. ATOM Chen et al. (2021) and POEM Ming et al. (2022b) mine boundary-near or uncertain examples, while DivOE Zhu et al. (2023) enhances diversity among outliers. MixOE Zhang et al. (2023) interpolates between known and unknown classes to construct effective training samples. Generative methods such as GANs Lee et al. (2018a) and diffusion models Du et al. (2024) have been used to synthesize unknowns in pixel space. Vision-language models like CLIP have also been employed for zero-shot detection (ZOC Esmaeilpour et al. (2022), CLIPN Wang et al. (2023), EOE Cao et al. (a)), leveraging text-conditioned representations for generating semantic outliers. Rather than operating in input space, latent methods generate synthetic unknowns directly in feature space. VOS Du et al. (2022b) samples virtual outliers from low-density regions of class-conditional distributions. NPOS Tao et al. (2023) adopts a non-parametric framework, and DREAM-OOD Du et al. (2024) employs text-driven diffusion to produce low-likelihood representations.

**Time-series OSR.** OSR in time-series remains underexplored compared to vision. Prior work includes embedding-based generalization for OOD detection Lu et al. (2024) and seasonal feature modeling Belkhouja et al. (2023). Our approach addresses this gap by synthesizing adaptive latent outliers during training, and is compatible with recent time-series representation learning methods Zhang et al. (2024); Goswami et al.; Jin et al. (2024); Cao et al. (b); Gruver et al. (2024). Unlike vision methods that operate on spatial features or image pixels, our method adapts to the sequential nature and modality-specific variances of time-series data by generating outliers in a feature-dense latent space produced by time-series encoders. This enables modeling dynamics that are common in real-world temporal datasets.

A more detailed discussion of related work is provided in the appendix.

## 3 METHOD

### 3.1 PRELIMINARIES

Let $\mathcal{D}_{\text{train-closed}} = \{(\mathbf{x}_i, y_i)\}_{i=1}^{N} \subset \mathcal{X} \times \mathcal{C}$ represent a labeled training dataset, where $\mathcal{X}$ denotes the space of multivariate time-series inputs, and $\mathcal{C}$ is the set of known classes. In the closed-set scenario, a classifier model is evaluated on a test dataset where all labels are strictly drawn from the set of known classes $\mathcal{C}$, therefore $\mathcal{D}_{\text{test-closed}} = \{(\mathbf{x}_i, y_i)\}_{i=1}^{M} \subset \mathcal{X} \times \mathcal{C}$. Here, $N$ and $M$ denote the number of samples in the training and test datasets, respectively. A closed-set model predicts a posterior probability distribution over the known classes, represented as $p(y \mid \mathbf{x})$ for $y \in \mathcal{C}$.

In contrast, an open-set model assumes test samples may originate from unknown classes, denoted as $\mathcal{U}$; therefore, $\mathcal{D}_{\text{test-open}} = \{(\mathbf{x}_i, y_i)\}_{i=1}^{M'} \subset \mathcal{X} \times (\mathcal{C} \cup \mathcal{U})$, where $\mathcal{C} \cap \mathcal{U} = \emptyset$. In this scenario, the model is tasked with two complementary objectives: computing the posterior distribution $p(y \mid \mathbf{x}, y \in \mathcal{C})$ to classify samples within the known class set $\mathcal{C}$, and estimating a confidence score $S(y \in \mathcal{C} \mid \mathbf{x})$ to determine whether a test sample belongs to the known classes or originates from the unknown set $\mathcal{U}$. The decision regarding whether a test sample belongs to the known classes $\mathcal{C}$ or the unknown set $\mathcal{U}$ is based on a thresholding mechanism defined as:

$$Decision(\mathbf{x}) = \begin{cases} \text{Known} & \text{if } S(\mathbf{x}) \geq \gamma \\ \text{Unknown} & \text{if } S(\mathbf{x}) < \gamma, \end{cases} \quad (1)$$

where $\gamma$ is the decision threshold. Samples with confidence scores $S(\mathbf{x})$ exceeding $\gamma$ are classified as belonging to the known classes $\mathcal{C}$, while those with scores below $\gamma$ are classified as unknown. A widely used scoring method for open-set detection is the Maximum Softmax Probability (MSP) Hendrycks & Gimpel (2016), where the confidence score is defined as the maximum predicted probability over the known classes, $S_{\text{MSP}}(\mathbf{x}) = \max_{y \in \mathcal{C}} p(y \mid \mathbf{x})$. However, classifiers may show overconfidence when encountering samples from unknown classes given that softmax losses such as cross-entropy, aim to assign high probabilities to one of the known classes for any input, including inputs from unknown distributions. One solution to mitigate this overconfidence is to regulate the confidence of the model during training. A common method for achieving this is OE Hendrycks et al. (2018).

The concept of OE involves training the model against an auxiliary or synthesized dataset of samples from unknown classes, denoted as $\mathcal{D}_{\text{unknown}}^{\text{OE}}$, to improve the model's ability to distinguish between samples from known and unknown classes. Let $f : \mathcal{X} \to \mathcal{C}$ represent the model, which maps an input $\mathbf{x} \in \mathcal{X}$ to a class label $y \in \mathcal{C}$. With a primary learning objective $\mathcal{L}$, OE optimizes the following objective:

$$\mathbb{E}_{(\mathbf{x}, y) \sim \mathcal{D}_{\text{train-closed}}} \left[ \mathcal{L}(f(\mathbf{x}), y) \right] + \lambda \mathbb{E}_{\mathbf{x}' \sim D_{\text{unknown}}^{\text{OE}}} \left[ \mathcal{L}_{\text{OE}}(f(\mathbf{x}')) \right], \quad (2)$$

where $\mathcal{L}_{\text{OE}}$ is task-dependent and chosen based on the downstream task. For instance, when using the MSP scoring method Hendrycks & Gimpel (2016) to differentiate between known and unknown samples, the most effective choice for $\mathcal{L}_{\text{OE}}$ is the cross-entropy between $f(\mathbf{x}')$ and the uniform distribution over known classes. This formulation encourages the model $f$ to minimize its confidence in regions corresponding to unknown data, thereby enhancing the scoring method's capacity to distinguish samples from known classes versus those from unknown classes. The primary challenge in OE lies in obtaining $\mathcal{D}_{\text{unknown}}^{\text{OE}}$, which must be derived from appropriate auxiliary datasets or synthetically generated to sufficiently represent unknown samples.

### 3.2 OUR APPROACH

Assume the model $f$ is composed of a time-series encoder $Enc$, which captures temporal dependencies and contextual features, and a classifier head $h$. The encoder $Enc : \mathcal{X} \to \mathcal{Z}$, where $\mathcal{Z} \subseteq \mathbb{R}^d$ is the $d$-dimensional latent space. The classifier head $h : \mathcal{Z} \to \mathcal{C}$ operates on the latent embeddings $\mathbf{z}$ to predict the posterior probability distribution over the set of known classes $\mathcal{C}$, $p(y \mid \mathbf{x}) = h(Enc(\mathbf{x})) = h(\mathbf{z})$. We generate synthetic unknown samples in the latent space $\mathcal{Z}$ to be used as outliers and expose them to the classifier $h$. To clearly differentiate between embeddings of known (real) samples and generated unknown samples, we denote the embeddings of known samples as $\mathbf{z}^{\text{known}} \subseteq \mathcal{Z}^{\text{known}}$ and the generated unknown samples as $\mathbf{z}^{\text{unknown}} \subseteq \mathcal{Z}^{\text{unknown}}$.

To generate the unknown samples $\mathbf{z}_i^{\text{unknown}}$ (in the latent space), we aim to capture the empty space around a given point $\mathbf{z}_i^{\text{known}}$ and generate the unknown samples within this area. Therefore, we

construct a local multivariate Gaussian distribution around $\mathbf{z}_i^{\text{known}}$. However, to adaptively fit this Gaussian to the empty space, we need to learn the standard deviations across different dimensions of our latent space. Therefore, given a $\mathbf{z}_i^{\text{known}}$, we use an estimator head, which we refer to as neighborhood estimator $q$, to predict classifier uncertainty values via standard deviations $\boldsymbol{\sigma}_i = q(\mathbf{z}_i^{\text{known}})$. We then sample the synthetic unknowns from the multivariate Gaussian with the predicted standard deviation. To ensure that the sampling process remains differentiable to enable the gradients to flow through the predicted $\boldsymbol{\sigma}_i$, we employ the reparametrization technique for sampling and generate the unknown samples as:

$$\mathbf{z}_i^{\text{unknown}} = \mathbf{z}_i^{\text{known}} + \boldsymbol{\epsilon}, \quad \boldsymbol{\epsilon} \sim \mathcal{N}(\mathbf{0}, \text{diag}(\boldsymbol{\sigma}_i^2)), \tag{3}$$

where $\text{diag}(\boldsymbol{\sigma}_i^2)$ is a diagonal covariance matrix determined by the predicted standard deviation vector $\boldsymbol{\sigma}_i$.

In practice, The classifier $h$, operating on latent embeddings, is trained using both embeddings of known samples $\mathbf{z}_i^{\text{known}}$ and synthetic unknown samples $\mathbf{z}_i^{\text{unknown}}$, predicting a probability distribution over the known classes $\mathcal{C}$.

To facilitate outlier exposure and jointly train the classifier to accurately classify known samples while reducing its confidence on generated unknown samples, we utilize a unified cross-entropy loss:

$$\mathcal{L}_{\text{CE}} = -\mathbb{E}_{\mathbf{z} \sim \mathcal{Z}} \left[ \mathbb{I}(\mathbf{z} \in \mathcal{Z}_{\text{known}}) \log p_\phi(y \mid \mathbf{z}) \right.$$

$$\left. + \mathbb{I}(\mathbf{z} \in \mathcal{Z}_{\text{unknown}}) \frac{\lambda}{|\mathcal{C}|} \sum_{c \in \mathcal{C}} \log p_\phi(c \mid \mathbf{z}) \right] = \mathcal{L}_{\text{CE}}^{known} + \mathcal{L}_{\text{CE}}^{unknown} \tag{4}$$

where $y \in \mathcal{C}$ is the true label corresponding to a sample embedding $\mathbf{z} \in \mathcal{Z}^{\text{known}}$, $c \in \mathcal{C}$ represents a class in the set of known classes, and $\mathbb{I}(\cdot)$ is an indicator function that distinguishes between embeddings of known samples ($\mathbf{z}_i^{\text{known}}$) and synthetic unknown samples ($\mathbf{z}_i^{\text{unknown}}$).

For embeddings of known samples, the classifier minimizes the cross-entropy loss with respect to the true label $y$, ensuring accurate classification. For synthetic unknown samples, the classifier reduces its confidence by predicting a uniform probability distribution over all known classes $\mathcal{C}$, scaled by the weight $\lambda$.

Next, to encourage diversity among synthetic unknown samples and ensure their separation from the embeddings of known samples, We expand the local Gaussian distributions constructed around each known sample by maximizing its predictive uncertainty, formalized as:

$$\mathcal{L}_{\text{STD}} = -\mathbb{E}_{\mathbf{z}_i^{\text{known}} \sim \mathcal{Z}_{\text{known}}} \left[ |\boldsymbol{\sigma}_i|_2^2 \right], \tag{5}$$

where $\boldsymbol{\sigma}_i$ is the standard deviation vector produced by the neighborhood estimator $q$ for the embedding $\mathbf{z}_i^{\text{known}}$. However, aggressive expansion of $\boldsymbol{\sigma}_i$ can cause synthetic unknown samples to intrude into the latent space of known classes, thereby degrading classification performance.

**Adaptive Uncertainty Regulation.** To prevent destructive overlap, we modulate the regularization term in Eq. equation 5 based on the classifier's stability. Specifically, we track an exponential moving average of the classification loss for known samples, and define a normalized deviation:

$$\delta(t) = \frac{\mathcal{L}_{\text{CE}}^{\text{known}}(t) - \mathcal{L}_{\text{CE}}^{\text{known}}(t-1)}{\mathcal{L}_{\text{CE}}^{\text{known}}(t) + \varepsilon}, \tag{6}$$

where $\varepsilon$ is a small constant for numerical stability. We then revise the uncertainty regularizer in equation 5 as:

$$\mathcal{L}_{\text{STD}} = \mathbb{E}_{\mathbf{z}_i^{\text{known}}} \left[ \tanh(\delta(t)) \cdot |\boldsymbol{\sigma}_i|_2^2 \right]. \tag{7}$$

When the classifier loss increases relative to its moving average ($\delta(t) > 0$), the $\tanh$ factor is positive, penalizing large uncertainties. Conversely, when the classifier improves ($\delta(t) < 0$), the penalty weakens and becomes negative, promoting controlled expansion. Furthermore, if $\delta(t) = 0$, we add a small constant $\varepsilon_p$ to ensure that the uncertainty regularization remains active and gradients do not vanish entirely. This feedback loop allows the model to balance uncertainty growth with classification robustness and ensures that $\boldsymbol{\sigma}_i$ remains within a reasonable range, where synthetic embeddings of unknown samples are diverse enough to effectively challenge the classifier's decision boundaries, yet remain separate from the latent regions of known samples. It should be noted that for

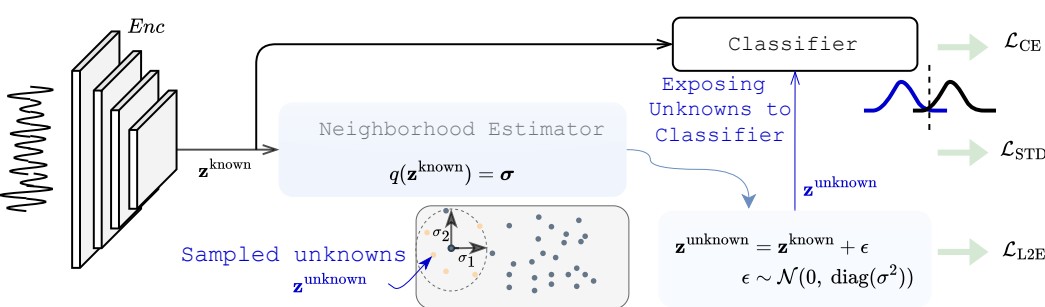

Figure 2: Framework of ALSET. Known feature embeddings ($\mathbf{z}^{\text{known}}$) are extracted through a time-series encoder and passed to a neighborhood estimator ($q$) to capture uncertainty by learning per-dimension variance ($\sigma$). Synthetic unknowns ($\mathbf{z}^{\text{unknown}}$) are then generated using this variance and exposed to the classifier.

known samples located within dense regions of the latent space, the classifier inherently restricts the growth of $\boldsymbol{\sigma}$. As a result, the learned standard deviations in these regions approach zero, effectively collapsing the distribution. In theory, Gaussians with such small standard deviations are unable to construct meaningful distributions and are thus disregarded.

Meanwhile, in regions not restricted by any sample from known classes, $\boldsymbol{\sigma}$ can grow excessively large due to the absence of constraints introduced by the classifier, allowing synthetic unknown samples to scatter far from the regions of known samples. These distant samples lose their relevance as meaningful outliers and disrupt the stability of the training process, as $\boldsymbol{\sigma}$ may grow indefinitely. To ensure that the synthetic unknown samples remain effective, we incorporate latent space regularization by constraining the overall embedding magnitudes using an $\ell_2$-regularization term applied to the latent representations:

$$\mathcal{L}_{\text{L2E}} = \mathbb{E}_{\mathbf{z} \sim \mathcal{Z}} \left[ \|\mathbf{z}\|_2^2 \right], \tag{8}$$

where $\mathbf{z} \in \mathcal{Z}$ includes both embeddings of known samples ($\mathbf{z}^{\text{known}}$) and synthetic unknown samples ($\mathbf{z}^{\text{unknown}}$). This regularization encourages the entire embedding space to align with a standard normal distribution, as discussed in Nguyen et al. (2022), by penalizing embeddings with excessively large values.

The overall training objective integrates the cross-entropy loss for classification with regularization terms for embedding dispersion, giving:

$$\mathcal{L} = \mathcal{L}_{\text{CE}} + \alpha \mathcal{L}_{\text{STD}} + \beta \mathcal{L}_{\text{L2E}}, \tag{9}$$

where $\alpha$ and $\beta$ are the hyperparameters that balance the contributions of dispersion maximization and latent space regularization, respectively. Figure 2 presents the architecture of our method.

## 4 EXPERIMENTS

### 4.1 EXPERIMENT SETUP

**Datasets.** We use the **UCR** Dau et al. (2018), **UEA** Bagnall et al. (2018), and **HAR** Asuncion et al. (2007) datasets to construct OSR datasets and evaluate our proposed method, as follows. The UCR repository consists of 128 unique univariate datasets. For our experiments, we create open-set versions of each dataset by designating the last class as the open-set (unknown) class. This approach is taken for all datasets with more than two classes, resulting in 83 datasets being selected for evaluation (exact dataset details are provided in the Appendix). The UEA repository consists of 30 multivariate datasets, for which we adopt the same methodology as UCR by designating the last class in each dataset as the unknown class. Datasets with fewer than three classes, which are unsuitable for constructing an open-set class, are excluded from our evaluations. This results in 28 datasets being selected from the UEA repository for evaluation (details provided in Appendix). A subset of UCR datasets have been z-score normalized, whereas the UEA datasets have not. We use all individual datasets as provided in the repositories. We also utilize the Human Activity Recognition (HAR) dataset from the UCI repository, which contains recordings of 30 individuals doing daily

Table 1: Average results for the datasets from the UCR repository.

| Method | Encoder | F1↑ (known) | FPR95↓ | | AUROC↑ | | AUPR (known)↑ | | AUPR (unknown)↑ | |
|---|---|---|---|---|---|---|---|---|---|---|
| | | | Energy | MSP | Energy | MSP | Energy | MSP | Energy | MSP |
| None | TS2VEC | 76.94 | 69.21 | 69.42 | 64.41 | 64.55 | 62.61 | 62.60 | 68.42 | 68.58 |
| ASH-p | TS2VEC | 36.50 | 81.58 | 83.56 | 53.40 | 53.28 | 52.93 | 53.09 | 59.18 | 59.78 |
| ASH-s | TS2VEC | 36.50 | 82.02 | 94.57 | 51.52 | 50.68 | 51.52 | 50.67 | 58.01 | 65.51 |
| Scale | TS2VEC | 76.94 | 78.92 | 81.57 | 57.45 | 57.28 | 55.94 | 55.80 | 61.99 | 62.33 |
| ReAct | TS2VEC | 76.94 | 69.75 | 69.91 | 63.99 | 64.07 | 62.19 | 62.24 | 68.05 | 68.22 |
| VOS | TS2VEC | 76.40 | 69.13 | 70.54 | 64.39 | 65.47 | 62.53 | 63.45 | 68.26 | 70.06 |
| NPOS | TS2VEC | 77.08 | 70.04 | 71.38 | 65.52 | 65.92 | 62.42 | 63.82 | 69.23 | 69.15 |
| **ALSET (ours)** | TS2VEC | 77.10 | **65.51** | **64.90** | **66.49** | **67.21** | **64.42** | **64.89** | **70.28** | **70.95** |
| None | SoftCLT | 74.33 | 71.41 | 69.36 | 61.30 | 63.44 | 60.07 | 61.58 | 66.40 | 68.07 |
| ASH-p | SoftCLT | 37.31 | 82.47 | 84.17 | 52.24 | 52.12 | 52.24 | 52.06 | 58.39 | 59.13 |
| ASH-s | SoftCLT | 37.31 | 81.17 | 94.52 | 52.92 | 51.18 | 52.41 | 51.49 | 59.06 | 65.64 |
| Scale | SoftCLT | 76.97 | 78.77 | 80.76 | 57.09 | 56.86 | 55.74 | 55.71 | 61.85 | 62.28 |
| ReAct | SoftCLT | 76.97 | 68.26 | 68.00 | 65.07 | 65.82 | 63.16 | 63.67 | 69.06 | 69.61 |
| VOS | SoftCLT | 77.33 | 66.75 | 68.94 | 67.09 | 66.91 | **64.71** | 64.26 | 70.45 | 71.53 |
| NPOS | SoftCLT | 77.23 | 68.21 | 68.15 | 66.23 | 66.73 | 64.03 | 46.19 | 69.72 | 70.12 |
| **ALSET (ours)** | SoftCLT | 77.05 | **63.44** | **63.34** | **67.31** | **67.75** | 64.14 | **64.42** | **71.66** | **71.96** |

activities while wearing a waist-mounted smartphone. In our analysis, we designate the class, *Lying*, as the unknown class. Additionally, the Appendix includes experiments using different classes as the unknown class.

**Baselines.** We benchmark ALSET against different OSR approaches: ASH-p and ASH-s Djurisic et al. (2022), Scale Xu et al. (2023), ReAct Sun et al. (2021), VOS Du et al. (2022b), and NPOS Tao et al. (2023). All these solutions have been originally proposed for the vision domain, which we carefully adapt to time-series. ASH, Scale, and ReAct aim to refine the latent space for better separation of known and unknown samples, while VOS and NPOS are outlier exposure methods that synthesize unknown samples.

**Encoders.** Since ALSET is designed to operate on top of any embedding method, we evaluate it on three popular state-of-the-art time-series representation learning pipelines: TS2VEC Yue et al. (2022), SoftCLT Lee et al. (2024), and Moment Goswami et al.. We train TS2VEC and SoftCLT separately on data from known classes for each dataset from the UCR and UEA repositories. In contrast, Moment is a time-series *foundation model* pre-trained on a large collection of datasets, including UCR and UEA. This violates the OSR assumption of encountering unknown classes. Therefore, we evaluate Moment using the HAR dataset.

For all three baselines, we apply all OSR solutions (including ours) to the classification heads, which are trained from scratch on the created open-set datasets and not jointly trained with the encoder. Moment supports both classification and forecasting tasks, making it suitable for our experiments compared to other recent time-series foundation models that focus predominantly on forecasting, like Jin et al. (2024); Cao et al. (b); Gruver et al. (2024).

**Evaluation.** To comprehensively measure the performance of all OSR methods, we apply two widely adopted evaluation approaches, MSP Hendrycks & Gimpel (2016) and Energy Liu et al. (2020). We evaluate our method using threshold-free metrics: AUROC (Area Under the Receiver Operating Characteristic curve), AUPR (Area Under the Precision-Recall curve), and FPR95 (False Positive Rate at 95% True Positive Rate), where FPR95 measures the probability that a sample from an unknown class is incorrectly classified as one of the known classes when the true positive rate reaches 95%. For AUPR, we present results for two cases: when known samples are treated as positive samples AUPR (known) and when unknown samples are treated as positive samples AUPR (unknown). Additionally, we report the F1 scores for the known samples. While the primary focus of OSR is not to optimize performance on known classes, an effective OSR technique should maintain high classification rates for the known samples, ensuring that performance on known samples is not compromised.

## 4.2 RESULTS

**Performance and comparisons.** We report the average results of 5 runs of our method on the UCR and UEA datasets in Tables 1 and 2, with detailed results for each dataset provided in the Appendix. Table 1 demonstrates that ALSET consistently outperforms other OSR methods, regardless of the encoder used. Although the F1 score calculated on known samples is generally at risk of a decline

Table 2: Average results for the datasets from the UEA repository.

| Method | Encoder | F1↑ (known) | FPR95↓ Energy | MSP | AUROC↑ Energy | MSP | AUPR (known)↑ Energy | MSP | AUPR (unknown)↑ Energy | MSP |
|---|---|---|---|---|---|---|---|---|---|---|
| None | TS2VEC | 66.63 | 68.56 | 71.03 | 62.40 | 61.40 | 61.94 | 65.47 | 65.73 | 69.32 |
| ASH-p | TS2VEC | 55.53 | 81.41 | 82.39 | 54.30 | 54.04 | 53.79 | 58.00 | 59.26 | 63.40 |
| ASH-s | TS2VEC | 55.53 | 81.92 | 89.45 | 55.97 | 55.13 | 54.48 | 58.22 | 59.87 | 64.55 |
| Scale | TS2VEC | 66.08 | 73.64 | 75.11 | 60.62 | 59.71 | 59.55 | 63.47 | 63.93 | 67.75 |
| ReAct | TS2VEC | 66.08 | **67.11** | 68.80 | 63.63 | 63.44 | **62.87** | 66.45 | **67.30** | 70.95 |
| VOS | TS2VEC | 66.47 | 68.05 | 69.39 | 62.58 | 62.10 | 62.41 | 65.79 | 66.13 | 69.46 |
| NPOS | TS2VEC | 66.36 | 69.15 | 70.82 | 61.93 | 61.28 | 60.14 | 65.74 | 65.63 | 69.18 |
| **ALSET (ours)** | TS2VEC | 66.37 | 67.82 | **68.79** | **63.75** | **63.76** | 62.50 | **66.96** | 67.13 | **71.21** |
| None | SoftCLT | 65.59 | 68.90 | 71.35 | 62.87 | 61.23 | 62.49 | 64.70 | 66.26 | 69.48 |
| ASH-p | SoftCLT | 54.65 | 79.82 | 81.79 | 55.46 | 53.84 | 54.72 | 57.46 | 60.28 | 63.67 |
| ASH-s | SoftCLT | 54.65 | 79.04 | 86.71 | 57.46 | 55.85 | 55.94 | 59.12 | 61.61 | 65.63 |
| Scale | SoftCLT | 65.59 | 73.17 | 75.02 | 61.48 | 60.69 | 60.70 | 63.69 | 64.85 | 68.85 |
| ReAct | SoftCLT | 65.59 | 67.28 | 68.90 | 63.34 | 62.82 | 62.20 | 65.39 | **67.63** | **71.18** |
| VOS | SoftCLT | 65.61 | 66.01 | **70.02** | 63.28 | 61.76 | **62.70** | 56.52 | 65.85 | 69.02 |
| NPOS | SoftCLT | 65.36 | 69.75 | 70.93 | 62.72 | 60.31 | 61.83 | 64.19 | 66.39 | 69.26 |
| **ALSET (ours)** | SoftCLT | 65.68 | **64.74** | 70.16 | **64.02** | **62.95** | 62.66 | **65.87** | 67.33 | 70.79 |

Table 3: Results on the HAR dataset.

| Method | Encoder | F1↑ (known) | FPR95↓ Energy | MSP | AUROC↑ Energy | MSP | AUPR (known)↑ Energy | MSP | AUPR (unknown)↑ Energy | MSP |
|---|---|---|---|---|---|---|---|---|---|---|
| None | Moment | 81.00 (0.00) | 79.00 (7.97) | 64.60 (6.66) | 51.60 (6.11) | 67.80 (3.03) | 61.20 (6.30) | 74.40 (3.36) | 47.40 (3.44) | 59.60 (2.79) |
| ASH-p | Moment | 28.20 (20.32) | 94.00 (5.96) | 91.00 (10.84) | 47.20 (15.45) | 51.00 (13.17) | 50.00 (11.22) | 52.60 (11.70) | 50.40 (12.76) | 50.60 (9.96) |
| ASH-s | Moment | 28.20 (20.61) | 89.60 (13.58) | 88.60 (16.15) | 50.20 (18.16) | 52.00 (14.44) | 53.00 (14.82) | 54.00 (14.85) | 52.60 (14.05) | 52.00 (10.37) |
| Scale | Moment | 81.00 (0.00) | 93.80 (3.83) | 94.80 (4.97) | 51.60 (11.33) | 49.00 (6.67) | 52.20 (7.82) | 49.80 (3.70) | 52.60 (11.04) | 51.80 (7.46) |
| ReAct | Moment | 80.00 (0.00) | 79.00 (7.97) | 64.60 (6.66) | 52.60 (6.11) | 68.80 (3.03) | 61.20 (6.30) | 74.40 (3.36) | 47.40 (3.44) | 59.60 (2.79) |
| VOS | Moment | 75.80 (1.95) | 82.10 (4.80) | 85.20 (5.25) | 54.30 (4.20) | 62.40 (4.00) | 61.50 (3.80) | 63.40 (4.10) | **55.63 (12.32)** | 61.00 (4.00) |
| NPOS | Moment | 77.10 (2.00) | 80.90 (5.00) | 83.80 (5.40) | 52.20 (3.90) | 68.50 (3.80) | 63.70 (3.70) | 65.20 (3.80) | 54.83 (4.97) | 61.80 (3.70) |
| **ALSET (ours)** | Moment | 77.20 (2.49) | **73.20 (6.49)** | **57.70 (9.57)** | **56.30 (5.60)** | **72.90 (4.58)** | **66.30 (5.29)** | **78.90 (5.04)** | 49.40 (3.69) | **63.40 (4.17)** |
| None | TS2VEC | 91.80 (0.45) | 70.20 (12.97) | 72.60 (25.74) | 59.80 (13.10) | 53.40 (19.05) | 67.60 (10.99) | 61.40 (19.13) | 56.80 (12.13) | 51.00 (11.96) |
| ASH-p | TS2VEC | 77.00 (7.14) | 70.20 (11.69) | 47.20 (11.39) | 57.40 (9.04) | 70.00 (9.41) | 50.00 (11.22) | 79.40 (7.44) | 54.00 (8.09) | 60.80 (8.47) |
| ASH-s | TS2VEC | 85.20 (2.95) | **29.60 (13.28)** | 36.20 (10.73) | 82.40 (7.92) | 79.40 (12.01) | **88.00 (4.46)** | 86.20 (8.14) | 75.40 (10.45) | 72.40 (14.01) |
| Scale | TS2VEC | 91.80 (0.45) | 57.40 (20.11) | 72.40 (27.86) | 68.80 (10.94) | 66.80 (13.48) | 77.00 (8.77) | 71.80 (13.59) | 59.60 (9.96) | 61.60 (11.10) |
| ReAct | TS2VEC | 91.10 (0.45) | 69.20 (12.97) | 72.27 (25.74) | 60.80 (11.19) | 55.40 (19.05) | 62.40 (10.99) | 62.40 (19.13) | 56.80 (12.13) | 54.00 (11.96) |
| VOS | TS2VEC | 93.92 (0.80) | 52.53 (6.20) | 58.24 (7.10) | 79.32 (4.15) | 74.91 (4.85) | 63.83 (4.22) | 46.03 (4.75) | 75.94 (4.30) | 72.38 (4.88) |
| NPOS | TS2VEC | 92.82 (0.92) | 60.18 (5.95) | 61.03 (6.50) | 80.72 (4.50) | 73.69 (5.12) | 68.01 (4.65) | 47.03 (4.89) | 73.05 (4.78) | 72.92 (5.01) |
| **ALSET (ours)** | TS2VEC | 90.60 (1.52) | 37.00 (25.33) | **35.80 (4.87)** | **83.40 (16.46)** | **81.00 (5.15)** | 87.20 (11.80) | **87.60 (2.97)** | **77.80 (17.63)** | 69.80 (6.65) |
| None | SoftCLT | 91.64 (0.67) | 75.36 (23.96) | 76.55 (18.32) | 53.73 (19.77) | 49.82 (16.36) | 58.73 (18.22) | 57.45 (15.36) | 56.73 (14.21) | 50.55 (8.23) |
| ASH-p | SoftCLT | 79.73 (6.15) | 69.45 (30.94) | 74.64 (22.18) | 60.18 (23.26) | 53.82 (20.39) | 65.73 (19.66) | 62.09 (17.35) | 59.09 (17.06) | 53.27 (14.64) |
| ASH-s | SoftCLT | 79.55 (6.02) | 39.64 (14.91) | 45.73 (12.90) | 78.91 (11.84) | 74.27 (11.38) | 85.18 (7.76) | 82.00 (8.19) | 71.09 (13.32) | 67.36 (12.62) |
| Scale | SoftCLT | 91.64 (0.67) | 43.09 (24.70) | 51.00 (20.40) | 73.55 (17.65) | 72.73 (13.20) | 81.09 (13.39) | 79.91 (9.17) | 65.82 (16.22) | 66.00 (14.03) |
| ReAct | SoftCLT | 91.64 (0.67) | 74.73 (24.21) | 76.00 (18.97) | 54.00 (19.89) | 50.36 (16.52) | 59.27 (18.06) | 58.00 (15.56) | 56.82 (14.37) | 50.73 (8.13) |
| VOS | SoftCLT | 92.90 (0.72) | 58.36 (7.10) | 61.63 (6.88) | 73.21 (5.20) | 78.54 (5.47) | 72.09 (4.98) | 68.84 (5.04) | 64.92 (4.66) | 58.34 (4.71) |
| NPOS | SoftCLT | 92.82 (0.91) | 61.03 (6.55) | 68.62 (6.88) | 70.37 (5.30) | 73.24 (5.70) | 67.91 (5.00) | 66.25 (5.12) | 67.13 (5.11) | 61.93 (5.20) |
| **ALSET (ours)** | SoftCLT | 90.60 (0.55) | **38.40 (2.41)** | **38.60 (8.65)** | **87.00 (9.35)** | **83.80 (6.34)** | **88.40 (4.93)** | **86.40 (1.82)** | **83.40 (15.32)** | **77.80 (11.54)** |

due to the inclusion of unknowns, ALSET maintains or even slightly improves the performance across all encoders. This consistency is shared with other OE methods like VOS and NPOS. In contrast, post-hoc methods that manipulate activation functions, such as ASH, often risk degrading performance on known samples. With OSR performance as measured by the other metrics, we observe that on both univariate (UCR) and multivariate (UEA) OSR datasets, our method generally outperforms the other solutions across all three encoders. On UCR, NPOS generally achieves the second-best results for OSR, while on UEA, ReAct performs second-best.

Table 3 presents the results on the HAR dataset, demonstrating ALSET's higher performance in the majority of cases. We observe that ALSET outperforms other methods by a large margin. With TS2VEC and MSP scoring, ASH-s, and ASH-p show competitive results. Using TS2VEC, although ASH-s achieves the highest OSR performance, it significantly compromises classifier accuracy on known samples, which is undesired in OSR.

To illustrate the trade-off between closed-set classification and OSR performance, Figure 3 presents the F1 score on known classes versus Energy-based AUROC for eight OSR methods, averaged across three encoder backbones using HAR dataset. Each point represents a distinct method, with higher and more rightward placement indicating superior joint performance in both closed-set and open-set tasks.

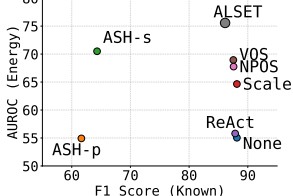

Figure 3: Scatter plot for the tradeoff analysis between averaged F1 scores (known) versus Energy-based AUROC for open-set detection across eight methods. ALSET achieves the best joint performance.

ALSET consistently achieves the best balance. In contrast, standard baselines like None and Re-Act cluster in the lower-right quadrant, reflecting strong closed-set performance but weak open-set detection. ASH variants (ASH-p, ASH-s) demonstrate a clear trade-off, sacrificing known-class accuracy for improved OSR. Although methods like VOS, NPOS, and Scale outperform traditional baselines in AUROC, they still lag behind ALSET in overall performance.

**Visualizing the Confidence Landscape.** We use the HAR dataset and the SoftCLT encoder to visualize the confidence of the model. Specifically, we explore the classifier's behavior in both the embedding space (output of the encoder $Enc$, and the logit space, which directly precedes the final classification layer. We first reduce the encoder outputs to two dimensions using PCA and construct a dense grid over this space. Each grid point is then projected back to the original feature space via inverse PCA and passed through the model to compute confidence. This produces a 2D map that captures the confidence distribution. A similar approach is applied to the logit space, where the model's logits are reduced using PCA to generate synthetic logit points for confidence estimation (Figure 4).

As illustrated in Figure 4a, ALSET effectively reduces confidence in regions that lie outside of the known sample distribution, pushing unknown samples into low-confidence zones between known classes in logit space. This contrasts with the baseline classifier (Figure 4b), which tends to assign high confidence in unknown regions, leading to poor discrimination between known and unknown inputs.

**Known vs. unknown classification results.** Building on prior works such as Cen et al. (2023) and Vaze et al. (2022), we explore the relationship between the performance of the model on samples from known classes vs. OSR performance metrics. While it has been suggested in Cen et al. (2023) and Vaze et al. (2022) that higher classifier performance leads to improved OSR outcomes, their investigations were focused on vision datasets. We extend this analysis to the area of time series for the first time and present the results in Figure 5, where we use the TS2VEC encoder to explore the relationship with and without our proposed ALSET method. The results confirm a direct relationship, supporting the fact that improving classifier performance enhances OSR metrics. By looking at the zoomed-in version of the fitted regression line in Figure A5 (Appendix), we also observe that our method improves the correlation between model performance on samples from known classes and OSR.

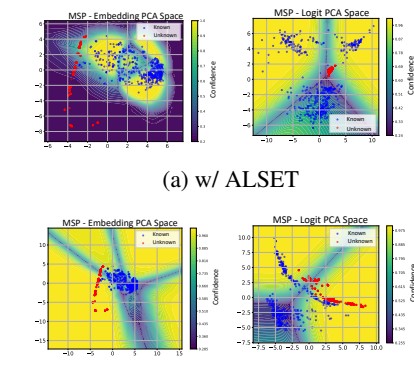

(a) w/ ALSET

(b) w/o ALSET

Figure 4: Comparison of the confidence of the model with (a) and without (b) ALSET. The latent space is depicted on the left while the logits are presented on the right. ALSET lowers confidence for unknowns near decision boundaries, while the baseline (without ALSET) assigns high confidence.

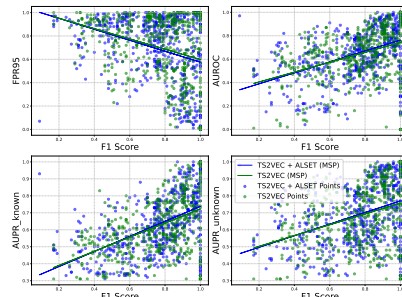

Figure 5: The performance of the model on samples from known vs. unknown classes is presented on the 83 dataset from UCR. The lines represent fitted linear regression models for the performance metrics, illustrating the relationship between the performance of the model on samples from known classes vs. OSR performance metrics. Our proposed method accelerates this improvement compared to the baseline (refer to the zoomed-in version in the Appendix Figure A5.)

## 5 CONCLUSION

We propose ALSET, a novel framework for OSR that mitigates model overconfidence in unknown regions via an outlier exposure mechanism. ALSET synthesizes outliers in the latent space by sampling from local Gaussian distributions with learnable per-dimension standard deviations, producing diverse examples that help the classifier calibrate its confidence while preserving accuracy on known classes. Compatible with any pre-trained time-series encoder, ALSET exploits their feature extraction power for effective OSR. Experiments demonstrate state-of-the-art performance across multiple datasets, enhancing unknown-class detection without degrading known-class classification.

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

## A APPENDIX

### A.1 RELATED WORK (EXTENDED)

**Open-set recognition.** A conventional approach for detecting samples from unknown classes relies on scoring methods that transform the OSR problem into a binary classification task. These methods use a score, with the most well-known being MSP Hendrycks & Gimpel (2016), and identify unknown samples by applying a threshold to the scores. Building on this foundation, more advanced methods have been developed, such as energy-based models Liu et al. (2020), which utilize logit energy to distinguish between samples from known and unknown classes. In Lee et al. (2018b), a distance-based approach was introduced to model class-conditional Gaussian distributions from deep network features and calculate the Mahalanobis distance to detect unknown samples. Another distance-based method, Sun et al. (2022), explores the effectiveness of non-parametric nearest-neighbor distances for identifying unknown samples. Additionally, ensemble techniques Lakshminarayanan et al. (2017) have been shown to effectively improve the robustness of detection by leveraging model diversity. Building upon basic scoring approaches, methods have explored inference-time interventions to enhance the reliability of methods like MSP. For instance, ODIN Liang et al. (2017) employs temperature scaling and input perturbation. Similarly, ReAct Sun et al. (2021) applies adaptive transformations to feature activations, to refine model predictions and confidence estimates. Further simplifying the process, ASH Djurisic et al. (2022) and Scale Xu et al. (2023) focus on reducing the complexity of sample activations at the penultimate layer to improve the detection of unknown samples without requiring additional data or fine-tuning on the pre-trained models.

**Outlier exposure.** Training-time regularization is another key strategy for improving the detection of samples from unknown classes by calibrating the model's confidence during the training phase, which includes OE Hendrycks et al. (2018) methods. As mentioned earlier, these solutions introduce additional data samples as outliers during training to help the model better distinguish between samples from known and unknown classes. Several proposed methods leverage the availability of auxiliary samples from unknown classes and aim to identify the most informative ones. ATOM Chen et al. (2021) performs auxiliary data mining to select the most informative samples for efficient utilization, while POEM Ming et al. (2022b) emphasizes identifying samples near the decision boundary. These approaches enhance the learning of more accurate decision boundaries between samples from known and unknown classes. DivOE Zhu et al. (2023) introduces a new learning objective to diversify auxiliary outliers by synthesizing more informative examples for extrapolation during training. MixOE Zhang et al. (2023) focuses on finding optimal outliers by mixing samples from known and available unknown classes.

Alternatively, some approaches propose synthesizing samples from unknown classes. For instance, Lee et al. (2018a) utilizes GANs to generate synthetic images in high-dimensional pixel space. More recently, large language and vision-language models like CLIP have been explored for this purpose, as proposed by Ming et al. (2022a). Building on this, methods such as CLIPN Wang et al. (2023), ZOC Esmaeilpour et al. (2022) and EOE Cao et al. (a) employ large language models to enhance the detection of unknown samples. Moreover, EOE Cao et al. (a) uses CLIP to generate potential outlier class labels for model exposure without requiring prior knowledge of these classes. However, these methods are often domain-specific and are not directly applicable to time-series data.

Generating synthetic samples in the latent space represents another promising direction in this field. VOS Du et al. (2022b) synthesizes virtual outliers that can meaningfully regularize the model's decision boundary during training by sampling from low-likelihood regions of the class-conditional distribution estimated in the feature space. NPOS Tao et al. (2023) introduces a non-parametric outlier synthesis framework that generates artificial training data for unknown samples without assuming any specific distribution for embeddings from known samples. DREAM-OOD Du et al. (2024) employs diffusion models to generate realistic outliers in pixel space by learning a text-conditioned latent space from samples of known classes and sampling outliers from low-likelihood regions.

**The gap: OSR in time-series.** While most detection methods focus on vision-based data, the time-series domain remains relatively under-explored in this context. In one of the few works in this area, out-of-distribution detection is tackled in Lu et al. (2024) alongside domain generalization by designing embedding spaces optimized for generalization. This approach relies on basic scoring methods, such as MSP Hendrycks & Gimpel (2016) and Mahalanobis distance Lee et al. (2018b),

for detecting unknown samples. In addition, Belkhouja et al. (2023) leverages seasonal patterns for detecting out-of-distribution samples in time-series data. In contrast, our work directly targets OSR by introducing a novel OE framework that synthesizes adaptive outliers in the latent space. Notably, our method can be built on top of any pre-trained time-series model, offering broad applicability. With the rapid progress in time-series representation learning Zhang et al. (2024) and the emergence of foundation models Goswami et al.; Jin et al. (2024); Cao et al. (b); Gruver et al. (2024) for time-series, which provide robust pre-trained embeddings, our framework complements these advancements to enable enhanced OSR performance through adaptive outlier synthesis and exposure.

## A.2 IMPLEMENTATION DETAILS

**Evaluation metrics.** Although these evaluation metrics are already discussed in the main text, we provide an explanation here for clarity and completeness before providing detailed results on different datasets from UCR and UEA. To comprehensively measure the performance of all OSR methods, we apply two widely adopted evaluation approaches, MSP Hendrycks & Gimpel (2016) and Energy Liu et al. (2020). We evaluate our method using threshold-free metrics: AUROC (Area Under the Receiver Operating Characteristic curve), AUPR (Area Under the Precision-Recall curve), and FPR95 (False Positive Rate at 95% True Positive Rate), where FPR95 measures the probability that a sample from an unknown class is incorrectly classified as one of the known classes when the true positive rate reaches 95%. For AUPR, we present results for two cases: when known samples are treated as positive samples AUPR (known) and when unknown samples are treated as positive samples AUPR (unknown). Additionally, we report the F1 scores for the known samples. While the primary focus of OSR is not to optimize performance on known classes, an effective OSR technique should maintain high classification rates for the known samples, ensuring that performance on known samples is not compromised.

Regarding the hyperparameters, for training TS2VEC and SoftCLT, we use the hyperparameters specified in their original papers. For Moment, we utilize the large model among the three published variants (small, base, and large) for our experiments. While the original papers employ an SVM for downstream classification evaluation, we replace it with an MLP network for consistency across all approaches. The MLP network consists of layers with 128, 64, and 32 units, batch normalization, and ReLU activation, followed by an output layer matching the number of classes. It is trained on each dataset using a batch size of 64, and up to 2000 epochs with early stopping.

For ALSET, the estimator head is integrated alongside the MLP. The MLP retains the same architecture and hyperparameters as described earlier. The estimator head is a fixed MLP consisting of two layers with 64 and 128 units, followed by an output layer matching the latent dimension. The output layer uses ReLU and Softplus activation functions. Matching the output size to the latent dimension is essential, as the estimator learns a standard deviation for each dimension. Each output value represents the standard deviation for its corresponding dimension in the latent space. The Softplus activation ensures that the predicted standard deviations remain non-negative. Other hyperparameters for ALSET include fixed values of $\lambda = 0.001$ and $\beta = 1e-4$ across all experiments, while the learning rate and $\alpha$ were optimized. For hyperparameter optimization, we utilize Optuna Akiba et al. (2019) to maximize the F1 score on known samples while simultaneously improving the AUROC for distinguishing between known and unknown samples. For ALSET on HAR dataset, $\alpha = 0.001$. The optimal hyperparameters for each method are as follows: the percentiles selected for ASH-s (MSP), ASH-s (Energy), ASH-p (MSP), ASH-p (Energy), Scale (Energy), and Scale (MSP) are 51, 45, 41, 65, 3, and 1, respectively. For ReAct (MSP) and ReAct (Energy), the optimal percentiles are 68 and 23, respectively.

For VOS, the hyperparameter $\epsilon$ is fixed at $1e-4$. For NPOS, the variance is set to 0.1, and $k$ for $k$NN is set to 5% of the number of samples. All hyperparameters in our experiments are empirically set to maximize performance.

## A.3 VISUALIZING THE CONFIDENCE LANDSCAPE

We illustrate how ALSET impacts the confidence of a classifier model during training in Figure A1. In this experiment, we use the two-class toy dataset Moon, where samples from different classes are represented by points of different colors. The sigmoid output indicates model confidence; values

near 0 or 1 indicate high confidence, and 0.5 indicates randomness. In the figure, colors encode the sigmoid output that forms a spectrum between 0 and 1. We observe that in the absence of our method, the model tends to be overconfident in regions where known data are not present (see Figure **??**). In contrast, ALSET ensures the model maintains high confidence in regions with known data while progressively reducing confidence in areas without known data. Figures A1b and A1c demonstrate this learning process, showcasing how ALSET refines the classifier's confidence after 500 epochs and beyond.

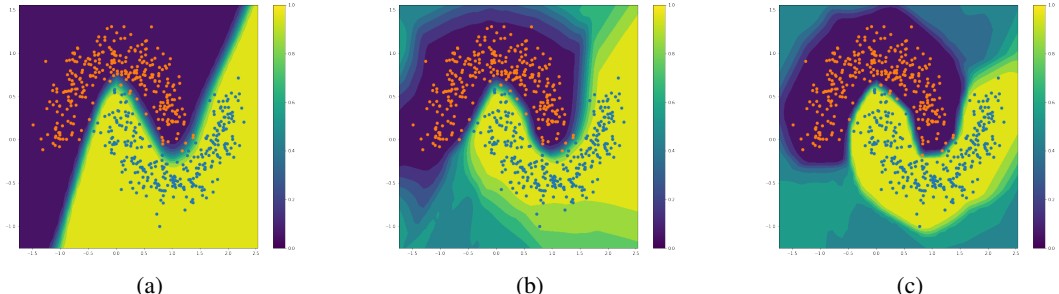

| (a) | (b) | (c) |

Figure A1: Comparison of confidence levels. (a) Baseline classifier without ALSET. (b) ALSET after 500 epochs. (c) ALSET after 1000 epochs. Purple and yellow denote confident regions for the two known classes; green highlights low-confidence areas corresponding to potential unknown samples.

### A.4 ANALYSIS OF THE HYPERPARAMETERS

First, we assess how uncertainty regularization ($\mathcal{L}_{\text{STD}}$) with coefficient ($\alpha$) and outlier exposure ($\mathcal{L}_{\text{CE}}^{\text{unknown}}$) with coefficient ($\lambda$) impact the performance the classifier on known samples, comparing two values for the coefficient of the $\mathcal{L}_{\text{L2E}}$ ($\beta = 0$ and $\beta = 0.0001$) as shown in Figure A2. The results indicate that smaller values of $\lambda$ and $\alpha$ maintain high accuracy, while excessively large values (for example, $\lambda = 1.0$ or $\alpha = 1.0$) can degrade performance, which is the result of destabilized training. This is expected, as high $\alpha$ values lead to large learned standard deviations, causing sampled unknowns to overlap significantly with known samples, thereby disrupting the decision boundaries. Similarly, high $\lambda$ values can overly suppress the model's confidence.

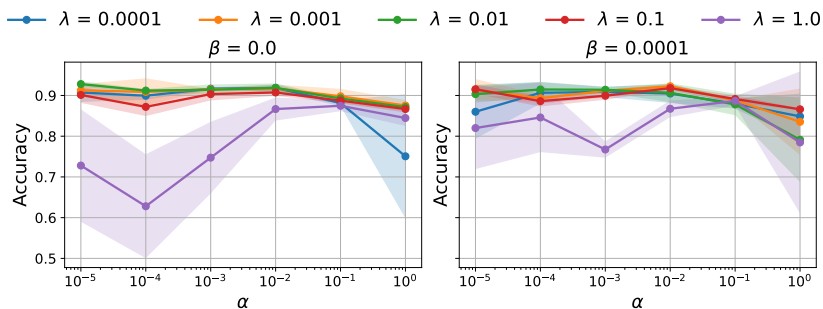

Figure A2: The impact of hyperparameters on the closed-set accuracy. Shaded areas represent ±1 standard deviation across 5 runs.

We further examine the role of the hyperparameter ($\beta$), as shown in Figure A3. The results indicate that introducing a small penalty ($\beta = 0.0001$) has minimal impact when the outlier exposure is low ($\lambda = 0.0001$). However, it significantly enhances training stability and accuracy when outlier exposure is high ($\lambda = 1.0$). This effect is also observed when the uncertainty regularization ($\alpha$) is large, even in the low outlier exposure setting ($\lambda = 0.0001$), where training can become unstable without the penalty. Therefore, in the case of instability, the penalty term helps preserve overall model performance.

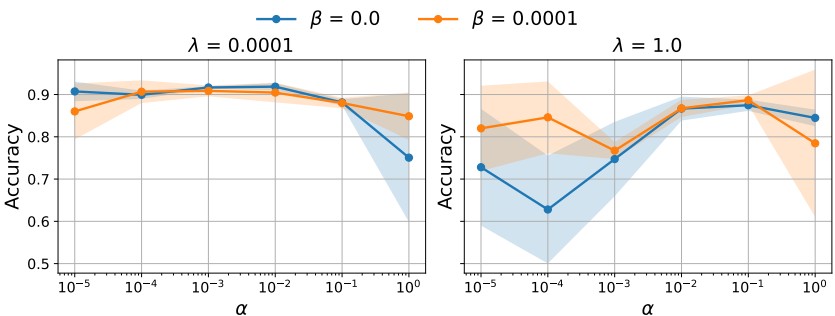

Figure A3: Effect of the hyperparameter ($\beta$) on closed-set accuracy under low ($\lambda = 0.0001$) and high ($\lambda = 1.0$) outlier exposure conditions. The penalty term ($\beta = 0.0001$) significantly improves stability and accuracy when either the uncertainty regularization ($\alpha$) or outlier exposure ($\lambda$) is high. Shaded regions represent ±1 standard deviation across 5 runs.

Figure A4 provides a view of how the parameter ($\alpha$) and the penalty term ($\beta$) jointly impact both accuracy and AUROC. The left panel ($\beta = 0$) clearly shows that, as $\alpha$ increases, the AUROC sharply declines, indicating that the model struggles to maintain its ability to distinguish known from unknown samples. This decline becomes particularly severe for large $\alpha$ values, where the learned uncertainty becomes excessively large, causing significant overlap between known and unknown regions in the feature space. In contrast, the accuracy remains relatively stable, only slightly decreasing as $\alpha$ approaches 1.0.

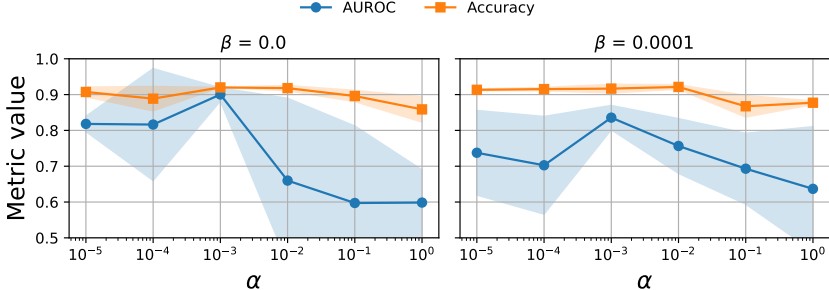

Figure A4: Impact of the hyperparameter ($\beta$) on AUROC and accuracy across different ($\alpha$) values. The left panel shows the performance with ($\beta = 0$), while the right panel illustrates the effect of a small penalty ($\beta = 0.0001$). Shaded regions represent ±1 standard deviation across 5 runs.

When a small penalty is introduced ($\beta = 0.0001$, right panel), this instability is noticeably reduced. The AUROC remains higher and more consistent across a wide range of $\alpha$ values. This stabilization occurs because the larger ($\beta$) constrains the scale of the generated unknown samples, preventing them from growing excessively large. From these results, we observe that using $\alpha = 10^{-3}$ provides the most stable overall performance, balancing high accuracy and robust OSR performance. This setting is likely related to the trade-off between the hyperparameters.

## A.5 PERFORMANCE OF ALSET WITH DIFFERENT UNKNOWN CLASSES

In all experiments, we structured the pipeline such that the last class in each dataset was treated as the unknown class. While this approach still involves a form of random selection, as the class order is arbitrary, we further provide results with different unknown classes in Table A1, using the HAR dataset, which includes six distinct human activity classes. Regardless of the choice of unknown class, we observe that introducing ALSET consistently improves OSR performance by a significant margin.

Table A1: Results on HAR dataset with different classes as the unknown class.

| Unknown class | F1↑ (known) | FPR95↓ Energy | MSP | AUROC↑ Energy | MSP | AUPR (known)↑ Energy | MSP | AUPR (unknown)↑ Energy | MSP |
|---|---|---|---|---|---|---|---|---|---|
| Walking (w/ ALSET) | 90.80 (0.79) | 50.00 (28.13) | 43.20 (19.77) | 85.00 (9.29) | 82.20 (5.05) | 84.60 (9.40) | 84.20 (5.05) | 80.80 (12.26) | 73.60 (8.13) |
| Walking (w/o ALSET) | 91.50 (0.84) | 69.67 (15.46) | 78.00 (23.86) | 59.00 (15.27) | 46.83 (20.23) | 66.67 (14.83) | 56.50 (19.18) | 57.83 (11.48) | 48.83 (13.85) |
| Stairs-Up (w/ ALSET) | 91.00 (0.94) | 58.60 (22.09) | 52.80 (14.20) | 67.00 (15.16) | 68.00 (10.79) | 73.20 (15.09) | 76.60 (7.79) | 65.40 (10.97) | 61.20 (8.55) |
| Stairs-Up (w/o ALSET) | 91.40 (0.55) | 61.60 (23.60) | 70.60 (20.35) | 62.20 (13.01) | 55.80 (8.01) | 68.80 (13.88) | 64.20 (11.88) | 56.40 (7.57) | 50.40 (4.22) |
| Stairs-Down (w/ ALSET) | 90.20 (0.79) | 25.20 (14.29) | 32.00 (8.89) | 88.00 (7.51) | 83.20 (6.44) | 91.20 (5.31) | 88.60 (3.92) | 82.60 (11.27) | 74.80 (9.10) |
| Stairs-Down (w/o ALSET) | 92.00 (0.00) | 93.00 (10.05) | 89.20 (13.14) | 38.60 (13.43) | 38.00 (14.12) | 46.00 (12.63) | 46.40 (12.58) | 45.60 (6.54) | 45.00 (7.18) |
| Sitting (w/ ALSET) | 90.80 (1.23) | 53.00 (26.97) | 37.80 (23.86) | 78.00 (12.63) | 77.20 (16.01) | 80.80 (10.14) | 82.20 (13.51) | 74.80 (12.41) | 73.00 (13.50) |
| Sitting (w/o ALSET) | 91.60 (0.55) | 73.80 (22.34) | 68.60 (18.04) | 52.20 (21.72) | 54.20 (14.13) | 58.80 (19.49) | 62.20 (14.18) | 52.80 (14.08) | 51.60 (9.04) |
| Standing (w/ ALSET) | 90.86 (0.36) | 14.40 (8.42) | 35.71 (7.84) | 92.60 (5.36) | 81.29 (7.21) | 94.00 (3.71) | 86.00 (4.44) | 88.20 (8.12) | 73.71 (8.71) |
| Standing (w/o ALSET) | 91.00 (0.71) | 77.00 (34.97) | 78.60 (30.16) | 45.40 (31.09) | 43.00 (25.07) | 54.40 (26.65) | 53.40 (23.42) | 53.40 (20.18) | 47.20 (12.89) |
| Lying (w/ ALSET) | 90.60 (0.55) | 38.40 (2.41) | 38.60 (8.65) | 87.00 (9.35) | 83.80 (6.34) | 88.40 (4.93) | 86.40 (1.82) | 83.40 (15.32) | 77.80 (11.54) |
| Lying (w/o ALSET) | 90.95 (1.47) | 75.52 (19.99) | 78.00 (19.40) | 51.52 (21.69) | 46.05 (18.75) | 59.86 (17.63) | 55.95 (16.35) | 54.05 (13.72) | 48.10 (9.82) |

## A.6 RATIO OF UNKNOWN CLASSES

To assess the sensitivity of ALSET to the ratio between known and unknown classes, we conduct the following study on the HAR dataset using the SoftCLT encoder. We fix *Lying* and *Standing* as the known classes and then incrementally introduce one additional unknown class at a time, in the order *Walking → Stairs-Up → Stairs-Down → Sitting*. After each addition, we evaluate the model, report OSR metrics, and compare the results against the baselines without ALSET. We present the results in the tables below. Compared to the baseline (Table A2, Table A3), ALSET consistently achieves lower FPR@95 and higher AUROC, AUPR, and Macro-F1 across all open-set difficulty levels. For example, with 4 unknown classes, ALSET improves Macro-F1 with a large margin. These gains confirm that ALSET provides more reliable OSR and better overall performance under increasing unknown class scenarios.

Table A2: Performance with ALSET under varying numbers of unknown classes. ALSET consistently lowers FPR@95 and improves AUROC, AUPR, and Macro-F1 as the number of unknown classes increases.

| # Unknown | FPR@95 (±std) | AUROC (±std) | AUPR-Unknown (±std) | Macro-F1 (±std) |
|---|---|---|---|---|
| 1 | 0.680 (±0.293) | 0.680 (±0.144) | 0.530 (±0.119) | 0.995 (±0.006) |
| 2 | 0.725 (±0.300) | 0.598 (±0.119) | 0.590 (±0.093) | 0.980 (±0.014) |
| 3 | 0.723 (±0.265) | 0.563 (±0.107) | 0.658 (±0.083) | 0.955 (±0.031) |
| 4 | 0.755 (±0.228) | 0.500 (±0.095) | 0.683 (±0.063) | 0.923 (±0.055) |

Table A3: Performance without ALSET (MSP baseline) under varying numbers of unknown classes. MSP fails to reject unknowns, resulting in consistently high FPR@95 and poor Macro-F1.

| # Unknown | FPR@95 (±std) | AUROC (±std) | AUPR-Unknown (±std) | Macro-F1 (±std) |
|---|---|---|---|---|
| 1 | 1.000 (±0.000) | 0.523 (±0.015) | 0.353 (±0.015) | 0.615 (±0.024) |
| 2 | 1.000 (±0.000) | 0.523 (±0.017) | 0.490 (±0.018) | 0.560 (±0.024) |
| 3 | 1.000 (±0.000) | 0.543 (±0.038) | 0.615 (±0.034) | 0.553 (±0.048) |
| 4 | 1.000 (±0.000) | 0.535 (±0.034) | 0.673 (±0.026) | 0.518 (±0.049) |

## A.7 CHALLENGES WITH INDEPENDENT VARIANCE ESTIMATION

Our initial choice of using diagonal covariance, $\Sigma = \mathrm{diag}(\sigma_1^2, \ldots, \sigma_d^2)$, was motivated by its computational efficiency and ease of optimization, requiring only $\mathcal{O}(d)$ parameters per sample. This design enabled stable training while allowing adaptive variance growth through the classifier-driven feedback loop.

To further improve flexibility, we extend ALSET to support *full-covariance* estimation. Specifically, we model the latent distribution of synthesized outliers as

$$z_i^{\mathrm{unknown}} = z_i^{\mathrm{known}} + L_i \epsilon, \qquad \epsilon \sim \mathcal{N}(0, I), \tag{10}$$

where $L_i$ is a learnable lower-triangular matrix such that

$$\Sigma_i = L_i L_i^\top \tag{11}$$

is symmetric and positive semi-definite. Instead of learning only per-dimension standard deviations, this formulation captures cross-dimensional correlations and encourages a richer modeling of uncertainty. To prevent uncontrolled expansion of covariance, we add a log-determinant regularization term:

$$\mathcal{L}_{\text{cov}} = \lambda \log \det(\Sigma_i), \tag{12}$$

which penalizes excessively large covariance volumes while preserving flexibility.

We evaluate this full-covariance variant, denoted as ALSET (Cov), on the HAR dataset using three different encoders. Results are reported in Table A4. Performance varies with encoder choice, which is expected given the importance of feature correlations for open-set detection. Using TS2Vec yields results comparable to ALSET, while SoftCLT shows slightly less stable behavior, reflected by larger standard deviations.

Table A4: Performance of ALSET with full covariance (ALSET-Cov) on the HAR dataset across three encoders. Full covariance improves AUROC and AUPR for most configurations, demonstrating the benefit of modeling feature correlations.

| Method | Encoder | F1↑ | FPR@95↓ | | AUROC↑ | | AUPR-Known↑ | | AUPR-Unknown↑ | |
|---|---|---|---|---|---|---|---|---|---|---|
| | | | Energy | MSP | Energy | MSP | Energy | MSP | Energy | MSP |
| ALSET (Cov) | Moment | 75.65 (1.57) | 92.00 (4.66) | 90.73 (4.40) | 51.58 (6.15) | 56.58 (4.09) | 43.04 (8.11) | 46.12 (9.53) | 59.27 (7.37) | 63.88 (5.02) |
| ALSET (Cov) | TS2VEC | 90.00 (1.00) | 45.20 (20.78) | 42.20 (13.41) | 83.60 (9.07) | 80.20 (4.97) | 87.20 (7.05) | 86.20 (3.27) | 77.80 (13.66) | 71.20 (6.83) |
| ALSET (Cov) | SoftCLT | 89.50 (0.97) | 39.58 (10.94) | 35.83 (7.76) | 77.96 (14.13) | 79.00 (4.60) | 80.25 (8.69) | 82.50 (6.01) | 69.50 (19.71) | 72.10 (7.54) |

## A.8 ADDITIONAL OSR METRICS

Our choice to focus on MSP and Energy as the primary scoring functions is directly motivated by the design and objective of ALSET. As an outlier exposure (OE)-based method, ALSET explicitly aims to regularize the classifier's confidence on unknown inputs. Confidence-based scoring functions such as MSP and Energy are thus the most appropriate, as they directly measure the classifier's confidence and reflect the intended effect of ALSET's boundary-localized regularization.

To further strengthen our comparison, we include two additional baselines: Mahalanobis distance and ODIN. We evaluate these methods on the HAR dataset using three encoders, and report results in Table A5. Except when using Moment as the encoder in combination with the Energy metric, ALSET consistently outperforms both baselines across all evaluation metrics.

Table A5: Comparison of ODIN and Mahalanobis distance baselines under the Energy scoring function on the HAR dataset. ALSET consistently achieves better performance across FPR@95, AUROC, and AUPR metrics for both known and unknown classes.

| Method | Encoder | F1 (Known)↑ | FPR@95↓ | AUROC↑ | AUPR (Known)↑ | AUPR (Unknown)↑ |
|---|---|---|---|---|---|---|
| ODIN | Moment | 78.36 (1.96) | 69.50 (9.91) | 63.40 (3.34) | 70.30 (5.36) | 56.40 (2.37) |
| ODIN | TS2VEC | 89.32 (0.53) | 55.40 (10.12) | 64.00 (6.52) | 74.80 (5.22) | 57.00 (5.87) |
| ODIN | SoftCLT | 88.74 (0.26) | 76.60 (20.48) | 48.20 (12.68) | 58.40 (12.60) | 48.40 (7.23) |
| Mahalanobis distance | Moment | 79.01 (1.25) | 90.10 (6.97) | 52.70 (6.24) | 50.90 (3.51) | 54.40 (7.31) |
| Mahalanobis distance | TS2VEC | 89.20 (0.84) | 55.00 (9.90) | 63.80 (9.47) | 56.60 (8.08) | 74.80 (6.53) |
| Mahalanobis distance | SoftCLT | 88.80 (0.45) | 64.60 (5.41) | 56.80 (10.38) | 52.60 (7.70) | 68.80 (6.76) |

## A.9 KNOWN VS. UNKNOWN CLASSIFICATION RESULTS.

In the main text of the paper, we presented scatterplots with fitted regression lines to illustrate the relationship between classifier performance and OSR metrics, building on prior works such as Vaze et al. (2022) and Cen et al. (2023). Here, we focus exclusively on the regression lines, as shown in Figure A5, to better demonstrate the impact of our proposed ALSET method. By isolating these fitted models, this visualization highlights the improved correlation between F1 scores on samples from known classes and OSR performance.

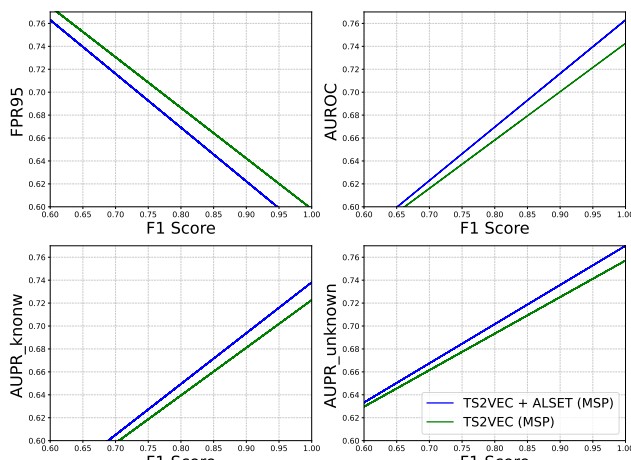

Figure A5: The performance of the model on samples from known vs. unknown classes is presented on the UCR datasets. As shown, our proposed method accelerates this improvement compared to the baseline.

### A.10   PER-DATASET RESULTS

In Tables A6 through A12, we present all the results for individual datasets of the UCR repository for our method as well as the baselines. Specifically, Table A6 presents the results of the baseline TS2VEC on all UCR datasets, whereas Tables A7, A8, A9, A10, and A11 present the results for prior works ASH-p, ASH-s, Scale, ReAct, and VOS, on OSR using TS2VEC as the time-series model. Finally, Table A12 presents the results of our method.

### A.11   DISCUSSION

**Design and efficiency.**   ALSET is an *encoder-agnostic*, latent-space OE method. It jointly trains the classifier and an uncertainty estimator in a closed loop, so the classifier guides where and how uncertainty expands near the decision boundary. This removes post-hoc boundary mining (e.g., kNN in NPOS), learns *per-sample* Gaussians matched to heterogeneous boundaries, and yields calibrated predictions without extra post-processing. The pipeline is lightweight compared to input-space generators.

If the latent geometry is clean, approximately isotropic class clusters with sharp boundaries, VOS/NPOS can be strong baselines and ALSET may add little. In more common time-series latents that are anisotropic or non-Gaussian, ALSET's sample-level adaptive uncertainty and classifier-guided exposure are advantageous.

**Generative alternatives and vision methods.**   Diffusion/flow models aim to learn a full unknown distribution, which is ill-defined in open-set settings and adds complexity. Vision OE benefits from CLIP-conditioned diffusion (e.g., DREAM-OOD Du et al. (2024)); comparable semantic priors are not available for time series. Hence we target boundary-localized unknowns in latent space which is simple, principled, and effective for sequential data.

**Distribution shift.**   ALSET is trained to assign low confidence in low-density regions, aiding detection of emerging classes under shift. If a sample remains in a known class, a strong encoder should keep it close to the known manifold, enabling generalization. Because ALSET does not require joint training with the encoder, it pairs naturally with shift-aware or foundation encoders.

In summary, ALSET introduces: (i) a classifier-driven feedback loop coupling classification and uncertainty; (ii) directional, adaptive variance growth; (iii) boundary-localized synthesis; and (iv) an uncertainty mechanism that reshapes confidence topology. We choose latent-space synthesis to (a) leverage pretrained time-series encoders without modifying backbones and (b) avoid the difficulties of input-space generation (long-range dependencies, irregular sampling, variable lengths, and lack

of CLIP/diffusion priors). To counter MSP's overconfidence on unknowns, ALSET uses a uniform softmax target for synthetic unknowns and an adaptive uncertainty regularizer, improving open-set discrimination without harming closed-set accuracy.

### A.12    LIMITATION

We conducted extensive experiments to evaluate our method across diverse time-series datasets, demonstrating ALSET's superiority. However, no benchmark currently exists in the time-series domain for OSR to evaluate model performance on near-unknown and far-unknown classes. Therefore, developing such a benchmark for future OSR techniques in the time-series domain remains an important direction.

### A.13    USE OF LARGE LANGUAGE MODELS

We used a large language model strictly for **grammar checking and stylistic polishing** of the manuscript text. The LLM was *not* used to generate technical content, design algorithms, run experiments, analyze data, or produce results. All conceptual contributions, methods, experiments, and conclusions are authored by the paper's authors.

Table A6: Results of the TS2VEC encoder with no OSR mechanism on the UCR datasets.

| Dataset | F1↑ (known) | FPR95↓ Energy | FPR95↓ MSP | AUROC↑ Energy | AUROC↑ MSP | AUPR (known)↑ Energy | AUPR (known)↑ MSP | AUPR (unknown)↑ Energy | AUPR (unknown)↑ MSP |
|---|---|---|---|---|---|---|---|---|---|
| ACSF1 | 81.30 (2.63) | 62.00 (7.44) | 52.70 (7.63) | 63.80 (5.01) | 72.20 (3.88) | 53.10 (3.93) | 61.10 (6.14) | 74.40 (3.78) | 80.30 (2.83) |
| Adiac | 58.44 (2.07) | 62.00 (14.92) | 56.78 (11.76) | 70.22 (11.11) | 76.78 (6.20) | 61.67 (10.62) | 68.22 (7.29) | 77.11 (8.01) | 81.33 (4.85) |
| AllGestureWiimoteX | 78.80 (0.84) | 80.40 (5.27) | 77.80 (4.87) | 68.60 (3.29) | 70.60 (1.52) | 63.60 (5.03) | 67.20 (1.92) | 68.20 (3.03) | 70.20 (2.59) |
| AllGestureWiimoteY | 74.00 (0.71) | 94.60 (4.10) | 95.40 (3.21) | 61.60 (6.27) | 61.40 (4.39) | 60.40 (5.94) | 59.80 (4.55) | 58.00 (7.42) | 58.00 (5.79) |
| AllGestureWiimoteZ | 72.40 (1.52) | 81.20 (5.07) | 82.00 (3.74) | 63.40 (4.28) | 63.20 (3.77) | 59.60 (4.83) | 60.20 (3.35) | 65.80 (4.21) | 65.20 (3.49) |
| ArrowHead | 84.20 (0.84) | 89.20 (10.23) | 95.40 (4.16) | 50.00 (13.17) | 45.00 (9.82) | 51.80 (9.65) | 48.60 (5.94) | 51.60 (9.58) | 48.00 (8.40) |
| BME | 97.40 (1.14) | 81.80 (21.08) | 80.80 (15.27) | 61.00 (14.51) | 64.60 (15.85) | 60.20 (7.46) | 64.40 (13.15) | 61.20 (15.47) | 63.60 (16.07) |
| Beef | 78.40 (3.13) | 77.60 (20.11) | 72.60 (14.67) | 55.80 (16.77) | 52.20 (10.28) | 52.20 (12.19) | 46.00 (4.64) | 62.60 (14.19) | 64.00 (8.03) |
| CBF | 100.00 (0.00) | 63.20 (22.06) | 77.60 (33.84) | 73.00 (19.07) | 62.20 (22.84) | 72.20 (17.77) | 67.20 (17.17) | 75.80 (16.32) | 62.00 (22.88) |
| Car | 95.40 (2.97) | 64.20 (22.54) | 59.40 (19.40) | 77.00 (9.00) | 76.80 (3.11) | 72.00 (11.73) | 72.80 (4.55) | 79.00 (11.73) | 80.40 (5.68) |
| ChlorineConcentration | 72.80 (18.39) | 94.40 (2.41) | 94.40 (2.30) | 54.80 (2.68) | 55.60 (3.13) | 54.00 (3.08) | 55.20 (1.92) | 54.60 (2.19) | 54.60 (1.82) |
| CinCECGTorso | 70.20 (3.11) | 90.00 (14.95) | 91.00 (9.67) | 51.40 (10.11) | 52.60 (6.11) | 50.80 (5.89) | 55.00 (2.92) | 53.00 (12.86) | 52.60 (9.29) |
| CricketX | 80.00 (0.71) | 49.40 (6.91) | 49.80 (7.22) | 77.20 (4.82) | 80.00 (3.16) | 67.80 (5.12) | 73.80 (3.56) | 82.20 (3.63) | 83.40 (3.36) |
| CricketY | 77.80 (2.17) | 85.20 (5.89) | 84.00 (3.81) | 59.20 (4.02) | 58.60 (3.44) | 57.80 (3.77) | 56.00 (2.35) | 61.60 (4.16) | 61.20 (3.27) |
| CricketZ | 79.20 (1.64) | 65.00 (10.25) | 67.60 (11.78) | 73.80 (4.09) | 73.80 (2.39) | 66.60 (5.68) | 67.20 (2.95) | 78.00 (3.08) | 77.40 (4.39) |
| Crop | 71.60 (0.55) | 94.00 (3.67) | 96.80 (1.30) | 48.60 (6.35) | 38.80 (5.07) | 49.60 (4.93) | 40.80 (2.17) | 49.80 (5.45) | 44.80 (3.11) |
| DiatomSizeReduction | 89.00 (5.52) | 36.00 (38.91) | 68.20 (29.44) | 76.00 (37.17) | 58.20 (33.36) | 75.60 (24.83) | 58.60 (19.63) | 82.20 (27.85) | 67.20 (23.37) |
| DistalPhalanxOutlineAgeGroup | 83.20 (2.17) | 94.80 (2.68) | 93.40 (2.97) | 34.80 (9.78) | 36.60 (2.41) | 39.60 (3.85) | 40.20 (1.30) | 44.80 (4.76) | 46.80 (2.05) |
| DistalPhalanxTW | 45.89 (3.14) | 94.22 (5.87) | 93.00 (3.74) | 12.22 (7.16) | 15.00 (7.05) | 33.00 (1.73) | 33.56 (1.67) | 37.67 (5.74) | 39.56 (4.36) |
| DodgerLoopDay | 41.40 (11.37) | 89.40 (4.72) | 88.40 (3.21) | 48.20 (6.14) | 42.60 (7.96) | 47.20 (5.07) | 43.00 (4.85) | 55.00 (3.39) | 53.40 (4.45) |
| ECG5000 | 70.40 (0.55) | 86.00 (8.15) | 74.80 (12.52) | 77.60 (4.62) | 79.60 (2.51) | 76.40 (2.97) | 76.20 (1.92) | 73.20 (4.76) | 76.80 (6.76) |
| EOGHorizontalSignal | 59.40 (1.52) | 98.40 (1.34) | 98.60 (1.14) | 21.20 (1.48) | 20.20 (3.63) | 34.80 (0.45) | 34.20 (1.10) | 35.80 (0.84) | 35.40 (1.52) |
| EOGVerticalSignal | 44.60 (2.70) | 83.40 (15.37) | 77.80 (16.84) | 49.60 (18.09) | 61.80 (16.81) | 47.20 (10.01) | 58.80 (16.15) | 58.00 (17.09) | 66.00 (14.58) |
| ElectricDevices | 72.00 (0.71) | 89.80 (5.36) | 86.20 (8.14) | 46.60 (3.78) | 53.20 (2.49) | 45.80 (2.39) | 49.40 (1.52) | 52.40 (5.13) | 58.40 (6.07) |
| EthanolLevel | 28.80 (11.45) | 94.40 (2.07) | 93.00 (2.55) | 47.20 (3.42) | 51.60 (2.97) | 47.60 (2.41) | 51.20 (1.79) | 49.20 (3.27) | 53.00 (3.54) |
| FaceAll | 89.80 (2.59) | 22.80 (9.23) | 25.60 (9.96) | 91.20 (3.03) | 89.60 (3.21) | 86.40 (4.22) | 84.20 (2.59) | 93.40 (2.51) | 92.60 (2.70) |
| FaceFour | 85.33 (7.81) | 69.22 (22.71) | 64.22 (22.08) | 67.56 (13.17) | 76.11 (8.33) | 59.89 (9.79) | 70.00 (9.86) | 72.22 (13.77) | 78.44 (9.33) |
| FacesUCR | 91.60 (1.14) | 32.00 (6.78) | 34.00 (4.80) | 88.80 (1.92) | 89.20 (1.10) | 84.60 (1.79) | 85.80 (1.79) | 90.80 (1.48) | 91.20 (1.10) |
| Fish | 93.20 (1.48) | 65.60 (18.61) | 63.40 (18.34) | 73.60 (9.13) | 73.80 (8.93) | 66.20 (8.67) | 65.40 (7.37) | 77.60 (8.76) | 78.20 (9.65) |
| Fungi | 91.60 (2.88) | 43.60 (7.09) | 44.00 (14.66) | 76.00 (9.19) | 73.20 (14.20) | 65.60 (12.64) | 62.00 (13.10) | 84.20 (5.07) | 82.40 (9.66) |
| GestureMidAirD1 | 61.80 (2.86) | 47.00 (24.27) | 49.60 (15.32) | 80.00 (10.49) | 72.20 (9.83) | 70.40 (10.24) | 60.00 (8.31) | 83.60 (10.11) | 80.00 (8.19) |
| GestureMidAirD2 | 51.40 (3.91) | 54.60 (22.77) | 63.20 (24.93) | 72.20 (9.55) | 68.00 (9.17) | 60.80 (6.22) | 57.40 (5.73) | 76.80 (12.30) | 72.40 (14.12) |
| GestureMidAirD3 | 30.60 (2.41) | 64.80 (26.05) | 72.40 (13.90) | 64.80 (14.47) | 60.60 (7.37) | 59.20 (11.03) | 51.40 (4.04) | 72.80 (14.70) | 67.20 (9.04) |
| GesturePebbleZ1 | 92.20 (2.04) | 92.20 (12.97) | 96.00 (5.25) | 45.30 (10.26) | 45.90 (5.17) | 47.40 (5.64) | 47.90 (3.51) | 48.40 (11.51) | 46.10 (5.30) |
| GesturePebbleZ2 | 86.60 (2.12) | 97.00 (4.81) | 98.10 (3.03) | 43.20 (7.87) | 42.40 (7.09) | 47.30 (5.38) | 47.60 (4.40) | 45.30 (8.10) | 43.90 (6.72) |
| Haptics | 51.30 (3.80) | 85.30 (6.48) | 79.20 (4.73) | 58.90 (9.90) | 64.60 (4.27) | 56.60 (7.81) | 59.90 (4.33) | 62.00 (7.67) | 67.40 (4.53) |
| InlineSkate | 38.20 (1.48) | 93.40 (4.22) | 93.20 (4.08) | 53.90 (6.92) | 53.70 (7.60) | 52.40 (5.87) | 52.60 (5.02) | 54.60 (5.62) | 54.30 (5.96) |
| InsectEPGRegularTrain | 100.00 (0.00) | 22.00 (34.15) | 18.80 (33.61) | 80.60 (34.12) | 85.20 (32.79) | 82.30 (28.39) | 87.50 (25.63) | 87.40 (22.86) | 90.10 (22.17) |
| InsectEPGSmallTrain | 100.00 (0.00) | 63.50 (48.11) | 64.50 (41.70) | 38.00 (49.17) | 41.30 (46.48) | 44.50 (47.23) | 45.90 (47.40) | 57.30 (34.16) | 60.00 (32.05) |
| InsectWingbeatSound | 59.90 (0.99) | 91.00 (5.98) | 89.40 (4.93) | 50.30 (11.10) | 51.30 (6.99) | 50.50 (9.01) | 48.80 (4.76) | 53.30 (8.38) | 55.30 (6.53) |
| LargeKitchenAppliances | 89.40 (1.90) | 89.20 (8.50) | 87.50 (8.77) | 60.80 (6.27) | 63.30 (4.37) | 58.40 (5.85) | 60.10 (3.70) | 60.50 (8.14) | 62.60 (6.57) |
| Lightning7 | 79.40 (1.95) | 88.20 (9.73) | 91.60 (5.90) | 56.20 (7.95) | 54.00 (5.39) | 57.20 (11.17) | 51.60 (3.65) | 57.20 (4.97) | 56.20 (7.60) |
| Mallat | 88.00 (2.00) | 44.40 (24.30) | 38.20 (12.52) | 86.40 (8.65) | 86.80 (4.09) | 85.40 (8.38) | 81.20 (5.36) | 86.80 (9.88) | 89.40 (4.16) |
| Meat | 98.00 (2.74) | 45.20 (29.30) | 37.00 (18.23) | 79.00 (17.85) | 80.80 (15.79) | 76.20 (18.05) | 75.80 (19.42) | 82.60 (14.62) | 86.40 (10.31) |
| MedicalImages | 83.40 (1.34) | 90.80 (3.49) | 90.80 (3.11) | 60.40 (3.97) | 60.60 (4.51) | 61.80 (2.59) | 59.80 (3.63) | 58.00 (4.47) | 58.20 (3.77) |
| MelbournePedestrian | 90.00 (0.55) | 40.20 (14.89) | 44.00 (13.42) | 88.60 (3.51) | 84.60 (3.29) | 83.80 (4.51) | 79.00 (3.24) | 87.80 (5.36) | 84.80 (4.27) |
| MiddlePhalanxOutlineAgeGroup | 57.80 (3.56) | 95.40 (4.67) | 90.60 (6.07) | 43.60 (9.66) | 45.00 (6.20) | 46.40 (6.19) | 46.00 (4.00) | 47.00 (7.07) | 51.40 (5.13) |
| MiddlePhalanxTW | 29.40 (7.27) | 93.60 (13.76) | 99.60 (0.89) | 15.20 (27.91) | 3.80 (2.77) | 36.40 (10.48) | 31.20 (0.45) | 39.60 (18.68) | 31.20 (0.45) |
| MixedShapesRegularTrain | 90.60 (0.55) | 37.00 (14.71) | 36.60 (11.19) | 86.00 (5.83) | 85.20 (5.17) | 80.60 (7.96) | 80.60 (5.68) | 88.80 (4.97) | 88.80 (4.15) |
| MixedShapesSmallTrain | 85.20 (1.10) | 36.00 (9.54) | 34.20 (12.56) | 87.80 (3.56) | 87.00 (4.18) | 82.00 (4.85) | 81.60 (4.88) | 90.20 (3.27) | 89.60 (3.97) |
| NonInvasiveFetalECGThorax1 | 91.40 (0.55) | 32.60 (16.85) | 40.40 (18.55) | 85.00 (10.24) | 79.40 (11.76) | 79.20 (7.45) | 71.00 (15.05) | 88.60 (7.30) | 85.20 (8.29) |
| NonInvasiveFetalECGThorax2 | 93.60 (0.55) | 5.40 (6.23) | 13.40 (7.09) | 98.40 (2.07) | 96.60 (2.30) | 96.20 (3.70) | 95.20 (3.11) | 98.80 (1.64) | 97.20 (1.48) |
| OSULeaf | 82.80 (0.84) | 71.60 (13.28) | 69.80 (13.14) | 74.40 (2.41) | 75.20 (3.63) | 68.80 (2.28) | 72.80 (2.95) | 76.20 (2.95) | 77.00 (4.64) |
| OliveOil | 88.00 (8.80) | 78.80 (15.34) | 75.60 (20.07) | 59.00 (16.09) | 63.80 (12.07) | 55.00 (12.63) | 59.40 (12.30) | 64.40 (15.52) | 67.80 (13.54) |
| PLAID | 54.60 (1.82) | 92.20 (4.76) | 92.60 (7.16) | 56.40 (4.34) | 58.20 (6.38) | 52.60 (5.50) | 52.60 (5.68) | 59.60 (3.21) | 60.60 (7.30) |
| PigAirwayPressure | 40.20 (5.07) | 77.60 (4.67) | 73.80 (14.55) | 54.60 (9.42) | 57.20 (7.95) | 47.80 (5.93) | 49.40 (5.86) | 63.20 (4.92) | 65.00 (7.78) |
| PigArtPressure | 82.20 (1.79) | 40.40 (8.68) | 32.00 (20.59) | 78.00 (6.71) | 87.40 (4.88) | 62.60 (6.07) | 77.20 (10.55) | 84.80 (4.27) | 89.80 (5.89) |
| PigCVP | 61.20 (5.89) | 51.20 (24.20) | 57.00 (14.51) | 69.00 (14.16) | 70.80 (5.36) | 56.40 (10.21) | 58.40 (5.27) | 77.00 (12.08) | 77.00 (5.00) |
| Plane | 100.00 (0.45) | 1.00 (1.00) | 1.20 (0.84) | 99.40 (0.39) | 99.60 (0.55) | 97.80 (3.83) | 99.40 (0.55) | 99.80 (0.45) | 99.80 (0.45) |
| ProximalPhalanxOutlineAgeGroup | 68.80 (3.90) | 92.60 (9.48) | 95.40 (5.90) | 23.40 (18.01) | 27.20 (13.10) | 36.20 (6.02) | 37.00 (4.18) | 42.00 (13.47) | 41.80 (9.07) |
| ProximalPhalanxTW | 50.60 (1.95) | 71.40 (8.02) | 65.80 (8.04) | 47.60 (7.77) | 49.80 (7.12) | 44.00 (3.54) | 45.20 (3.56) | 63.20 (5.40) | 66.20 (5.67) |
| RefrigerationDevices | 78.40 (2.30) | 90.00 (5.34) | 89.40 (6.11) | 57.00 (5.39) | 54.80 (3.77) | 55.80 (3.91) | 54.40 (3.91) | 57.60 (6.77) | 57.20 (4.66) |
| Rock | 76.00 (7.09) | 75.20 (17.48) | 74.20 (17.85) | 71.00 (6.60) | 73.00 (6.96) | 67.47 (8.87) | 67.87 (10.88) | 73.07 (7.36) | 74.87 (6.91) |
| ScreenType | 42.17 (9.96) | 93.62 (3.49) | 91.33 (6.01) | 48.54 (8.65) | 49.75 (11.58) | 49.15 (6.34) | 50.25 (7.89) | 50.92 (6.02) | 52.25 (8.58) |
| SemgHandMovementCh2 | 86.60 (0.97) | 81.80 (14.37) | 82.00 (9.79) | 68.70 (6.63) | 68.70 (4.67) | 66.70 (6.62) | 64.90 (3.84) | 67.70 (9.07) | 68.50 (5.95) |
| SemgHandSubjectCh2 | 96.40 (0.52) | 95.00 (8.27) | 93.70 (11.78) | 67.40 (8.29) | 70.40 (6.64) | 69.40 (6.36) | 71.90 (4.46) | 60.10 (9.68) | 62.30 (9.21) |
| ShakeGestureWiimoteZ | 86.60 (3.41) | 23.10 (28.59) | 23.40 (25.33) | 90.50 (10.84) | 91.10 (7.53) | 84.90 (15.70) | 86.50 (10.49) | 91.80 (12.27) | 92.60 (8.10) |
| ShapesAll | 82.67 (0.82) | 31.00 (2.83) | 26.17 (6.59) | 83.83 (2.48) | 86.67 (2.66) | 71.50 (3.78) | 75.00 (3.74) | 89.67 (1.21) | 91.17 (1.94) |
| SmallKitchenAppliances | 85.00 (1.58) | 86.40 (13.76) | 87.40 (9.84) | 59.20 (21.15) | 57.40 (10.97) | 56.40 (16.41) | 56.20 (8.53) | 60.80 (16.78) | 59.20 (10.50) |
| SmoothSubspace | 98.60 (1.52) | 75.80 (13.70) | 80.60 (10.43) | 74.20 (4.15) | 75.00 (1.87) | 74.40 (4.72) | 73.80 (3.42) | 72.80 (7.05) | 72.80 (3.77) |
| StarLightCurves | 100.00 (0.00) | 77.40 (25.38) | 84.00 (29.08) | 62.60 (22.84) | 46.60 (25.47) | 61.00 (19.74) | 50.40 (22.17) | 65.80 (19.12) | 54.20 (21.70) |
| SwedishLeaf | 93.00 (0.71) | 34.30 (13.13) | 74.40 (16.65) | 75.80 (6.26) | 76.20 (4.15) | 59.80 (7.85) | 72.60 (4.93) | 76.80 (6.76) | 76.60 (5.50) |
| Symbols | 92.60 (1.95) | 32.20 (26.94) | 22.60 (8.08) | 81.80 (18.39) | 88.20 (6.06) | 75.40 (20.57) | 80.00 (10.79) | 87.00 (13.91) | 92.20 (3.63) |
| SyntheticControl | 100.00 (0.00) | 28.60 (10.19) | 24.20 (7.29) | 89.40 (4.83) | 92.00 (2.55) | 84.00 (5.39) | 86.60 (2.97) | 92.20 (3.77) | 94.20 (2.17) |
| Trace | 100.00 (0.00) | 80.80 (24.15) | 83.20 (17.50) | 33.80 (33.49) | 40.80 (28.67) | 44.40 (21.52) | 48.60 (20.98) | 50.00 (26.01) | 54.00 (21.49) |
| TwoPatterns | 100.00 (0.00) | 62.00 (32.79) | 76.80 (27.17) | 91.00 (5.10) | 90.40 (3.36) | 93.80 (3.42) | 93.80 (1.92) | 84.40 (9.74) | 81.60 (6.73) |
| UMD | 99.00 (0.71) | 89.40 (20.97) | 91.20 (13.41) | 68.60 (19.50) | 70.00 (15.67) | 72.40 (17.84) | 76.40 (9.15) | 63.00 (17.59) | 63.40 (16.06) |
| UWaveGestureLibraryAll | 91.10 (0.32) | 49.50 (9.68) | 51.80 (9.14) | 85.50 (2.95) | 83.40 (2.80) | 82.10 (2.85) | 80.10 (2.42) | 86.50 (3.63) | 85.30 (3.43) |
| UWaveGestureLibraryX | 76.20 (0.42) | 52.90 (7.14) | 54.10 (5.61) | 82.30 (4.22) | 80.70 (3.27) | 77.80 (5.41) | 76.80 (4.02) | 84.10 (3.81) | 82.90 (3.00) |
| UWaveGestureLibraryY | 70.60 (0.52) | 93.70 (4.67) | 95.40 (1.90) | 61.30 (4.19) | 60.40 (1.84) | 59.40 (4.12) | 59.40 (1.71) | 57.80 (4.94) | 56.60 (2.17) |
| UWaveGestureLibraryZ | 76.90 (0.32) | 93.90 (2.56) | 95.30 (2.31) | 51.30 (4.35) | 48.50 (2.55) | 51.30 (3.86) | 49.10 (1.73) | 52.40 (4.35) | 49.50 (2.99) |
| WordSynonyms | 49.30 (3.09) | 52.10 (14.05) | 48.80 (14.60) | 74.50 (7.12) | 75.20 (7.58) | 64.20 (6.84) | 64.80 (7.73) | 81.20 (6.00) | 82.00 (6.11) |
| Worms | 64.30 (5.56) | 86.90 (8.24) | 81.60 (10.38) | 47.50 (18.08) | 51.80 (10.50) | 50.10 (13.99) | 48.50 (7.14) | 56.20 (12.12) | 59.70 (9.36) |
| **Average** | **76.94** | **69.21** | **69.42** | **64.41** | **64.55** | **62.61** | **62.60** | **68.42** | **68.58** |

Table A7: Results of ASH-P on the UCR datasets.

| Dataset | F1↑ (known) | FPR95↓ Energy | FPR95↓ MSP | AUROC↑ Energy | AUROC↑ MSP | AUPR (known)↑ Energy | AUPR (known)↑ MSP | AUPR (unknown)↑ Energy | AUPR (unknown)↑ MSP |
|---|---|---|---|---|---|---|---|---|---|
| ACSF1 | 23.00 (2.67) | 61.40 (19.61) | 58.40 (20.52) | 55.90 (19.75) | 60.00 (22.48) | 49.60 (10.22) | 54.50 (14.38) | 70.00 (14.79) | 72.20 (15.74) |
| Adiac | 0.89 (1.36) | 82.56 (20.68) | 82.67 (22.84) | 44.11 (24.02) | 55.33 (23.48) | 45.11 (10.88) | 49.22 (21.52) | 57.44 (18.22) | 69.56 (14.22) |
| AllGestureWiimoteX | 25.00 (2.65) | 80.60 (10.31) | 83.60 (5.59) | 60.40 (8.96) | 57.20 (6.61) | 56.60 (7.02) | 54.20 (5.22) | 64.60 (9.21) | 61.40 (6.88) |
| AllGestureWiimoteY | 23.20 (7.66) | 89.00 (6.40) | 88.00 (4.69) | 52.00 (6.78) | 50.20 (5.26) | 53.60 (5.27) | 49.40 (3.05) | 55.00 (7.00) | 54.60 (5.59) |
| AllGestureWiimoteZ | 22.80 (3.42) | 84.60 (8.79) | 86.60 (8.05) | 52.80 (12.74) | 53.20 (7.79) | 51.00 (11.94) | 52.20 (6.38) | 59.00 (10.20) | 57.40 (6.80) |
| ArrowHead | 45.00 (12.10) | 91.20 (7.85) | 95.60 (6.66) | 38.40 (13.22) | 33.20 (6.53) | 42.20 (7.40) | 39.00 (2.74) | 47.60 (10.90) | 45.80 (4.02) |
| BME | 74.80 (12.36) | 95.20 (3.49) | 88.00 (10.91) | 42.20 (10.50) | 54.60 (9.45) | 47.60 (8.20) | 55.60 (7.37) | 47.00 (8.09) | 55.60 (10.90) |
| Beef | 23.40 (15.90) | 89.20 (14.24) | 90.00 (16.49) | 38.20 (20.50) | 41.60 (19.32) | 42.00 (8.86) | 38.40 (12.54) | 48.60 (16.32) | 57.20 (15.17) |
| CBF | 96.60 (3.65) | 84.20 (20.60) | 86.80 (20.78) | 48.40 (21.94) | 46.40 (22.50) | 50.80 (13.16) | 52.00 (13.36) | 54.80 (17.73) | 51.60 (18.28) |
| Car | 26.00 (11.38) | 89.80 (9.58) | 89.20 (9.98) | 45.40 (10.24) | 43.40 (11.24) | 45.00 (5.24) | 43.60 (6.62) | 52.60 (10.99) | 53.00 (10.27) |
| ChlorineConcentration | 41.20 (7.16) | 93.00 (2.55) | 98.60 (3.13) | 50.60 (1.67) | 50.60 (1.67) | 50.60 (1.67) | 50.60 (1.14) | 51.80 (2.77) | 58.20 (6.72) |
| CinCECGTorso | 48.80 (9.52) | 86.40 (5.98) | 89.60 (3.36) | 55.60 (8.71) | 57.20 (3.42) | 52.20 (6.76) | 54.20 (4.09) | 59.20 (7.16) | 58.20 (1.92) |
| CricketX | 30.20 (11.39) | 94.80 (5.85) | 95.80 (5.12) | 41.80 (8.41) | 46.00 (11.31) | 44.80 (5.36) | 47.40 (9.76) | 47.40 (7.09) | 49.60 (7.99) |
| CricketY | 25.60 (8.29) | 85.60 (9.37) | 85.20 (6.38) | 58.20 (9.73) | 56.80 (5.36) | 54.80 (6.76) | 54.40 (4.22) | 60.40 (6.73) | 59.80 (4.02) |
| CricketZ | 22.40 (6.80) | 95.80 (3.96) | 92.40 (6.43) | 45.80 (6.46) | 45.80 (7.69) | 47.20 (5.26) | 47.80 (5.76) | 49.20 (6.06) | 49.80 (6.30) |
| Crop | 18.60 (2.88) | 84.00 (8.03) | 95.20 (0.84) | 64.60 (10.06) | 54.60 (3.85) | 62.80 (9.60) | 54.20 (4.15) | 64.60 (9.07) | 53.20 (1.79) |
| DiatomSizeReduction | 20.40 (4.98) | 73.80 (15.59) | 89.80 (12.83) | 56.00 (29.56) | 46.60 (23.53) | 56.20 (20.09) | 49.80 (19.89) | 65.80 (21.00) | 57.00 (15.76) |
| DistalPhalanxOutlineAgeGroup | 50.40 (8.96) | 90.80 (6.42) | 96.20 (6.50) | 50.80 (10.26) | 51.40 (11.52) | 50.60 (7.92) | 51.80 (9.09) | 53.80 (9.42) | 57.00 (12.35) |
| DistalPhalanxTW | 17.67 (7.18) | 66.89 (35.60) | 69.33 (21.76) | 54.44 (29.18) | 59.00 (18.14) | 55.44 (21.63) | 56.56 (15.94) | 63.56 (24.05) | 66.56 (16.33) |
| DodgerLoopDay | 20.80 (7.33) | 79.40 (16.80) | 78.40 (14.98) | 62.40 (15.21) | 56.80 (19.74) | 62.20 (14.55) | 52.60 (12.66) | 66.80 (12.15) | 65.40 (14.94) |
| ECG5000 | 36.80 (6.83) | 79.40 (9.81) | 83.00 (12.14) | 52.60 (11.78) | 49.60 (12.05) | 50.00 (9.70) | 48.60 (9.32) | 59.00 (9.62) | 56.20 (9.60) |
| EOGHorizontalSignal | 17.20 (7.26) | 91.80 (6.18) | 95.60 (3.85) | 43.60 (12.66) | 36.80 (12.46) | 44.00 (7.04) | 42.00 (7.07) | 50.80 (8.93) | 44.80 (6.76) |
| EOGVerticalSignal | 16.80 (5.17) | 85.00 (15.60) | 89.00 (10.70) | 43.40 (17.46) | 45.60 (19.28) | 45.00 (11.77) | 47.20 (14.34) | 53.80 (14.53) | 53.40 (16.06) |
| ElectricDevices | 58.00 (2.74) | 79.80 (7.73) | 77.40 (6.02) | 63.40 (7.44) | 62.20 (7.69) | 59.80 (5.26) | 58.20 (7.95) | 65.80 (7.85) | 67.00 (6.60) |
| EthanolLevel | 21.00 (5.83) | 93.20 (3.90) | 97.20 (2.59) | 53.40 (4.83) | 50.20 (6.53) | 51.60 (3.71) | 50.40 (5.64) | 53.60 (4.04) | 50.60 (4.51) |
| FaceAll | 53.20 (7.56) | 74.20 (10.66) | 75.60 (10.00) | 69.20 (4.97) | 66.40 (4.62) | 65.20 (5.67) | 61.20 (5.67) | 72.40 (6.69) | 70.20 (6.14) |
| FaceFour | 36.78 (18.19) | 89.56 (9.51) | 88.22 (7.61) | 45.67 (8.66) | 43.33 (14.70) | 45.11 (4.94) | 46.67 (11.97) | 52.56 (8.56) | 52.44 (10.26) |
| FacesUCR | 48.20 (5.97) | 85.00 (7.97) | 87.40 (7.96) | 57.40 (9.45) | 55.80 (7.40) | 53.80 (8.17) | 54.80 (7.60) | 60.40 (8.62) | 58.40 (6.15) |
| Fish | 20.40 (10.09) | 92.80 (7.66) | 94.40 (5.22) | 30.80 (12.52) | 29.20 (9.31) | 39.00 (5.74) | 37.80 (3.49) | 42.80 (9.26) | 42.60 (5.46) |
| Fungi | 7.60 (6.07) | 91.20 (10.52) | 86.80 (20.39) | 25.00 (23.61) | 35.60 (21.42) | 37.60 (9.42) | 40.80 (8.53) | 44.40 (16.27) | 49.20 (17.47) |
| GestureMidAirD1 | 9.60 (2.19) | 72.40 (19.24) | 77.80 (22.54) | 66.80 (4.92) | 62.60 (11.52) | 58.20 (15.78) | 54.40 (8.11) | 70.20 (6.76) | 65.60 (11.93) |
| GestureMidAirD2 | 12.20 (5.07) | 96.60 (3.97) | 97.20 (3.03) | 29.60 (17.74) | 32.20 (17.51) | 41.40 (11.74) | 39.60 (9.40) | 41.40 (8.11) | 42.20 (10.57) |
| GestureMidAirD3 | 6.00 (1.00) | 87.00 (12.51) | 84.60 (11.35) | 40.80 (9.44) | 42.40 (11.17) | 41.40 (3.78) | 42.20 (5.40) | 51.60 (9.81) | 53.80 (10.76) |
| GesturePebbleZ1 | 62.30 (6.22) | 69.60 (16.41) | 71.40 (17.63) | 73.50 (10.11) | 72.00 (11.85) | 70.00 (11.15) | 68.10 (12.09) | 75.60 (9.49) | 74.50 (10.70) |
| GesturePebbleZ2 | 65.10 (9.21) | 59.40 (24.55) | 60.70 (20.73) | 73.60 (10.88) | 72.00 (9.01) | 68.20 (11.49) | 66.90 (9.60) | 76.20 (11.92) | 76.00 (10.02) |
| Haptics | 25.10 (4.50) | 93.60 (4.12) | 93.40 (4.58) | 45.10 (8.33) | 45.30 (5.33) | 44.60 (5.16) | 47.10 (3.28) | 49.00 (6.72) | 48.50 (4.90) |
| InlineSkate | 13.40 (5.36) | 92.00 (5.14) | 90.10 (5.40) | 52.60 (6.22) | 54.40 (5.78) | 50.50 (4.55) | 53.20 (5.22) | 54.80 (5.77) | 56.20 (5.55) |
| InsectEPGRegularTrain | 100.00 (0.00) | 15.60 (29.90) | 16.20 (25.62) | 86.40 (29.07) | 87.30 (21.92) | 87.50 (25.09) | 84.70 (23.13) | 91.20 (19.49) | 91.80 (14.67) |
| InsectEPGSmallTrain | 91.40 (20.66) | 64.50 (36.57) | 58.40 (33.07) | 40.50 (35.08) | 46.60 (33.28) | 47.40 (21.42) | 50.00 (21.00) | 59.00 (26.29) | 63.90 (24.22) |
| InsectWingbeatSound | 24.40 (5.17) | 91.20 (5.96) | 91.90 (7.26) | 46.20 (10.86) | 46.90 (10.24) | 46.60 (7.46) | 47.60 (7.28) | 51.50 (8.33) | 51.10 (9.00) |
| LargeKitchenAppliances | 76.00 (10.39) | 88.40 (10.64) | 88.90 (10.00) | 55.60 (8.91) | 55.20 (9.96) | 54.50 (8.20) | 52.70 (6.98) | 56.70 (8.88) | 57.20 (9.46) |
| Lightning7 | 34.60 (15.87) | 96.20 (4.49) | 93.40 (5.59) | 47.80 (13.86) | 39.00 (15.41) | 50.80 (10.43) | 44.60 (11.26) | 49.20 (9.71) | 47.20 (11.19) |
| Mallat | 21.40 (4.93) | 93.00 (2.12) | 90.60 (5.94) | 45.20 (12.19) | 47.40 (10.76) | 47.40 (9.69) | 47.60 (8.08) | 49.40 (7.64) | 52.40 (8.76) |
| Meat | 33.00 (0.00) | 87.80 (12.17) | 100.00 (0.00) | 44.60 (11.04) | 47.80 (6.87) | 45.20 (9.63) | 54.80 (18.99) | 53.80 (11.39) | 70.20 (6.38) |
| MedicalImages | 23.80 (4.15) | 92.00 (2.24) | 93.80 (4.49) | 57.40 (3.97) | 54.00 (4.06) | 55.00 (3.32) | 53.00 (3.00) | 56.60 (3.71) | 53.60 (4.72) |
| MelbournePedestrian | 20.60 (11.01) | 92.20 (3.70) | 93.40 (9.74) | 43.40 (9.86) | 41.20 (14.18) | 46.80 (6.30) | 44.80 (8.32) | 48.40 (6.88) | 49.80 (10.64) |
| MiddlePhalanxOutlineAgeGroup | 45.20 (12.76) | 95.80 (4.09) | 91.20 (9.12) | 46.40 (8.50) | 49.80 (8.47) | 48.40 (5.68) | 49.60 (6.43) | 47.80 (6.38) | 52.00 (6.86) |
| MiddlePhalanxTW | 7.40 (3.65) | 85.00 (20.57) | 72.00 (27.70) | 33.40 (38.18) | 48.20 (42.19) | 46.60 (20.94) | 55.00 (21.95) | 50.20 (25.86) | 62.40 (28.32) |
| MixedShapesRegularTrain | 56.20 (16.69) | 74.60 (16.83) | 75.40 (14.83) | 67.40 (10.81) | 65.40 (9.79) | 63.60 (8.82) | 61.20 (8.29) | 69.80 (12.56) | 68.40 (11.63) |
| MixedShapesSmallTrain | 53.20 (11.69) | 83.80 (11.17) | 84.60 (10.83) | 53.40 (9.13) | 48.60 (9.99) | 50.20 (6.50) | 47.60 (5.59) | 58.60 (10.31) | 55.80 (9.83) |
| NonInvasiveFetalECGThorax1 | 0.80 (0.84) | 58.40 (35.03) | 80.00 (23.62) | 65.00 (32.00) | 58.00 (21.70) | 64.20 (24.63) | 58.40 (17.47) | 71.80 (26.19) | 67.20 (16.24) |
| NonInvasiveFetalECGThorax2 | 1.20 (0.45) | 81.60 (19.41) | 85.00 (12.85) | 52.40 (25.20) | 58.20 (9.26) | 53.40 (19.60) | 53.80 (6.38) | 59.00 (21.40) | 62.60 (9.45) |
| OSULeaf | 31.00 (12.10) | 91.60 (4.04) | 93.40 (6.88) | 48.60 (4.98) | 47.60 (4.16) | 46.60 (3.36) | 46.60 (1.82) | 53.40 (5.55) | 52.60 (3.36) |
| OliveOil | 21.20 (12.13) | 86.40 (13.37) | 100.00 (0.00) | 56.00 (11.38) | 54.40 (9.84) | 50.00 (0.96) | 72.00 (6.71) | 62.80 (6.53) | 74.60 (0.89) |
| PLAID | 10.20 (3.03) | 96.00 (5.15) | 92.00 (9.90) | 43.40 (9.71) | 47.60 (9.61) | 44.40 (3.97) | 47.00 (5.15) | 48.60 (7.70) | 54.00 (8.37) |
| PigAirwayPressure | 2.20 (0.84) | 85.20 (13.99) | 94.20 (4.55) | 47.80 (19.15) | 32.80 (11.69) | 46.00 (10.68) | 38.60 (4.56) | 55.60 (15.04) | 44.20 (5.40) |
| PigArtPressure | 8.00 (3.08) | 80.00 (16.26) | 84.00 (15.64) | 44.20 (20.72) | 42.20 (20.86) | 43.80 (9.65) | 42.80 (9.20) | 56.20 (16.99) | 54.00 (17.03) |
| PigCVP | 6.60 (2.07) | 69.00 (13.29) | 69.80 (20.55) | 60.20 (10.78) | 64.20 (6.87) | 52.00 (8.86) | 54.00 (7.55) | 68.80 (7.26) | 69.40 (8.44) |
| Plane | 69.20 (8.35) | 46.00 (16.00) | 40.60 (8.38) | 70.60 (15.47) | 79.20 (6.83) | 61.20 (15.99) | 68.20 (5.12) | 80.40 (10.62) | 85.40 (4.04) |
| ProximalPhalanxOutlineAgeGroup | 45.40 (18.43) | 61.40 (30.09) | 81.20 (26.88) | 71.00 (18.15) | 68.40 (19.86) | 68.40 (18.66) | 70.80 (16.10) | 72.40 (16.30) | 75.40 (15.44) |
| ProximalPhalanxTW | 15.00 (8.86) | 49.20 (22.96) | 79.60 (28.01) | 81.80 (13.55) | 67.60 (16.01) | 77.00 (16.00) | 66.40 (14.86) | 84.20 (11.90) | 78.00 (6.36) |
| RefrigerationDevices | 48.00 (10.32) | 82.40 (4.77) | 81.20 (4.76) | 61.60 (3.91) | 61.80 (3.56) | 59.80 (5.26) | 59.20 (4.02) | 64.60 (3.39) | 64.80 (2.77) |
| Rock | 57.13 (9.51) | 89.53 (8.91) | 89.80 (8.47) | 51.80 (15.83) | 52.60 (14.64) | 53.07 (12.31) | 52.87 (11.50) | 57.00 (11.72) | 57.13 (10.67) |
| ScreenType | 38.83 (5.94) | 93.58 (3.26) | 92.00 (5.26) | 47.67 (8.62) | 47.67 (8.72) | 47.92 (5.78) | 48.42 (6.29) | 50.08 (6.46) | 50.92 (7.56) |
| SemgHandMovementCh2 | 31.60 (7.32) | 91.80 (8.36) | 91.40 (9.85) | 52.80 (7.55) | 51.80 (7.86) | 50.90 (4.79) | 51.00 (5.75) | 54.20 (7.22) | 54.60 (7.78) |
| SemgHandSubjectCh2 | 17.20 (4.42) | 89.80 (6.46) | 94.50 (7.84) | 57.40 (5.08) | 58.60 (4.81) | 58.00 (4.19) | 58.90 (4.48) | 56.50 (5.19) | 62.40 (5.34) |
| ShakeGestureWiimoteZ | 35.80 (6.48) | 78.30 (17.59) | 80.10 (20.70) | 56.40 (20.49) | 44.90 (20.29) | 52.50 (12.23) | 46.00 (13.00) | 62.40 (16.28) | 55.80 (16.21) |
| ShapesAll | 6.67 (2.42) | 88.67 (10.05) | 91.50 (7.15) | 44.67 (19.20) | 52.50 (14.63) | 46.50 (12.03) | 50.67 (9.24) | 52.83 (14.30) | 55.67 (11.79) |
| SmallKitchenAppliances | 73.60 (7.92) | 76.60 (23.46) | 71.20 (18.50) | 67.80 (17.56) | 76.00 (4.06) | 64.40 (14.19) | 72.00 (2.55) | 68.60 (17.47) | 76.60 (6.27) |
| SmoothSubspace | 91.20 (3.19) | 66.00 (11.22) | 66.20 (11.56) | 72.40 (6.95) | 73.40 (3.05) | 68.20 (8.07) | 67.60 (3.21) | 74.20 (9.01) | 75.60 (6.66) |
| StarLightCurves | 98.20 (2.95) | 99.40 (0.55) | 99.80 (0.45) | 14.40 (1.82) | 14.80 (1.64) | 33.20 (0.45) | 33.00 (0.71) | 33.20 (0.84) | 35.20 (2.77) |
| SwedishLeaf | 12.20 (5.02) | 83.40 (6.19) | 84.00 (16.51) | 58.20 (12.46) | 57.40 (12.78) | 54.80 (10.06) | 62.60 (10.16) | 62.80 (10.18) | 62.00 (12.55) |
| Symbols | 30.80 (4.97) | 54.80 (22.48) | 67.00 (15.92) | 65.20 (18.21) | 67.00 (14.16) | 56.80 (15.45) | 62.20 (13.85) | 75.20 (13.54) | 72.60 (10.85) |
| SyntheticControl | 98.40 (0.55) | 33.80 (19.64) | 35.00 (22.59) | 89.00 (7.65) | 89.60 (8.68) | 83.40 (11.80) | 85.80 (11.39) | 91.60 (5.50) | 91.80 (6.76) |
| Trace | 87.40 (17.69) | 83.00 (18.20) | 82.80 (21.56) | 38.20 (19.04) | 37.60 (19.29) | 44.80 (7.65) | 41.00 (7.65) | 52.20 (15.47) | 51.80 (17.57) |
| TwoPatterns | 96.80 (2.39) | 51.60 (5.59) | 52.60 (3.05) | 79.60 (2.97) | 80.00 (3.08) | 71.60 (5.59) | 74.60 (5.37) | 83.40 (2.07) | 83.60 (1.14) |
| UMD | 78.40 (10.19) | 95.00 (6.63) | 95.20 (5.02) | 52.20 (9.26) | 56.20 (10.99) | 50.60 (6.66) | 57.20 (10.38) | 52.00 (7.81) | 58.20 (5.07) |
| UWaveGestureLibraryAll | 40.10 (7.58) | 80.20 (4.34) | 82.60 (3.66) | 61.50 (4.45) | 59.10 (3.96) | 56.30 (5.46) | 55.30 (3.16) | 65.70 (3.83) | 63.20 (3.68) |
| UWaveGestureLibraryX | 29.70 (5.17) | 91.20 (4.39) | 90.40 (4.35) | 50.10 (6.30) | 53.10 (4.28) | 48.90 (4.82) | 50.70 (2.83) | 52.90 (5.61) | 55.20 (5.07) |
| UWaveGestureLibraryY | 18.80 (7.04) | 95.20 (2.53) | 92.70 (4.22) | 48.30 (5.83) | 49.70 (7.21) | 48.50 (4.50) | 49.50 (5.13) | 49.30 (4.83) | 51.50 (6.50) |
| UWaveGestureLibraryZ | 32.30 (8.11) | 85.20 (8.93) | 87.70 (4.11) | 56.60 (10.50) | 54.50 (3.31) | 54.30 (9.71) | 51.80 (2.10) | 59.90 (9.57) | 57.70 (4.00) |
| WordSynonyms | 11.70 (3.27) | 84.40 (7.97) | 87.40 (5.93) | 56.40 (8.91) | 52.50 (10.60) | 53.70 (7.85) | 50.40 (7.86) | 61.10 (7.74) | 58.20 (7.66) |
| Worms | 30.20 (7.44) | 81.90 (8.02) | 81.20 (13.64) | 57.00 (18.21) | 58.50 (14.75) | 55.10 (16.50) | 55.50 (11.56) | 64.20 (12.99) | 64.30 (12.17) |
| **Average** | **36.50** | **81.58** | **83.56** | **53.40** | **53.28** | **52.93** | **53.09** | **59.18** | **59.78** |

Table A8: Results of ASH-s on the UCR datasets.

| Dataset | F1↑ (known) | FPR95↓ Energy | FPR95↓ MSP | AUROC↑ Energy | AUROC↑ MSP | AUPR (known)↑ Energy | AUPR (known)↑ MSP | AUPR (unknown)↑ Energy | AUPR (unknown)↑ MSP |
|---|---|---|---|---|---|---|---|---|---|
| Adiac | 0.89 (1.36) | 72.67 (19.89) | 100.00 (0.00) | 55.44 (18.34) | 49.67 (5.81) | 51.44 (9.85) | 33.78 (13.60) | 65.33 (17.03) | 73.00 (1.87) |
| AllGestureWiimoteX | 25.00 (2.65) | 85.80 (6.61) | 100.00 (0.00) | 53.00 (9.62) | 52.20 (4.66) | 50.00 (7.97) | 51.20 (3.63) | 57.60 (6.91) | 62.40 (2.61) |
| AllGestureWiimoteY | 23.20 (7.66) | 96.80 (1.48) | 100.00 (0.00) | 43.60 (5.77) | 48.20 (5.72) | 46.20 (4.60) | 48.40 (4.04) | 46.60 (5.32) | 57.00 (1.87) |
| AllGestureWiimoteZ | 22.80 (3.42) | 83.00 (9.59) | 100.00 (0.00) | 55.60 (10.60) | 49.00 (2.35) | 51.60 (6.69) | 49.20 (2.05) | 61.60 (11.26) | 60.60 (4.88) |
| ArrowHead | 45.00 (12.10) | 93.60 (5.13) | 100.00 (0.00) | 37.40 (9.63) | 43.40 (2.88) | 42.20 (5.85) | 35.80 (6.80) | 45.80 (6.87) | 68.00 (2.55) |
| BME | 74.80 (12.36) | 91.40 (8.53) | 100.00 (0.00) | 46.60 (6.80) | 50.20 (8.50) | 49.40 (5.46) | 50.60 (6.84) | 50.00 (6.63) | 63.80 (6.69) |
| Beef | 23.40 (15.90) | 84.20 (10.80) | 100.00 (0.00) | 44.20 (18.86) | 45.80 (4.38) | 44.40 (7.64) | 39.80 (9.09) | 53.80 (12.91) | 65.80 (5.02) |
| CBF | 96.60 (3.65) | 56.40 (11.70) | 100.00 (0.00) | 69.00 (13.06) | 57.60 (14.19) | 62.20 (16.28) | 55.60 (23.13) | 77.00 (8.94) | 73.00 (8.09) |
| Car | 26.00 (11.38) | 91.00 (8.51) | 100.00 (0.00) | 51.20 (16.68) | 51.40 (6.35) | 53.80 (13.63) | 51.00 (24.74) | 55.40 (10.19) | 74.20 (2.59) |
| ChlorineConcentration | 41.20 (7.16) | 95.20 (2.68) | 98.00 (4.47) | 48.80 (3.56) | 49.00 (0.71) | 48.40 (2.30) | 43.60 (10.45) | 49.40 (4.10) | 68.80 (9.52) |
| CinCECGTorso | 48.80 (9.52) | 93.80 (4.27) | 100.00 (0.00) | 43.00 (6.89) | 47.20 (3.83) | 44.00 (3.87) | 49.20 (16.89) | 47.60 (5.18) | 64.00 (8.66) |
| CricketX | 30.20 (11.39) | 88.80 (5.07) | 100.00 (0.00) | 51.00 (7.97) | 49.60 (5.98) | 49.20 (4.97) | 50.20 (4.76) | 55.40 (6.58) | 60.60 (3.78) |
| CricketY | 25.60 (8.29) | 90.20 (2.77) | 100.00 (0.00) | 52.20 (6.26) | 52.20 (1.64) | 51.00 (4.36) | 52.40 (1.82) | 55.20 (5.85) | 62.40 (1.14) |
| CricketZ | 22.40 (6.80) | 96.00 (2.74) | 100.00 (0.00) | 48.80 (7.73) | 50.40 (7.33) | 51.00 (6.04) | 49.60 (5.13) | 50.20 (5.59) | 62.40 (4.22) |
| Crop | 18.60 (2.88) | 89.00 (8.94) | 100.00 (0.00) | 49.40 (18.66) | 41.40 (14.60) | 51.40 (11.67) | 45.60 (7.89) | 52.80 (14.31) | 49.60 (7.27) |
| DiatomSizeReduction | 20.40 (4.98) | 81.20 (26.94) | 100.00 (0.00) | 41.60 (28.40) | 49.40 (1.34) | 46.00 (18.37) | 55.00 (27.39) | 53.40 (23.65) | 74.60 (0.89) |
| DistalPhalanxOutlineAgeGroup | 50.40 (8.96) | 95.40 (2.88) | 100.00 (0.00) | 67.00 (7.11) | 50.00 (0.00) | 66.00 (9.30) | 75.00 (0.00) | 61.60 (5.68) | 75.00 (0.00) |
| DistalPhalanxTW | 17.67 (7.18) | 26.22 (28.19) | 100.00 (0.00) | 85.00 (24.89) | 51.22 (8.39) | 83.22 (19.89) | 60.00 (20.49) | 89.22 (18.46) | 73.78 (3.90) |
| DodgerLoopDay | 20.80 (7.33) | 77.80 (17.50) | 100.00 (0.00) | 72.20 (8.23) | 60.20 (11.23) | 67.40 (9.69) | 58.80 (8.67) | 73.20 (11.63) | 64.80 (9.93) |
| ECG5000 | 36.80 (6.83) | 83.60 (8.73) | 100.00 (0.00) | 60.60 (7.47) | 55.60 (10.64) | 54.60 (5.13) | 52.60 (9.91) | 64.00 (8.69) | 68.80 (3.63) |
| EOGHorizontalSignal | 17.20 (7.26) | 30.80 (13.65) | 50.00 (11.51) | 90.40 (4.51) | 80.80 (5.89) | 84.20 (7.36) | 77.60 (7.37) | 92.60 (3.21) | 84.20 (4.82) |
| EOGVerticalSignal | 16.80 (5.17) | 67.60 (10.81) | 82.20 (4.92) | 68.80 (9.28) | 53.80 (10.35) | 61.00 (10.44) | 50.60 (8.08) | 74.20 (7.92) | 61.20 (6.76) |
| ElectricDevices | 58.00 (2.74) | 99.40 (0.89) | 100.00 (0.00) | 29.40 (11.78) | 44.80 (8.32) | 38.40 (5.59) | 46.40 (6.35) | 38.40 (4.16) | 57.80 (4.92) |
| EthanolLevel | 21.00 (5.83) | 96.00 (2.00) | 96.60 (3.29) | 51.20 (5.17) | 49.80 (1.64) | 56.00 (3.13) | 56.20 (11.08) | 51.00 (3.16) | 60.60 (13.16) |
| FaceAll | 53.20 (7.56) | 82.40 (8.11) | 100.00 (0.00) | 56.60 (11.55) | 52.80 (3.90) | 52.80 (8.38) | 51.80 (3.70) | 62.60 (8.20) | 65.00 (1.41) |
| FaceFour | 36.78 (18.19) | 87.22 (10.39) | 100.00 (0.00) | 46.56 (16.33) | 50.56 (11.20) | 47.00 (9.66) | 47.78 (14.08) | 54.56 (12.94) | 68.44 (6.33) |
| FacesUCR | 48.20 (5.97) | 87.40 (6.39) | 100.00 (0.00) | 52.80 (8.11) | 51.40 (6.43) | 50.20 (5.81) | 50.40 (5.41) | 57.60 (8.14) | 60.80 (3.03) |
| Fish | 20.40 (10.09) | 88.40 (2.51) | 100.00 (0.00) | 44.20 (13.54) | 46.00 (5.70) | 44.20 (7.56) | 35.00 (22.36) | 55.20 (7.85) | 72.00 (4.53) |
| Fungi | 7.60 (6.07) | 66.60 (28.97) | 76.20 (26.80) | 48.40 (35.51) | 55.20 (26.66) | 50.00 (18.04) | 54.60 (19.01) | 64.00 (25.20) | 65.80 (18.55) |
| GestureMidAirD1 | 9.60 (2.19) | 91.80 (9.60) | 91.00 (12.33) | 24.60 (12.93) | 47.20 (16.57) | 35.80 (3.70) | 46.80 (7.85) | 42.60 (10.69) | 55.40 (10.31) |
| GestureMidAirD2 | 12.20 (5.07) | 84.40 (12.30) | 82.40 (18.62) | 48.00 (16.78) | 48.80 (16.81) | 49.60 (12.12) | 46.20 (8.61) | 56.20 (12.19) | 58.60 (11.93) |
| GestureMidAirD3 | 6.00 (1.00) | 82.40 (16.35) | 82.00 (20.31) | 50.60 (11.10) | 57.60 (11.89) | 47.40 (6.88) | 52.60 (7.16) | 58.00 (10.42) | 60.80 (12.44) |
| GesturePebbleZ1 | 62.30 (6.22) | 85.90 (16.93) | 95.00 (15.81) | 45.90 (20.90) | 47.20 (18.26) | 48.90 (13.14) | 47.80 (12.94) | 54.40 (16.08) | 61.80 (10.85) |
| GesturePebbleZ2 | 65.10 (9.21) | 98.80 (0.79) | 100.00 (0.00) | 24.10 (10.06) | 31.60 (8.03) | 37.10 (4.79) | 36.70 (7.51) | 36.60 (3.34) | 52.80 (3.77) |
| Haptics | 25.10 (4.38) | 90.70 (6.29) | 100.00 (0.00) | 53.60 (6.45) | 50.30 (7.26) | 52.40 (3.84) | 49.80 (5.37) | 55.80 (7.69) | 61.30 (5.52) |
| InlineSkate | 13.40 (5.36) | 95.00 (4.14) | 99.20 (2.53) | 48.10 (5.11) | 51.20 (6.00) | 48.30 (4.08) | 47.60 (9.29) | 50.20 (4.32) | 60.40 (7.23) |
| InsectEPGRegularTrain | 100.00 (0.00) | 21.50 (30.41) | 20.40 (27.91) | 80.00 (27.06) | 81.80 (23.67) | 78.10 (28.45) | 78.80 (27.39) | 87.80 (17.45) | 89.10 (14.56) |
| InsectEPGSmallTrain | 91.40 (20.66) | 50.90 (34.19) | 52.90 (35.51) | 51.20 (33.11) | 55.00 (30.15) | 53.10 (23.90) | 51.70 (24.68) | 69.00 (23.53) | 72.90 (20.00) |
| InsectWingbeatSound | 24.40 (5.17) | 67.10 (12.48) | 100.00 (0.00) | 72.40 (8.73) | 64.10 (6.84) | 67.00 (9.67) | 61.10 (6.89) | 75.40 (8.38) | 69.80 (6.65) |
| LargeKitchenAppliances | 76.00 (10.39) | 91.00 (5.75) | 100.00 (0.00) | 52.40 (8.28) | 48.60 (9.36) | 51.40 (6.60) | 47.10 (9.67) | 54.50 (6.10) | 65.00 (4.45) |
| Lightning7 | 34.60 (15.87) | 96.80 (4.55) | 100.00 (0.00) | 38.40 (17.52) | 37.40 (7.27) | 44.40 (13.24) | 41.60 (3.58) | 44.60 (10.83) | 52.20 (5.72) |
| Mallat | 21.40 (4.93) | 83.80 (16.69) | 100.00 (0.00) | 57.00 (11.98) | 54.20 (9.60) | 54.40 (10.31) | 53.20 (9.34) | 60.00 (12.55) | 67.00 (2.92) |
| Meat | 33.00 (0.00) | 84.40 (19.22) | 100.00 (0.00) | 46.20 (14.17) | 50.00 (0.00) | 47.20 (9.34) | 75.00 (0.00) | 54.80 (13.88) | 75.00 (0.00) |
| MedicalImages | 23.40 (4.15) | 92.80 (4.87) | 100.00 (0.00) | 56.80 (1.92) | 53.80 (2.77) | 55.40 (3.36) | 53.00 (1.87) | 56.00 (3.32) | 64.60 (4.28) |
| MelbournePedestrian | 20.60 (11.01) | 92.60 (6.66) | 100.00 (0.00) | 48.00 (8.98) | 45.80 (5.76) | 42.60 (2.88) | 46.00 (4.30) | 46.80 (8.23) | 61.00 (7.48) |
| MiddlePhalanxOutlineAgeGroup | 45.20 (12.76) | 83.00 (15.07) | 98.80 (2.68) | 54.80 (9.20) | 52.40 (3.51) | 53.20 (8.26) | 61.40 (8.50) | 58.20 (10.69) | 67.80 (11.34) |
| MiddlePhalanxTW | 7.40 (3.65) | 66.20 (38.72) | 91.60 (18.78) | 53.20 (43.33) | 40.20 (22.66) | 60.60 (30.66) | 41.40 (12.92) | 63.80 (31.41) | 58.60 (16.67) |
| MixedShapesRegularTrain | 56.20 (16.69) | 91.60 (5.18) | 100.00 (0.00) | 49.60 (11.28) | 51.60 (2.30) | 49.80 (9.20) | 49.80 (3.70) | 52.40 (9.40) | 69.00 (3.00) |
| MixedShapesSmallTrain | 53.20 (11.69) | 86.20 (8.35) | 100.00 (0.00) | 48.80 (14.02) | 47.00 (13.66) | 48.80 (9.28) | 46.80 (10.40) | 56.20 (10.78) | 62.60 (7.30) |
| NonInvasiveFetalECGThorax1 | 0.80 (0.84) | 77.20 (27.79) | 100.00 (0.00) | 61.60 (25.48) | 50.60 (4.77) | 61.80 (19.58) | 50.60 (5.13) | 64.00 (22.06) | 69.60 (2.30) |
| NonInvasiveFetalECGThorax2 | 1.20 (0.45) | 94.20 (6.42) | 100.00 (0.00) | 34.00 (17.90) | 48.00 (7.52) | 39.80 (6.61) | 45.60 (10.41) | 45.20 (10.99) | 69.80 (3.83) |
| OSULeaf | 31.00 (12.10) | 90.20 (5.40) | 100.00 (0.00) | 54.80 (6.83) | 53.20 (4.76) | 50.00 (4.53) | 52.20 (5.50) | 59.20 (5.40) | 67.00 (2.92) |
| OliveOil | 21.20 (12.13) | 74.20 (23.18) | 100.00 (0.00) | 55.20 (18.43) | 51.00 (4.12) | 52.80 (13.44) | 65.80 (22.87) | 63.40 (18.69) | 75.00 (1.41) |
| PLAID | 10.20 (3.03) | 98.00 (3.46) | 100.00 (0.00) | 27.60 (7.60) | 47.80 (7.92) | 37.80 (3.96) | 47.20 (15.21) | 39.00 (4.64) | 67.20 (10.28) |
| PigAirwayPressure | 2.20 (0.84) | 70.60 (20.51) | 96.80 (7.16) | 52.20 (11.03) | 37.80 (10.52) | 45.40 (5.03) | 39.40 (5.03) | 63.40 (12.70) | 54.40 (6.50) |
| PigArtPressure | 8.00 (3.08) | 78.40 (14.83) | 74.80 (34.86) | 42.60 (8.44) | 58.20 (19.80) | 41.40 (3.65) | 51.20 (10.92) | 56.60 (8.56) | 66.80 (18.86) |
| PigCVP | 6.60 (2.07) | 74.00 (18.53) | 82.80 (17.14) | 47.00 (19.79) | 44.00 (14.37) | 46.40 (5.93) | 42.80 (5.93) | 60.60 (16.27) | 55.20 (14.34) |
| Plane | 69.20 (8.35) | 64.60 (24.05) | 76.40 (32.59) | 47.40 (27.76) | 52.20 (35.01) | 47.20 (14.53) | 51.60 (26.37) | 64.60 (18.64) | 67.80 (22.03) |
| ProximalPhalanxOutlineAgeGroup | 45.40 (18.43) | 96.60 (2.41) | 100.00 (0.00) | 25.80 (17.30) | 48.80 (2.17) | 39.40 (9.96) | 55.00 (27.39) | 39.40 (7.37) | 74.00 (1.73) |
| ProximalPhalanxTW | 15.00 (8.86) | 75.00 (19.14) | 100.00 (0.00) | 44.60 (14.34) | 47.00 (4.64) | 55.20 (12.24) | 42.60 (21.22) | 66.80 (14.32) | 71.80 (3.49) |
| RefrigerationDevices | 48.00 (10.32) | 92.40 (2.88) | 100.00 (0.00) | 51.60 (8.79) | 47.00 (6.20) | 53.00 (9.92) | 47.80 (4.32) | 53.00 (5.15) | 55.80 (2.59) |
| Rock | 57.13 (9.51) | 90.73 (9.38) | 98.33 (6.45) | 42.87 (13.72) | 39.73 (11.64) | 46.00 (10.39) | 40.93 (8.91) | 51.20 (9.69) | 54.40 (7.88) |
| ScreenType | 38.83 (5.94) | 93.92 (3.60) | 93.42 (3.48) | 51.33 (6.60) | 49.33 (5.85) | 50.83 (4.34) | 49.50 (4.12) | 52.33 (6.50) | 51.50 (4.87) |
| SemgHandMovementCh2 | 31.60 (7.32) | 88.40 (10.20) | 100.00 (0.00) | 59.70 (7.70) | 56.10 (4.82) | 58.50 (6.80) | 54.80 (4.89) | 59.70 (9.36) | 66.80 (3.82) |
| SemgHandSubjectCh2 | 17.20 (4.42) | 98.20 (1.55) | 100.00 (0.00) | 46.30 (7.28) | 46.00 (3.71) | 42.70 (5.42) | 45.00 (4.42) | 41.20 (3.71) | 65.40 (4.09) |
| ShakeGestureWiimoteZ | 35.80 (6.48) | 89.20 (11.97) | 83.20 (15.53) | 42.80 (19.15) | 51.20 (13.55) | 45.40 (10.02) | 48.20 (14.47) | 51.70 (13.97) | 61.60 (9.89) |
| ShapesAll | 6.67 (2.42) | 84.67 (9.99) | 100.00 (0.00) | 54.17 (17.75) | 50.33 (9.89) | 52.33 (16.10) | 46.33 (12.24) | 61.00 (10.97) | 66.00 (4.15) |
| SmallKitchenAppliances | 73.60 (7.92) | 92.80 (1.30) | 100.00 (0.00) | 43.80 (8.64) | 41.40 (8.73) | 45.85 (5.89) | 47.20 (16.50) | 50.20 (4.09) | 63.40 (9.56) |
| SmoothSubspace | 91.20 (3.19) | 74.00 (12.83) | 100.00 (0.00) | 67.20 (13.65) | 64.20 (9.47) | 63.20 (11.90) | 62.80 (9.42) | 70.80 (11.08) | 73.20 (5.40) |
| StarLightCurves | 98.20 (2.95) | 86.00 (20.51) | 100.00 (0.00) | 50.20 (18.90) | 51.20 (1.30) | 55.20 (12.24) | 50.50 (5.80) | 53.60 (18.06) | 75.00 (0.00) |
| SwedishLeaf | 12.20 (5.02) | 77.40 (9.61) | 100.00 (0.00) | 53.20 (11.26) | 49.80 (10.76) | 48.80 (6.38) | 49.00 (7.58) | 61.60 (10.50) | 63.80 (6.10) |
| Symbols | 30.80 (4.97) | 66.60 (25.23) | 82.20 (27.33) | 62.60 (23.30) | 52.80 (23.19) | 59.80 (19.47) | 52.20 (17.63) | 69.60 (18.81) | 67.20 (15.37) |
| SyntheticControl | 98.40 (0.55) | 75.40 (13.01) | 100.00 (0.00) | 44.20 (18.14) | 48.40 (18.30) | 43.80 (8.41) | 46.00 (19.56) | 60.20 (12.68) | 67.20 (8.53) |
| Trace | 87.40 (17.69) | 39.20 (10.83) | 87.20 (28.62) | 80.60 (3.36) | 61.60 (14.99) | 74.20 (7.60) | 57.00 (15.17) | 86.00 (2.92) | 73.80 (8.61) |
| TwoPatterns | 96.80 (2.39) | 73.80 (1.64) | 100.00 (0.00) | 50.60 (9.69) | 50.00 (6.04) | 47.00 (7.58) | 49.00 (13.67) | 50.50 (5.05) | 71.00 (3.00) |
| UMD | 78.40 (10.19) | 88.60 (10.74) | 100.00 (0.00) | 54.80 (13.16) | 51.60 (5.59) | 54.80 (11.99) | 52.20 (18.36) | 57.40 (12.12) | 69.40 (5.32) |
| UWaveGestureLibraryAll | 40.10 (7.58) | 88.50 (3.78) | 100.00 (0.00) | 55.00 (7.85) | 50.80 (4.16) | 53.60 (7.75) | 50.70 (5.40) | 57.60 (6.33) | 68.20 (1.81) |
| UWaveGestureLibraryX | 29.70 (5.17) | 92.00 (3.56) | 100.00 (0.00) | 46.60 (7.79) | 49.10 (4.86) | 46.20 (4.49) | 49.40 (3.69) | 51.20 (6.20) | 61.70 (3.30) |
| UWaveGestureLibraryY | 18.80 (7.04) | 90.30 (3.56) | 100.00 (0.00) | 56.40 (4.74) | 54.20 (3.26) | 55.20 (4.32) | 53.00 (2.58) | 57.10 (4.77) | 63.80 (3.22) |
| UWaveGestureLibraryZ | 32.30 (8.11) | 89.80 (5.75) | 100.00 (0.00) | 46.50 (13.98) | 48.90 (7.59) | 47.40 (9.05) | 48.80 (5.47) | 52.60 (10.16) | 61.40 (4.58) |
| WordSynonyms | 11.70 (3.27) | 95.00 (4.69) | 100.00 (0.00) | 47.70 (11.09) | 49.80 (7.76) | 47.60 (7.47) | 47.60 (6.70) | 50.60 (6.22) | 62.80 (2.53) |
| Worms | 30.20 (7.44) | 82.60 (20.29) | 92.80 (16.95) | 46.70 (22.17) | 45.30 (14.80) | 47.80 (14.69) | 42.30 (10.56) | 57.80 (17.03) | 60.90 (8.61) |
| Average | 36.50 | 82.02 | 94.57 | 51.52 | 50.68 | 51.52 | 50.67 | 58.01 | 65.51 |

Table A9: Results Scale on the UCR datasets.

| Dataset | F1↑ (known) | FPR95↓ Energy | FPR95↓ MSP | AUROC↑ Energy | AUROC↑ MSP | AUPR (known)↑ Energy | AUPR (known)↑ MSP | AUPR (unknown)↑ Energy | AUPR (unknown)↑ MSP |
|---|---|---|---|---|---|---|---|---|---|
| ACSF1 | 81.30 (2.63) | 52.60 (12.79) | 49.80 (19.33) | 66.00 (11.89) | 71.20 (17.57) | 55.60 (9.17) | 64.40 (16.39) | 76.90 (8.45) | 79.20 (12.26) |
| Adiac | 58.44 (2.07) | 87.22 (10.18) | 91.67 (12.69) | 51.22 (19.05) | 54.00 (17.69) | 50.56 (13.57) | 52.67 (13.68) | 57.44 (13.69) | 63.89 (13.59) |
| AllGestureWiimoteX | 78.80 (0.84) | 86.80 (7.92) | 88.20 (4.71) | 58.60 (7.89) | 59.40 (7.13) | 54.80 (7.53) | 57.20 (8.44) | 61.20 (8.11) | 61.60 (6.23) |
| AllGestureWiimoteY | 74.00 (0.71) | 93.40 (7.70) | 93.20 (8.17) | 45.20 (13.05) | 45.40 (13.46) | 48.60 (8.38) | 47.20 (8.04) | 48.80 (10.80) | 49.60 (12.46) |
| AllGestureWiimoteZ | 72.40 (1.52) | 82.60 (9.71) | 85.40 (10.50) | 61.20 (4.71) | 60.40 (6.19) | 56.40 (3.44) | 58.00 (4.69) | 63.40 (5.59) | 61.20 (8.17) |
| ArrowHead | 84.20 (0.84) | 97.00 (2.74) | 96.60 (4.56) | 42.40 (10.95) | 36.60 (11.15) | 46.80 (8.04) | 43.40 (7.40) | 45.20 (6.22) | 42.80 (6.72) |
| BME | 97.40 (1.14) | 89.20 (10.26) | 85.80 (10.59) | 49.40 (11.89) | 49.60 (8.91) | 51.00 (6.20) | 49.40 (5.13) | 52.80 (11.65) | 55.60 (8.56) |
| Beef | 78.40 (3.13) | 86.00 (13.69) | 84.20 (20.77) | 53.20 (16.35) | 44.80 (15.01) | 50.80 (13.59) | 44.40 (8.35) | 58.40 (11.37) | 53.40 (14.24) |
| CBF | 100.00 (0.00) | 53.00 (12.88) | 58.80 (22.55) | 74.00 (13.45) | 71.40 (13.94) | 69.60 (15.14) | 67.20 (13.70) | 80.40 (8.96) | 75.40 (15.39) |
| Car | 95.40 (2.97) | 79.60 (11.89) | 83.00 (9.87) | 54.00 (11.42) | 53.00 (9.33) | 51.00 (10.05) | 52.40 (9.13) | 61.60 (9.71) | 59.40 (8.02) |
| ChlorineConcentration | 72.80 (18.39) | 93.40 (2.19) | 96.00 (3.67) | 50.80 (3.83) | 50.40 (4.04) | 50.20 (3.03) | 50.60 (2.51) | 52.00 (4.00) | 56.40 (6.54) |
| CinCECGTorso | 70.20 (3.11) | 90.20 (12.05) | 92.80 (6.72) | 49.60 (12.90) | 48.80 (9.52) | 50.00 (9.80) | 50.20 (6.30) | 52.00 (12.55) | 51.00 (8.63) |
| CricketX | 80.00 (0.71) | 89.40 (8.14) | 87.80 (6.22) | 53.00 (4.47) | 56.20 (6.98) | 51.80 (3.56) | 53.80 (5.81) | 56.20 (5.63) | 57.80 (7.50) |
| CricketY | 77.80 (2.17) | 83.20 (10.83) | 84.60 (9.99) | 58.20 (12.72) | 57.00 (12.55) | 55.60 (9.76) | 55.60 (11.13) | 60.80 (12.03) | 60.20 (11.92) |
| CricketZ | 79.20 (1.64) | 92.40 (3.36) | 90.00 (5.34) | 49.00 (9.80) | 50.20 (7.69) | 50.20 (7.79) | 51.20 (6.10) | 52.40 (7.54) | 54.00 (7.58) |
| Crop | 71.60 (0.55) | 94.40 (3.29) | 95.40 (2.88) | 57.40 (7.99) | 48.20 (5.63) | 59.80 (4.76) | 51.20 (5.26) | 53.60 (6.84) | 47.60 (4.16) |
| DiatomSizeReduction | 89.00 (5.52) | 62.20 (36.51) | 88.20 (25.83) | 55.80 (37.73) | 59.40 (26.39) | 58.40 (28.11) | 52.60 (22.33) | 66.20 (29.89) | 70.80 (20.29) |
| DistalPhalanxOutlineAgeGroup | 83.20 (2.17) | 90.00 (6.12) | 100.00 (0.00) | 59.20 (7.19) | 63.00 (3.81) | 54.60 (5.37) | 61.00 (3.74) | 59.00 (7.52) | 71.00 (6.32) |
| DistalPhalanxTW | 45.89 (3.14) | 39.89 (33.51) | 53.56 (24.22) | 72.78 (30.39) | 72.44 (19.14) | 70.33 (24.47) | 67.00 (16.22) | 80.11 (22.15) | 78.00 (16.70) |
| DodgerLoopDay | 41.40 (11.37) | 84.60 (15.14) | 79.60 (17.99) | 58.80 (17.98) | 61.00 (18.83) | 54.20 (11.54) | 59.20 (13.99) | 63.00 (17.07) | 65.00 (17.93) |
| ECG5000 | 70.40 (0.55) | 86.80 (10.23) | 84.00 (11.02) | 72.60 (3.78) | 71.80 (5.67) | 71.20 (4.44) | 69.00 (6.89) | 68.60 (5.27) | 69.40 (4.39) |
| EOGHorizontalSignal | 59.40 (1.52) | 77.80 (9.60) | 84.80 (6.22) | 42.80 (12.60) | 40.80 (14.46) | 42.60 (5.59) | 42.80 (7.22) | 58.20 (9.36) | 53.80 (9.20) |
| EOGVerticalSignal | 44.60 (2.70) | 78.00 (10.65) | 81.00 (6.96) | 55.60 (7.47) | 64.00 (12.39) | 49.00 (5.00) | 61.00 (13.91) | 64.80 (6.10) | 67.80 (9.65) |
| ElectricDevices | 72.00 (0.71) | 97.40 (1.14) | 97.00 (2.55) | 32.00 (5.92) | 38.80 (3.70) | 39.40 (3.05) | 41.80 (1.92) | 40.00 (3.32) | 43.80 (3.11) |
| EthanolLevel | 28.80 (11.45) | 95.40 (3.36) | 95.20 (2.17) | 47.80 (6.46) | 50.00 (4.69) | 50.00 (4.69) | 51.20 (3.56) | 48.00 (5.24) | 50.20 (3.77) |
| FaceAll | 89.80 (2.59) | 60.60 (19.67) | 65.20 (15.93) | 79.00 (5.29) | 77.20 (5.22) | 74.50 (5.34) | 71.60 (5.41) | 80.60 (7.96) | 79.40 (7.23) |
| FaceFour | 85.33 (7.81) | 84.78 (13.52) | 87.11 (10.15) | 46.67 (18.47) | 47.00 (15.00) | 46.67 (11.50) | 48.22 (11.44) | 54.33 (15.01) | 54.67 (11.40) |
| FacesUCR | 91.60 (1.14) | 65.20 (14.25) | 67.80 (11.37) | 70.80 (9.26) | 70.00 (9.46) | 64.40 (8.02) | 63.40 (8.08) | 75.20 (8.23) | 74.60 (8.65) |
| Fish | 93.20 (1.48) | 78.40 (10.21) | 78.20 (15.05) | 56.20 (18.34) | 57.40 (21.40) | 52.20 (12.52) | 56.40 (15.58) | 64.00 (13.82) | 64.40 (17.39) |
| Fungi | 91.60 (2.88) | 57.20 (28.00) | 59.60 (29.59) | 63.00 (22.46) | 61.60 (29.16) | 58.20 (23.64) | 59.40 (23.49) | 73.60 (16.52) | 72.80 (20.17) |
| GestureMidAirD1 | 61.80 (2.86) | 84.40 (11.30) | 77.80 (16.12) | 54.40 (19.36) | 47.80 (10.99) | 54.40 (16.92) | 44.40 (4.98) | 59.80 (11.65) | 59.00 (11.29) |
| GestureMidAirD2 | 51.40 (3.91) | 84.00 (11.42) | 82.80 (19.12) | 55.00 (14.82) | 57.80 (12.03) | 52.00 (10.37) | 54.40 (10.38) | 59.40 (12.86) | 60.20 (13.92) |
| GestureMidAirD3 | 30.60 (2.41) | 83.80 (12.15) | 83.00 (13.82) | 51.80 (13.75) | 52.40 (10.71) | 49.40 (9.61) | 48.40 (5.90) | 58.60 (10.53) | 58.60 (10.95) |
| GesturePebbleZ1 | 92.20 (2.04) | 82.80 (19.20) | 83.80 (14.23) | 60.20 (14.63) | 60.50 (11.69) | 59.70 (11.14) | 58.10 (8.33) | 60.60 (15.76) | 61.80 (12.37) |
| GesturePebbleZ2 | 86.60 (2.12) | 94.00 (6.62) | 94.70 (6.67) | 51.60 (12.32) | 52.50 (11.60) | 55.30 (9.82) | 55.90 (8.56) | 51.40 (10.84) | 51.70 (10.26) |
| Haptics | 51.30 (3.80) | 89.70 (3.77) | 89.10 (4.43) | 54.90 (7.00) | 55.00 (4.08) | 52.80 (6.18) | 53.70 (3.30) | 57.40 (4.99) | 57.10 (3.84) |
| InlineSkate | 38.20 (1.48) | 94.50 (4.35) | 94.00 (3.59) | 47.50 (8.80) | 49.40 (5.64) | 48.00 (7.07) | 48.20 (4.66) | 50.20 (6.63) | 50.80 (5.31) |
| InsectEPGRegularTrain | 100.00 (0.00) | 10.80 (17.61) | 12.40 (20.01) | 90.20 (17.83) | 88.30 (19.12) | 87.00 (21.97) | 85.20 (23.93) | 94.50 (9.92) | 93.40 (10.75) |
| InsectEPGSmallTrain | 100.00 (0.00) | 38.60 (32.87) | 28.00 (21.03) | 63.00 (32.26) | 74.60 (20.35) | 61.40 (26.49) | 66.00 (21.23) | 77.20 (21.79) | 85.40 (12.37) |
| InsectWingbeatSound | 59.90 (0.99) | 83.20 (7.87) | 84.40 (7.14) | 57.30 (8.01) | 58.10 (6.74) | 53.30 (6.88) | 55.40 (5.58) | 62.00 (7.27) | 60.80 (6.84) |
| LargeKitchenAppliances | 89.40 (1.90) | 87.40 (10.49) | 88.20 (10.58) | 57.00 (6.39) | 57.10 (7.19) | 54.50 (4.97) | 54.90 (5.26) | 58.20 (8.15) | 58.30 (8.83) |
| Lightning7 | 79.40 (1.95) | 89.40 (12.52) | 87.80 (20.79) | 47.60 (12.60) | 43.00 (12.86) | 49.60 (6.73) | 45.40 (8.38) | 51.60 (11.41) | 49.80 (13.94) |
| Mallat | 88.00 (2.00) | 93.20 (4.09) | 92.40 (4.16) | 56.20 (10.96) | 54.00 (0.71) | 56.80 (11.65) | 53.00 (1.22) | 54.80 (8.17) | 54.60 (3.36) |
| Meat | 98.00 (2.74) | 71.20 (18.91) | 100.00 (0.00) | 51.60 (26.37) | 52.40 (7.30) | 51.20 (15.83) | 60.40 (14.57) | 65.00 (18.15) | 69.40 (5.77) |
| MedicalImages | 83.40 (1.34) | 96.40 (1.52) | 96.40 (1.82) | 51.20 (2.68) | 48.20 (3.42) | 53.60 (2.70) | 50.00 (2.35) | 50.00 (2.83) | 48.00 (3.32) |
| MelbournePedestrian | 96.00 (0.00) | 94.60 (3.91) | 93.60 (4.72) | 58.40 (10.21) | 51.80 (9.36) | 61.00 (7.38) | 52.80 (7.69) | 55.20 (8.04) | 51.60 (7.96) |
| MiddlePhalanxOutlineAgeGroup | 57.80 (3.56) | 92.00 (5.70) | 93.80 (4.44) | 50.00 (8.46) | 51.00 (2.00) | 50.00 (4.06) | 52.00 (3.20) | 52.20 (6.63) | 52.20 (2.49) |
| MiddlePhalanxTW | 29.40 (7.27) | 72.40 (34.59) | 73.80 (32.88) | 45.60 (38.69) | 51.00 (36.80) | 51.80 (24.28) | 56.60 (26.43) | 58.00 (29.25) | 60.20 (27.26) |
| MixedShapesRegularTrain | 90.60 (0.55) | 75.60 (7.64) | 73.80 (4.27) | 62.60 (14.84) | 64.60 (7.23) | 59.00 (16.31) | 60.20 (8.93) | 68.20 (11.03) | 69.60 (5.81) |
| MixedShapesSmallTrain | 85.20 (1.10) | 48.00 (12.71) | 57.00 (14.70) | 81.60 (7.83) | 75.80 (10.33) | 75.20 (10.38) | 70.40 (11.61) | 84.00 (5.79) | 79.40 (8.62) |
| NonInvasiveFetalECGThorax1 | 91.40 (0.55) | 82.00 (6.96) | 93.80 (5.85) | 52.20 (9.42) | 51.60 (9.58) | 48.20 (5.89) | 49.80 (5.54) | 59.60 (8.85) | 56.40 (8.50) |
| NonInvasiveFetalECGThorax2 | 93.60 (0.55) | 76.80 (22.14) | 89.60 (9.61) | 54.00 (17.33) | 51.20 (4.21) | 54.20 (10.23) | 49.20 (3.11) | 62.20 (15.09) | 56.80 (6.06) |
| OSULeaf | 82.80 (0.84) | 87.40 (6.31) | 87.00 (7.48) | 59.00 (2.24) | 60.00 (4.80) | 54.80 (2.77) | 55.60 (4.51) | 62.40 (3.44) | 63.00 (4.42) |
| OliveOil | 88.00 (8.80) | 83.20 (23.10) | 100.00 (0.00) | 36.80 (17.98) | 47.80 (4.92) | 43.20 (14.66) | 65.00 (22.36) | 50.20 (17.37) | 73.20 (4.02) |
| PLAID | 54.60 (1.82) | 96.60 (4.72) | 97.00 (6.16) | 45.00 (9.49) | 46.80 (9.04) | 48.80 (7.05) | 48.60 (6.58) | 47.60 (7.67) | 48.80 (7.98) |
| PigAirwayPressure | 40.20 (5.07) | 78.60 (15.08) | 76.20 (10.13) | 53.00 (16.39) | 57.60 (14.54) | 48.20 (9.78) | 50.40 (8.08) | 61.00 (13.71) | 64.80 (10.18) |
| PigArtPressure | 82.20 (1.70) | 65.60 (30.14) | 76.40 (21.27) | 52.20 (24.75) | 53.20 (19.92) | 48.00 (14.88) | 48.20 (11.41) | 64.00 (21.13) | 61.20 (18.55) |
| PigCVP | 61.20 (5.89) | 57.00 (21.42) | 56.60 (28.47) | 64.20 (22.29) | 64.40 (24.31) | 55.60 (14.05) | 55.40 (13.76) | 73.20 (17.46) | 72.60 (19.48) |
| Plane | 99.80 (0.45) | 54.00 (12.25) | 54.60 (11.06) | 75.40 (10.74) | 76.40 (10.69) | 66.00 (11.77) | 67.40 (12.86) | 81.80 (7.29) | 82.40 (6.95) |
| ProximalPhalanxOutlineAgeGroup | 68.80 (3.90) | 57.40 (29.42) | 96.60 (6.07) | 70.20 (32.15) | 47.80 (29.10) | 63.40 (20.96) | 53.00 (22.15) | 72.80 (22.58) | 60.00 (19.01) |
| ProximalPhalanxTW | 50.60 (1.95) | 70.40 (31.45) | 88.60 (25.49) | 47.60 (30.90) | 46.60 (21.03) | 50.20 (21.91) | 44.60 (18.08) | 59.80 (25.72) | 64.00 (14.68) |
| RefrigerationDevices | 78.40 (2.30) | 86.80 (8.67) | 89.40 (4.67) | 60.20 (5.63) | 57.00 (3.94) | 59.00 (5.15) | 53.80 (3.27) | 61.00 (7.62) | 59.20 (6.06) |
| Rock | 76.00 (7.09) | 85.93 (11.58) | 86.67 (9.44) | 57.73 (12.10) | 58.07 (6.73) | 58.00 (11.21) | 57.27 (7.27) | 61.13 (10.67) | 60.40 (7.53) |
| ScreenType | 42.17 (9.96) | 93.50 (4.10) | 92.75 (4.99) | 49.08 (8.55) | 49.42 (8.84) | 49.33 (6.43) | 49.67 (6.23) | 51.67 (6.26) | 51.83 (6.58) |
| SemgHandMovementCh2 | 86.60 (0.97) | 86.90 (7.06) | 87.20 (8.07) | 63.00 (6.02) | 61.20 (6.09) | 59.50 (5.11) | 59.20 (6.51) | 62.60 (6.65) | 61.30 (6.52) |
| SemgHandSubjectCh2 | 96.40 (0.52) | 96.70 (4.08) | 95.70 (4.24) | 47.40 (8.67) | 50.00 (6.53) | 50.40 (6.36) | 50.80 (4.73) | 47.60 (6.70) | 50.20 (5.85) |
| ShakeGestureWiimoteZ | 86.60 (3.41) | 48.10 (26.32) | 48.50 (18.56) | 80.10 (11.20) | 75.10 (12.76) | 74.40 (13.27) | 66.20 (13.10) | 82.70 (12.03) | 81.20 (10.45) |
| ShapesAll | 82.67 (0.82) | 79.00 (11.51) | 84.33 (11.64) | 61.50 (9.27) | 60.00 (9.08) | 55.00 (6.60) | 56.00 (8.85) | 66.67 (8.89) | 63.67 (8.33) |
| SmallKitchenAppliances | 85.00 (1.58) | 84.00 (14.68) | 90.40 (5.18) | 55.20 (22.60) | 51.20 (12.21) | 56.80 (17.48) | 51.40 (8.96) | 59.40 (17.70) | 54.80 (9.28) |
| SmoothSubspace | 98.60 (1.52) | 76.80 (4.97) | 76.00 (8.03) | 73.40 (6.15) | 75.00 (3.81) | 70.00 (7.57) | 73.00 (3.74) | 73.00 (6.32) | 74.00 (3.08) |
| StarLightCurves | 100.00 (0.00) | 97.40 (0.89) | 99.00 (1.41) | 26.40 (3.91) | 25.80 (4.49) | 36.40 (0.89) | 36.40 (1.34) | 39.60 (1.67) | 41.80 (3.11) |
| SwedishLeaf | 93.00 (0.71) | 76.40 (11.04) | 77.20 (15.25) | 59.20 (13.83) | 61.00 (14.14) | 54.20 (10.23) | 57.00 (9.82) | 66.00 (11.25) | 66.00 (13.91) |
| Symbols | 92.60 (1.95) | 57.00 (30.68) | 66.20 (19.87) | 75.80 (22.16) | 72.40 (8.96) | 74.80 (20.81) | 68.00 (9.67) | 77.40 (21.08) | 75.00 (9.75) |
| SyntheticControl | 100.00 (0.00) | 52.60 (17.40) | 42.40 (8.20) | 75.00 (10.68) | 79.20 (7.63) | 64.40 (10.69) | 67.20 (8.96) | 81.40 (8.85) | 85.00 (5.15) |
| Trace | 100.00 (0.00) | 76.20 (21.21) | 70.20 (16.19) | 58.80 (18.79) | 55.40 (8.23) | 56.20 (14.65) | 49.20 (5.97) | 64.00 (18.08) | 66.00 (10.42) |
| TwoPatterns | 100.00 (0.00) | 66.00 (11.64) | 78.60 (29.91) | 77.60 (9.18) | 80.60 (9.29) | 73.00 (11.62) | 78.80 (10.73) | 78.40 (7.30) | 81.60 (8.17) |
| UMD | 99.00 (0.71) | 95.20 (6.57) | 89.00 (19.49) | 54.60 (11.91) | 59.20 (11.97) | 56.20 (9.88) | 63.00 (6.20) | 52.60 (10.88) | 56.00 (15.15) |
| UWaveGestureLibraryAll | 91.10 (0.32) | 76.90 (5.59) | 77.40 (5.56) | 68.50 (3.54) | 66.10 (2.92) | 65.50 (4.35) | 62.40 (2.27) | 70.60 (3.63) | 68.70 (3.71) |
| UWaveGestureLibraryX | 76.20 (0.42) | 83.30 (5.85) | 84.10 (5.43) | 60.70 (6.86) | 60.50 (6.15) | 56.70 (6.52) | 57.20 (4.32) | 63.50 (6.57) | 62.80 (6.56) |
| UWaveGestureLibraryY | 70.60 (0.52) | 83.50 (7.31) | 83.30 (5.46) | 62.70 (2.67) | 61.80 (2.62) | 59.10 (2.85) | 58.70 (1.77) | 63.50 (4.60) | 63.20 (4.39) |
| UWaveGestureLibraryZ | 76.90 (0.32) | 91.90 (6.06) | 92.50 (5.36) | 48.10 (14.16) | 48.60 (10.96) | 49.90 (9.43) | 49.40 (6.59) | 52.00 (11.65) | 51.60 (9.06) |
| WordSynonyms | 49.30 (3.09) | 84.20 (12.14) | 83.40 (13.05) | 55.40 (13.56) | 55.00 (13.48) | 53.10 (9.62) | 52.10 (8.85) | 60.60 (12.56) | 60.60 (12.90) |
| Worms | 64.30 (5.56) | 88.10 (12.02) | 88.50 (8.42) | 54.30 (11.85) | 51.50 (11.19) | 52.30 (8.87) | 50.60 (9.65) | 58.30 (11.48) | 57.80 (8.52) |
| Average | 76.94 | 78.92 | 81.57 | 57.45 | 57.28 | 55.94 | 55.80 | 61.99 | 62.33 |

Table A10: Results of ReAct on the UCR datasets.

| Dataset | F1↑ (known) | FPR95↓ Energy | FPR95↓ MSP | AUROC↑ Energy | AUROC↑ MSP | AUPR (known)↑ Energy | AUPR (known)↑ MSP | AUPR (unknown)↑ Energy | AUPR (unknown)↑ MSP |
|---|---|---|---|---|---|---|---|---|---|
| ACSF1 | 81.30 (2.63) | 62.50 (6.52) | 54.00 (6.96) | 63.10 (6.35) | 71.90 (4.41) | 52.70 (4.88) | 60.30 (4.81) | 74.00 (3.92) | 79.90 (3.28) |
| Adiac | 58.44 (2.07) | 62.00 (14.92) | 56.78 (11.76) | 70.22 (11.11) | 76.78 (6.20) | 61.67 (10.62) | 68.22 (7.29) | 77.11 (8.01) | 81.33 (4.85) |
| AllGestureWiimoteX | 78.80 (0.84) | 80.40 (5.27) | 77.80 (4.87) | 68.60 (3.29) | 70.60 (1.52) | 63.60 (5.03) | 67.20 (1.92) | 68.20 (3.03) | 70.20 (2.59) |
| AllGestureWiimoteY | 74.00 (0.71) | 95.20 (3.63) | 96.00 (3.08) | 61.80 (6.46) | 61.60 (4.34) | 60.80 (5.89) | 60.40 (4.39) | 57.40 (7.50) | 57.80 (5.93) |
| AllGestureWiimoteZ | 72.40 (1.52) | 81.80 (5.50) | 82.40 (3.65) | 63.00 (4.53) | 63.20 (3.77) | 59.00 (4.74) | 59.80 (3.27) | 65.40 (4.39) | 65.00 (3.67) |
| ArrowHead | 84.20 (0.84) | 89.40 (10.19) | 94.40 (6.27) | 49.60 (12.36) | 44.80 (9.31) | 51.80 (9.65) | 48.40 (5.98) | 52.20 (9.93) | 48.20 (8.56) |
| BME | 97.40 (1.14) | 81.80 (21.08) | 80.80 (15.27) | 61.00 (14.51) | 64.60 (15.85) | 60.20 (7.46) | 64.40 (13.15) | 61.20 (15.47) | 63.60 (16.07) |
| Beef | 78.40 (3.13) | 77.60 (20.11) | 72.60 (14.67) | 55.80 (16.77) | 52.20 (10.28) | 52.20 (12.19) | 46.00 (1.30) | 62.60 (14.19) | 64.00 (8.56) |
| CBF | 100.00 (0.00) | 63.40 (21.87) | 77.60 (33.84) | 73.00 (19.07) | 62.20 (22.84) | 72.20 (17.77) | 67.20 (17.17) | 75.80 (16.32) | 62.00 (22.88) |
| Car | 95.40 (2.97) | 63.80 (23.13) | 57.60 (20.96) | 77.20 (8.98) | 77.00 (3.08) | 72.20 (11.56) | 73.00 (4.85) | 79.00 (12.35) | 80.60 (5.94) |
| ChlorineConcentration | 72.80 (18.39) | 94.80 (2.59) | 94.60 (2.07) | 54.60 (2.97) | 55.80 (3.03) | 53.60 (3.58) | 55.40 (2.19) | 53.80 (3.03) | 54.20 (1.92) |
| CinCECGTorso | 70.20 (3.11) | 90.00 (14.95) | 91.20 (9.83) | 51.20 (10.13) | 52.80 (5.81) | 50.60 (6.11) | 55.00 (2.92) | 52.80 (12.95) | 52.60 (9.29) |
| CricketX | 80.00 (0.71) | 49.00 (5.96) | 48.60 (5.37) | 77.20 (4.82) | 80.00 (3.16) | 67.60 (5.32) | 73.80 (3.56) | 82.40 (3.44) | 83.60 (3.29) |
| CricketY | 77.80 (2.17) | 86.00 (5.10) | 83.80 (3.63) | 59.40 (4.16) | 58.60 (3.29) | 57.60 (3.85) | 56.20 (3.03) | 61.60 (4.39) | 61.40 (3.21) |
| CricketZ | 79.20 (1.64) | 65.60 (9.76) | 68.20 (11.52) | 73.80 (4.09) | 74.00 (2.55) | 66.40 (5.50) | 67.80 (3.03) | 78.00 (3.08) | 77.40 (4.04) |
| Crop | 71.60 (0.55) | 93.20 (6.02) | 96.40 (1.14) | 48.60 (6.35) | 38.80 (5.07) | 49.60 (5.18) | 41.00 (2.45) | 50.20 (5.81) | 45.00 (2.92) |
| DiatomSizeReduction | 89.00 (5.52) | 36.00 (38.91) | 68.40 (29.23) | 76.00 (37.17) | 58.20 (33.36) | 75.60 (24.83) | 58.60 (19.63) | 82.20 (27.85) | 67.20 (23.37) |
| DistalPhalanxOutlineAgeGroup | 83.20 (2.17) | 94.80 (2.68) | 93.40 (2.97) | 34.80 (9.78) | 36.60 (2.41) | 39.60 (3.85) | 40.20 (1.30) | 44.80 (4.76) | 46.80 (2.05) |
| DistalPhalanxTW | 45.89 (3.14) | 94.22 (5.87) | 93.00 (3.74) | 12.22 (7.16) | 15.00 (7.05) | 33.00 (1.73) | 33.56 (1.67) | 37.67 (5.74) | 39.56 (4.36) |
| DodgerLoopDay | 41.40 (11.37) | 86.40 (10.88) | 84.20 (14.22) | 47.40 (6.73) | 47.40 (10.92) | 45.60 (4.22) | 45.40 (6.07) | 55.40 (9.76) | 57.40 (11.10) |
| ECG5000 | 70.40 (0.55) | 86.00 (8.15) | 74.80 (22.32) | 77.60 (4.62) | 79.60 (2.51) | 76.40 (2.97) | 76.20 (1.92) | 73.20 (4.76) | 76.80 (6.76) |
| EOGHorizontalSignal | 59.40 (1.52) | 98.40 (1.34) | 98.60 (1.14) | 21.20 (1.48) | 20.00 (3.61) | 34.80 (0.45) | 34.20 (1.10) | 35.80 (0.84) | 35.40 (1.52) |
| EOGVerticalSignal | 44.60 (2.70) | 83.40 (15.37) | 77.80 (16.84) | 49.60 (18.09) | 61.80 (16.81) | 47.20 (10.01) | 58.80 (16.10) | 58.00 (17.09) | 66.00 (14.58) |
| ElectricDevices | 72.00 (0.71) | 89.80 (5.36) | 86.20 (8.14) | 46.60 (3.78) | 53.20 (2.49) | 49.80 (2.39) | 49.40 (1.52) | 52.40 (5.13) | 58.40 (6.07) |
| EthanolLevel | 28.80 (11.45) | 94.20 (3.27) | 92.00 (3.08) | 47.00 (3.54) | 51.40 (3.58) | 47.60 (2.41) | 52.20 (3.42) | 49.00 (3.08) | 53.20 (3.63) |
| FaceAll | 89.80 (2.59) | 27.00 (12.86) | 29.40 (9.94) | 90.00 (4.06) | 88.20 (3.77) | 85.00 (4.95) | 82.20 (3.96) | 92.60 (3.36) | 91.40 (2.70) |
| FaceFour | 85.33 (7.81) | 70.00 (22.64) | 66.00 (20.81) | 67.67 (13.36) | 75.89 (9.03) | 60.33 (10.01) | 71.44 (10.76) | 72.11 (13.76) | 78.00 (9.22) |
| FacesUCR | 91.60 (1.14) | 38.00 (8.57) | 39.60 (6.50) | 86.20 (2.95) | 87.40 (1.14) | 80.80 (5.26) | 84.20 (1.64) | 88.80 (2.17) | 89.40 (1.14) |
| Fish | 93.20 (1.48) | 65.60 (18.61) | 63.60 (18.57) | 73.40 (9.50) | 73.60 (9.34) | 66.00 (9.03) | 65.60 (7.37) | 77.60 (8.76) | 78.00 (10.07) |
| Fungi | 91.60 (2.88) | 43.60 (7.09) | 44.00 (14.66) | 76.00 (9.19) | 73.20 (14.20) | 65.60 (12.64) | 62.00 (13.10) | 84.20 (5.07) | 82.40 (9.66) |
| GestureMidAirD1 | 61.80 (2.86) | 47.60 (24.35) | 49.60 (15.32) | 80.00 (10.49) | 72.20 (9.83) | 70.40 (10.24) | 59.80 (7.92) | 83.20 (10.21) | 79.80 (8.17) |
| GestureMidAirD2 | 51.40 (3.91) | 54.40 (22.78) | 63.20 (24.93) | 72.20 (9.55) | 68.00 (9.17) | 60.80 (6.22) | 57.40 (5.73) | 76.80 (12.30) | 72.40 (14.12) |
| GestureMidAirD3 | 30.60 (2.41) | 64.80 (26.05) | 72.20 (13.81) | 68.40 (14.47) | 60.60 (7.37) | 59.20 (11.03) | 51.40 (4.04) | 72.80 (14.70) | 67.40 (8.65) |
| GesturePebbleZ1 | 92.20 (2.04) | 92.00 (12.87) | 95.90 (5.34) | 45.40 (10.25) | 45.90 (5.17) | 47.50 (5.44) | 47.90 (3.51) | 48.30 (11.31) | 46.20 (6.35) |
| GesturePebbleZ2 | 86.60 (2.12) | 96.90 (4.93) | 97.90 (3.45) | 43.20 (7.87) | 42.80 (7.08) | 47.50 (5.44) | 47.60 (4.40) | 45.50 (8.45) | 44.20 (6.88) |
| Haptics | 51.30 (3.80) | 91.80 (6.07) | 88.70 (6.33) | 47.60 (8.81) | 49.00 (6.43) | 48.00 (5.33) | 48.50 (4.50) | 51.00 (7.09) | 53.50 (6.87) |
| InlineSkate | 38.20 (1.48) | 93.60 (4.12) | 92.60 (4.25) | 53.90 (6.38) | 54.00 (7.02) | 52.50 (5.72) | 52.60 (5.04) | 54.70 (5.36) | 54.30 (5.72) |
| InsectEPGRegularTrain | 100.00 (0.00) | 22.00 (34.15) | 18.80 (33.61) | 80.60 (34.12) | 85.20 (32.79) | 82.30 (28.39) | 87.50 (25.63) | 87.40 (22.86) | 90.10 (22.17) |
| InsectEPGSmallTrain | 100.00 (0.00) | 63.50 (48.11) | 64.50 (41.70) | 38.00 (49.17) | 41.30 (46.48) | 44.70 (30.26) | 45.20 (30.16) | 57.30 (34.16) | 56.00 (32.05) |
| InsectWingbeatSound | 59.90 (0.99) | 90.90 (6.05) | 89.10 (5.15) | 50.60 (11.32) | 51.40 (7.24) | 50.90 (9.10) | 48.90 (4.84) | 53.50 (8.62) | 55.50 (6.62) |
| LargeKitchenAppliances | 89.40 (1.90) | 88.90 (9.34) | 87.90 (8.60) | 60.50 (6.28) | 63.10 (4.25) | 58.10 (5.86) | 59.40 (3.60) | 60.40 (7.99) | 62.50 (6.74) |
| Lightning7 | 79.40 (1.95) | 89.00 (8.46) | 91.20 (5.67) | 56.40 (8.05) | 54.80 (4.87) | 57.40 (11.08) | 52.00 (3.46) | 57.40 (5.13) | 56.20 (7.33) |
| Mallat | 88.00 (2.00) | 44.80 (24.83) | 38.00 (12.63) | 86.60 (8.38) | 86.80 (4.21) | 85.40 (8.56) | 81.40 (5.13) | 87.20 (9.36) | 89.40 (4.22) |
| Meat | 98.00 (2.54) | 45.20 (29.30) | 37.00 (18.23) | 79.00 (17.85) | 80.80 (15.79) | 76.20 (18.95) | 75.80 (19.42) | 82.60 (14.62) | 86.40 (10.31) |
| MedicalImages | 83.40 (1.34) | 91.00 (3.08) | 91.00 (2.55) | 59.80 (3.70) | 60.20 (4.21) | 61.20 (2.39) | 59.60 (3.65) | 57.80 (4.02) | 57.60 (4.04) |
| MelbournePedestrian | 96.00 (0.00) | 38.80 (16.24) | 44.00 (13.42) | 88.60 (3.51) | 84.60 (3.29) | 85.60 (4.98) | 79.00 (3.24) | 88.00 (5.48) | 85.20 (3.90) |
| MiddlePhalanxOutlineAgeGroup | 57.80 (3.56) | 95.40 (4.67) | 90.60 (6.07) | 43.60 (4.96) | 45.00 (6.20) | 46.40 (6.19) | 46.00 (4.00) | 47.00 (7.07) | 51.40 (5.13) |
| MiddlePhalanxTW | 29.40 (7.27) | 93.60 (13.76) | 99.60 (0.89) | 15.20 (27.91) | 3.80 (2.77) | 36.40 (10.48) | 31.20 (0.45) | 39.60 (18.68) | 31.20 (0.45) |
| MixedShapesRegularTrain | 90.60 (0.55) | 37.60 (14.52) | 37.00 (10.95) | 85.80 (6.06) | 85.20 (5.17) | 80.60 (7.96) | 80.20 (6.91) | 88.80 (4.97) | 88.20 (4.15) |
| MixedShapesSmallTrain | 85.20 (1.10) | 36.00 (10.20) | 34.80 (12.60) | 87.80 (3.56) | 86.60 (4.22) | 81.80 (4.92) | 81.60 (5.22) | 90.00 (3.61) | 89.40 (3.85) |
| NonInvasiveFetalECGThorax1 | 91.40 (0.55) | 45.60 (19.11) | 52.80 (20.25) | 75.20 (14.29) | 70.40 (16.10) | 66.60 (14.66) | 62.60 (15.47) | 81.60 (10.31) | 78.20 (11.88) |
| NonInvasiveFetalECGThorax2 | 93.60 (0.55) | 10.60 (9.74) | 18.60 (10.29) | 96.20 (4.27) | 94.40 (3.58) | 92.60 (8.29) | 91.60 (6.80) | 97.60 (2.79) | 95.80 (2.86) |
| OSULeaf | 82.80 (0.84) | 71.20 (13.50) | 70.00 (13.11) | 74.80 (2.17) | 75.80 (3.42) | 69.20 (2.28) | 72.60 (2.97) | 76.40 (2.61) | 77.40 (4.22) |
| OliveOil | 88.00 (8.80) | 78.80 (15.34) | 75.60 (20.07) | 59.00 (16.09) | 63.80 (12.07) | 55.00 (12.23) | 59.40 (12.30) | 64.40 (15.52) | 67.80 (13.54) |
| PLAID | 54.60 (1.82) | 95.00 (5.83) | 94.20 (6.06) | 54.60 (3.85) | 54.40 (5.32) | 51.00 (4.74) | 50.20 (3.83) | 56.80 (4.09) | 57.60 (7.67) |
| PigAirwayPressure | 40.20 (5.07) | 79.40 (7.16) | 70.00 (18.59) | 51.40 (3.71) | 49.20 (15.02) | 45.60 (2.88) | 44.80 (6.98) | 60.80 (3.70) | 62.80 (13.10) |
| PigArtPressure | 82.20 (1.79) | 40.40 (9.04) | 31.80 (20.90) | 78.00 (6.28) | 87.40 (4.88) | 62.20 (5.93) | 77.20 (10.55) | 84.40 (4.28) | 89.80 (5.89) |
| PigCVP | 61.20 (5.89) | 51.20 (24.20) | 57.00 (14.51) | 69.20 (14.46) | 70.80 (5.36) | 56.60 (10.50) | 58.40 (5.27) | 77.00 (12.08) | 77.00 (5.00) |
| Plane | 99.80 (0.45) | 1.00 (1.00) | 1.20 (0.84) | 99.40 (0.89) | 99.60 (0.55) | 97.80 (3.83) | 99.40 (0.55) | 99.80 (0.45) | 99.80 (0.45) |
| ProximalPhalanxOutlineAgeGroup | 68.80 (3.09) | 92.60 (9.48) | 95.40 (5.90) | 23.40 (18.01) | 27.20 (13.10) | 36.20 (6.02) | 37.00 (4.18) | 42.00 (13.47) | 41.80 (9.07) |
| ProximalPhalanxTW | 50.60 (1.95) | 71.40 (8.02) | 65.80 (8.04) | 47.60 (7.77) | 49.80 (7.12) | 44.00 (3.54) | 45.20 (3.56) | 63.20 (5.40) | 66.20 (5.67) |
| RefrigerationDevices | 78.40 (2.30) | 89.00 (3.94) | 88.00 (3.81) | 57.80 (5.67) | 55.60 (4.10) | 57.40 (8.18) | 56.20 (6.32) | 58.80 (5.50) | 59.00 (4.03) |
| Rock | 76.00 (7.09) | 75.40 (17.63) | 74.00 (17.57) | 70.87 (6.65) | 72.93 (6.94) | 67.47 (8.88) | 68.13 (10.53) | 73.00 (7.29) | 74.87 (6.70) |
| ScreenType | 42.17 (9.96) | 93.25 (3.79) | 91.75 (5.72) | 48.50 (8.81) | 49.58 (11.43) | 49.08 (6.52) | 50.50 (8.10) | 51.08 (6.36) | 51.83 (8.94) |
| SemgHandMovementCh2 | 86.60 (0.97) | 83.40 (12.76) | 83.00 (9.55) | 68.30 (6.25) | 68.10 (4.31) | 62.60 (6.46) | 64.20 (3.88) | 67.30 (8.27) | 68.10 (5.40) |
| SemgHandSubjectCh2 | 96.40 (0.52) | 95.30 (6.55) | 96.00 (5.14) | 67.50 (8.37) | 70.60 (6.57) | 69.40 (6.36) | 72.10 (4.48) | 59.90 (9.28) | 62.50 (8.80) |
| ShakeGestureWiimoteZ | 86.60 (3.41) | 23.10 (28.59) | 23.40 (25.33) | 90.50 (10.84) | 91.10 (7.53) | 86.50 (14.89) | 86.50 (10.49) | 91.80 (12.27) | 92.60 (8.10) |
| ShapesAll | 82.67 (0.82) | 30.83 (2.71) | 26.33 (6.68) | 83.83 (2.48) | 86.50 (2.35) | 71.50 (4.18) | 75.17 (3.60) | 89.50 (1.52) | 91.17 (1.94) |
| SmallKitchenAppliances | 85.00 (1.58) | 86.60 (14.22) | 87.40 (9.84) | 59.20 (21.15) | 57.40 (10.97) | 61.20 (15.24) | 56.20 (8.53) | 60.80 (16.78) | 59.20 (10.50) |
| SmoothSubspace | 98.60 (1.52) | 75.80 (13.70) | 80.60 (10.43) | 74.20 (4.15) | 75.00 (1.87) | 71.40 (4.72) | 73.80 (3.42) | 72.80 (7.05) | 72.80 (3.77) |
| StarLightCurves | 100.00 (0.00) | 77.40 (25.81) | 84.00 (29.08) | 62.60 (22.84) | 47.00 (25.28) | 61.00 (19.74) | 50.40 (22.17) | 65.80 (19.12) | 54.40 (22.14) |
| SwedishLeaf | 93.00 (0.71) | 73.40 (13.13) | 74.40 (16.65) | 75.80 (6.26) | 76.20 (4.15) | 69.80 (7.85) | 72.60 (4.93) | 76.60 (6.58) | 76.60 (5.50) |
| Symbols | 92.60 (1.95) | 33.60 (27.66) | 23.60 (9.34) | 81.40 (18.61) | 87.80 (6.14) | 75.00 (20.87) | 79.40 (11.13) | 86.40 (14.98) | 92.00 (4.00) |
| SyntheticControl | 100.00 (0.00) | 28.60 (10.19) | 24.20 (7.29) | 89.40 (4.83) | 92.00 (2.55) | 84.00 (5.39) | 86.60 (2.97) | 92.20 (3.77) | 94.20 (2.17) |
| Trace | 100.00 (0.00) | 80.80 (28.15) | 83.20 (17.50) | 33.80 (33.49) | 40.80 (28.27) | 44.40 (21.52) | 48.60 (20.98) | 50.00 (26.01) | 54.00 (21.49) |
| TwoPatterns | 100.00 (0.00) | 62.00 (32.79) | 76.80 (27.17) | 91.00 (5.10) | 90.40 (3.36) | 93.80 (3.42) | 93.80 (1.92) | 84.40 (9.74) | 81.60 (6.73) |
| UMD | 99.00 (0.71) | 89.40 (20.97) | 91.20 (13.41) | 68.60 (19.50) | 70.00 (15.67) | 72.40 (17.84) | 76.40 (9.15) | 63.00 (17.59) | 63.40 (16.06) |
| UWaveGestureLibraryAll | 91.10 (0.32) | 49.80 (10.00) | 52.30 (9.06) | 85.40 (2.80) | 83.30 (2.79) | 82.20 (2.62) | 79.90 (2.47) | 86.50 (3.63) | 85.30 (3.43) |
| UWaveGestureLibraryX | 76.20 (0.42) | 52.90 (7.14) | 54.20 (5.55) | 82.30 (4.22) | 80.70 (3.27) | 77.80 (5.41) | 76.80 (4.02) | 84.10 (3.81) | 82.90 (3.00) |
| UWaveGestureLibraryY | 70.60 (0.52) | 93.70 (4.67) | 95.40 (1.90) | 61.40 (4.09) | 60.60 (1.78) | 59.60 (4.22) | 59.50 (1.58) | 57.70 (4.69) | 56.60 (2.17) |
| UWaveGestureLibraryZ | 76.90 (0.32) | 93.80 (2.57) | 95.20 (2.20) | 51.30 (4.35) | 48.40 (2.41) | 51.30 (3.86) | 49.20 (1.62) | 52.40 (4.35) | 49.50 (2.99) |
| WordSynonyms | 49.30 (3.09) | 52.40 (14.04) | 49.10 (14.89) | 74.40 (7.17) | 75.10 (7.52) | 64.10 (6.79) | 64.90 (7.87) | 81.20 (6.00) | 81.90 (6.24) |
| Worms | 64.30 (5.56) | 89.50 (10.68) | 88.10 (8.74) | 47.30 (16.47) | 48.00 (9.66) | 49.40 (12.81) | 46.80 (6.83) | 53.90 (12.44) | 55.00 (6.99) |
| Average | 76.94 | 69.75 | 69.91 | 63.99 | 64.07 | 62.19 | 62.24 | 68.05 | 68.22 |

Table A11: Results of VOS on the UCR datasets.

| Dataset | F1↑ (known) | FPR95↓ Energy | FPR95↓ MSP | AUROC↑ Energy | AUROC↑ MSP | AUPR (known)↑ Energy | AUPR (known)↑ MSP | AUPR (unknown)↑ Energy | AUPR (unknown)↑ MSP |
|---|---|---|---|---|---|---|---|---|---|
| ACSF1 | 80.75 (3.31) | 78.83 (12.46) | 75.75 (12.17) | 58.50 (7.99) | 61.25 (6.98) | 51.83 (6.77) | 54.42 (8.26) | 65.83 (7.41) | 69.25 (5.74) |
| Adiac | 54.30 (15.66) | 58.00 (13.16) | 62.90 (19.13) | 75.40 (4.84) | 71.10 (9.64) | 65.40 (4.81) | 61.40 (7.15) | 81.10 (4.82) | 76.80 (10.15) |
| AllGestureWiimoteX | 76.10 (2.73) | 86.50 (3.17) | 86.20 (4.59) | 63.70 (3.97) | 59.60 (8.77) | 61.00 (3.53) | 55.60 (8.36) | 62.50 (4.30) | 61.20 (6.60) |
| AllGestureWiimoteY | 71.40 (2.32) | 82.50 (13.75) | 81.70 (13.86) | 60.90 (4.31) | 70.00 (8.55) | 56.90 (6.54) | 69.00 (8.04) | 63.40 (7.56) | 67.60 (11.49) |
| AllGestureWiimoteZ | 71.10 (1.91) | 81.70 (8.47) | 82.60 (7.86) | 62.10 (4.33) | 60.80 (4.52) | 58.20 (3.36) | 56.10 (3.81) | 64.70 (4.95) | 64.10 (4.31) |
| ArrowHead | 78.90 (3.73) | 88.10 (10.57) | 88.70 (11.59) | 54.70 (11.24) | 52.00 (8.41) | 54.00 (7.09) | 52.20 (5.07) | 57.30 (11.60) | 54.90 (10.25) |
| BME | 96.60 (2.50) | 75.70 (27.92) | 76.60 (19.55) | 66.40 (22.12) | 75.00 (7.51) | 67.40 (19.07) | 71.90 (5.78) | 65.20 (20.54) | 71.90 (10.16) |
| Beef | 80.90 (4.82) | 88.00 (8.56) | 84.30 (10.56) | 41.40 (9.48) | 41.10 (9.83) | 42.60 (5.74) | 43.00 (7.86) | 52.40 (7.55) | 53.30 (8.72) |
| CBF | 91.60 (9.35) | 66.80 (33.14) | 93.40 (6.29) | 65.50 (26.37) | 49.60 (15.31) | 68.00 (21.81) | 54.30 (12.02) | 68.50 (23.12) | 51.10 (11.77) |
| Car | 91.50 (6.06) | 74.30 (21.99) | 69.90 (21.06) | 67.20 (14.57) | 75.60 (8.85) | 62.60 (12.84) | 70.80 (8.85) | 70.80 (14.34) | 77.00 (9.21) |
| ChlorineConcentration | 82.50 (1.58) | 92.40 (6.33) | 93.90 (7.77) | 58.40 (4.74) | 59.60 (4.45) | 56.80 (2.78) | 57.50 (3.14) | 57.00 (6.41) | 61.30 (2.98) |
| CinCECGTorso | 72.70 (3.53) | 77.20 (12.87) | 70.80 (17.02) | 64.10 (9.12) | 67.20 (10.44) | 59.50 (6.64) | 62.80 (8.35) | 67.00 (11.13) | 70.60 (12.03) |
| CricketX | 74.00 (5.62) | 62.70 (12.98) | 59.40 (11.61) | 66.60 (14.29) | 76.60 (3.13) | 61.00 (13.19) | 67.20 (10.50) | 75.00 (8.98) | 82.60 (5.41) |
| CricketY | 73.40 (4.99) | 87.30 (6.95) | 88.90 (7.78) | 56.10 (5.02) | 62.80 (4.89) | 53.90 (5.63) | 62.80 (6.81) | 59.90 (4.04) | 62.90 (5.69) |
| CricketZ | 73.40 (7.03) | 73.90 (9.69) | 68.10 (12.20) | 60.70 (11.39) | 72.40 (5.70) | 56.50 (10.10) | 67.90 (8.62) | 68.60 (7.29) | 75.70 (6.43) |
| Crop | 70.80 (0.42) | 96.30 (5.17) | 94.20 (10.16) | 46.20 (9.11) | 58.30 (6.88) | 44.70 (6.75) | 57.80 (6.46) | 47.90 (7.13) | 63.00 (4.32) |
| DiatomSizeReduction | 89.80 (3.97) | 31.00 (30.79) | 27.60 (20.34) | 82.20 (17.29) | 89.00 (8.51) | 74.50 (19.16) | 81.90 (11.93) | 87.10 (14.90) | 92.00 (6.38) |
| DistalPhalanxOutlineAgeGroup | 76.10 (11.45) | 95.10 (8.58) | 95.50 (7.74) | 30.50 (12.37) | 29.90 (11.19) | 39.80 (5.43) | 39.00 (4.16) | 41.80 (10.53) | 41.20 (9.81) |
| DistalPhalanxTW | 40.30 (8.10) | 96.80 (4.13) | 96.60 (4.14) | 9.20 (6.99) | 10.50 (6.62) | 32.30 (1.49) | 32.60 (1.58) | 34.80 (4.80) | 35.50 (4.74) |
| DodgerLoopDay | 36.10 (7.98) | 88.80 (6.92) | 86.10 (8.44) | 43.60 (12.27) | 53.50 (16.03) | 47.70 (11.64) | 54.30 (12.77) | 52.10 (7.71) | 58.60 (10.66) |
| ECG5000 | 69.20 (0.63) | 72.10 (17.28) | 85.80 (17.29) | 81.60 (4.81) | 82.30 (5.70) | 77.90 (3.21) | 79.90 (6.31) | 77.40 (7.68) | 81.20 (3.49) |
| EOGHorizontalSignal | 59.40 (2.88) | 98.80 (1.64) | 99.80 (0.45) | 17.40 (11.15) | 15.40 (8.65) | 34.20 (3.90) | 33.80 (2.95) | 34.80 (4.38) | 33.60 (2.51) |
| EOGVerticalSignal | 45.40 (2.97) | 81.00 (13.32) | 83.20 (10.08) | 60.40 (4.28) | 59.00 (4.30) | 55.60 (4.28) | 54.60 (3.85) | 62.80 (6.57) | 61.60 (6.23) |
| ElectricDevices | 68.00 (0.93) | 88.62 (6.74) | 98.12 (5.30) | 53.88 (6.08) | 54.25 (4.40) | 50.62 (4.44) | 53.37 (3.93) | 56.75 (7.05) | 62.75 (3.81) |
| EthanolLevel | 46.40 (8.71) | 98.40 (0.89) | 100.00 (0.00) | 29.00 (3.32) | 34.00 (2.92) | 38.20 (1.92) | 40.20 (2.39) | 37.80 (1.48) | 48.60 (2.19) |
| FaceAll | 84.20 (3.63) | 28.80 (3.83) | 30.40 (3.58) | 88.20 (1.30) | 86.80 (2.59) | 80.20 (1.92) | 80.00 (3.39) | 91.60 (1.14) | 90.20 (1.79) |
| FaceFour | 90.00 (5.83) | 66.80 (22.49) | 69.20 (24.35) | 81.60 (6.11) | 81.20 (3.03) | 76.00 (5.83) | 79.80 (3.70) | 80.60 (6.43) | 80.20 (7.29) |
| FacesUCR | 91.40 (0.55) | 33.40 (4.16) | 32.60 (3.97) | 89.80 (2.17) | 88.40 (1.14) | 84.80 (5.26) | 83.40 (2.70) | 91.80 (0.84) | 90.80 (0.45) |
| Fish | 90.60 (6.02) | 77.00 (18.07) | 67.20 (22.85) | 59.20 (15.45) | 67.60 (11.84) | 53.60 (10.97) | 59.60 (7.99) | 66.20 (13.55) | 72.60 (12.72) |
| Fungi | 94.60 (3.21) | 24.00 (31.02) | 26.60 (17.08) | 84.40 (20.98) | 83.20 (12.19) | 80.00 (26.50) | 72.20 (14.41) | 90.00 (13.47) | 89.40 (7.70) |
| GestureMidAirD1 | 58.40 (2.41) | 44.20 (18.05) | 53.40 (12.58) | 74.20 (15.96) | 75.40 (10.26) | 62.00 (13.10) | 65.80 (15.04) | 81.40 (11.72) | 80.80 (7.60) |
| GestureMidAirD2 | 51.60 (3.21) | 31.00 (18.40) | 36.20 (18.07) | 81.80 (11.05) | 77.60 (14.06) | 70.00 (16.40) | 66.80 (16.99) | 87.80 (7.92) | 85.00 (9.38) |
| GestureMidAirD3 | 31.60 (1.67) | 70.40 (14.57) | 63.60 (18.93) | 65.00 (8.72) | 67.20 (14.08) | 57.40 (5.90) | 58.20 (11.10) | 70.80 (8.76) | 73.60 (11.33) |
| GesturePebbleZ1 | 91.40 (0.55) | 98.20 (2.39) | 96.20 (2.39) | 46.40 (8.14) | 48.40 (4.39) | 45.20 (2.89) | 48.80 (2.59) | 46.80 (6.02) | 48.80 (3.70) |
| GesturePebbleZ2 | 86.60 (2.41) | 99.00 (1.22) | 98.60 (2.07) | 40.60 (5.41) | 43.20 (7.66) | 45.40 (2.88) | 47.40 (3.85) | 42.20 (3.56) | 43.80 (5.45) |
| Haptics | 50.60 (1.52) | 86.20 (5.76) | 86.00 (5.70) | 59.00 (5.00) | 60.00 (3.87) | 56.40 (4.22) | 57.60 (4.39) | 60.20 (4.44) | 60.80 (3.56) |
| InlineSkate | 40.80 (1.48) | 96.40 (3.97) | 96.20 (3.70) | 54.40 (3.65) | 52.20 (4.38) | 55.00 (4.00) | 52.20 (3.70) | 52.60 (4.93) | 51.40 (5.50) |
| InsectEPGRegularTrain | 100.00 (0.00) | 0.00 (0.00) | 0.00 (0.00) | 100.00 (0.00) | 100.00 (0.00) | 100.00 (0.00) | 100.00 (0.00) | 100.00 (0.00) | 100.00 (0.00) |
| InsectEPGSmallTrain | 100.00 (0.00) | 31.40 (45.60) | 39.20 (53.70) | 68.60 (45.60) | 63.60 (50.25) | 74.60 (35.00) | 73.00 (36.99) | 79.40 (30.80) | 73.80 (35.96) |
| InsectWingbeatSound | 57.80 (1.48) | 85.60 (8.14) | 85.40 (5.77) | 59.20 (8.93) | 60.40 (4.22) | 55.80 (8.07) | 56.60 (3.78) | 62.20 (7.40) | 63.00 (4.80) |
| LargeKitchenAppliances | 90.80 (0.84) | 91.80 (10.83) | 93.40 (9.58) | 63.00 (5.05) | 63.20 (6.50) | 59.40 (3.97) | 60.00 (4.69) | 60.20 (7.46) | 62.40 (8.93) |
| Lightning7 | 78.80 (2.86) | 86.00 (10.07) | 92.00 (5.43) | 57.20 (4.87) | 51.40 (3.65) | 59.40 (8.17) | 50.80 (4.66) | 60.00 (7.11) | 55.20 (5.76) |
| Mallat | 90.40 (4.04) | 38.40 (9.45) | 40.40 (6.23) | 89.40 (4.88) | 85.60 (2.07) | 87.80 (6.50) | 81.60 (3.65) | 90.60 (3.91) | 88.20 (2.59) |
| Meat | 96.20 (2.95) | 28.60 (16.41) | 30.60 (12.30) | 87.00 (9.72) | 85.20 (5.97) | 81.40 (15.18) | 76.60 (10.41) | 90.40 (5.73) | 89.80 (2.86) |
| MedicalImages | 82.60 (1.52) | 93.00 (4.12) | 92.20 (2.59) | 57.80 (3.03) | 57.60 (1.67) | 59.00 (2.55) | 57.60 (1.14) | 55.20 (4.09) | 55.20 (1.64) |
| MelbournePedestrian | 95.40 (0.55) | 68.40 (20.77) | 75.40 (34.65) | 76.00 (7.94) | 81.80 (5.07) | 71.80 (8.44) | 76.00 (6.16) | 74.40 (9.58) | 84.40 (4.77) |
| MiddlePhalanxOutlineAgeGroup | 52.20 (5.93) | 93.40 (6.80) | 92.80 (8.70) | 45.20 (8.35) | 45.40 (8.32) | 40.80 (0.39) | 46.80 (4.21) | 48.40 (7.27) | 49.40 (7.50) |
| MiddlePhalanxTW | 30.80 (2.39) | 96.20 (2.39) | 95.80 (1.79) | 13.80 (5.31) | 14.60 (2.88) | 33.40 (1.52) | 33.20 (0.84) | 36.40 (3.44) | 36.40 (2.07) |
| MixedShapesRegularTrain | 89.60 (0.55) | 29.20 (4.60) | 29.00 (5.52) | 88.00 (2.74) | 88.20 (3.19) | 81.20 (4.55) | 81.80 (5.45) | 91.20 (1.92) | 91.20 (2.28) |
| MixedShapesSmallTrain | 85.60 (0.55) | 39.20 (13.22) | 38.20 (11.63) | 86.40 (5.18) | 85.80 (4.44) | 81.40 (5.86) | 81.20 (5.89) | 89.00 (4.18) | 89.00 (3.54) |
| NonInvasiveFetalECGThorax1 | 91.60 (0.55) | 50.20 (13.90) | 41.80 (33.68) | 81.00 (6.16) | 86.80 (6.38) | 73.40 (8.20) | 80.80 (7.76) | 84.80 (5.31) | 89.80 (6.06) |
| NonInvasiveFetalECGThorax2 | 93.00 (0.00) | 15.20 (6.02) | 12.80 (3.27) | 94.60 (2.30) | 95.00 (1.00) | 89.80 (3.42) | 90.80 (2.28) | 96.00 (1.58) | 96.40 (0.55) |
| OSULeaf | 82.80 (2.05) | 68.80 (8.76) | 64.40 (13.16) | 75.20 (3.27) | 76.80 (5.36) | 70.20 (4.27) | 70.60 (7.27) | 77.20 (2.17) | 79.60 (2.88) |
| OliveOil | 90.60 (6.77) | 63.40 (13.52) | 66.80 (10.16) | 72.40 (10.16) | 72.80 (4.32) | 66.60 (6.75) | 61.80 (4.92) | 76.80 (8.35) | 77.80 (2.95) |
| PLAID | 54.80 (2.59) | 98.40 (1.82) | 98.20 (2.49) | 45.00 (4.69) | 49.40 (4.93) | 46.60 (3.36) | 48.60 (2.07) | 46.20 (2.86) | 49.80 (3.49) |
| PigAirwayPressure | 39.60 (3.85) | 84.80 (9.42) | 79.40 (8.96) | 31.00 (12.04) | 42.20 (12.21) | 37.20 (3.77) | 42.00 (5.05) | 49.20 (7.95) | 56.60 (9.40) |
| PigArtPressure | 84.20 (2.97) | 26.60 (8.44) | 19.80 (11.23) | 87.60 (7.96) | 90.60 (7.83) | 72.20 (9.20) | 79.60 (11.55) | 91.40 (4.45) | 93.40 (5.18) |
| PigCVP | 65.40 (3.65) | 47.60 (14.06) | 43.20 (16.16) | 75.40 (9.24) | 79.20 (7.36) | 61.60 (9.45) | 70.60 (13.45) | 81.80 (7.26) | 84.00 (6.67) |
| Plane | 100.00 (0.00) | 1.60 (0.55) | 2.00 (0.71) | 99.00 (0.71) | 98.60 (0.55) | 96.40 (4.04) | 96.00 (3.32) | 99.40 (0.55) | 99.00 (0.00) |
| ProximalPhalanxOutlineAgeGroup | 66.80 (9.41) | 82.60 (14.28) | 86.40 (18.64) | 50.80 (13.37) | 48.60 (13.94) | 47.80 (8.93) | 46.80 (8.93) | 58.60 (12.92) | 57.80 (12.95) |
| ProximalPhalanxTW | 46.40 (8.11) | 77.60 (26.71) | 77.80 (21.04) | 34.00 (27.08) | 36.40 (24.31) | 42.60 (16.46) | 42.60 (14.26) | 52.60 (20.02) | 54.40 (17.53) |
| RefrigerationDevices | 76.60 (1.82) | 96.40 (1.52) | 100.00 (0.00) | 53.00 (3.39) | 53.40 (3.21) | 52.80 (3.27) | 52.80 (1.64) | 52.00 (3.08) | 55.60 (2.97) |
| Rock | 74.80 (7.69) | 75.60 (11.78) | 81.60 (14.01) | 63.60 (10.29) | 64.60 (9.32) | 57.40 (10.24) | 58.60 (6.77) | 69.80 (8.32) | 68.80 (9.50) |
| ScreenType | 57.80 (1.64) | 94.60 (0.89) | 100.00 (0.00) | 52.20 (1.64) | 52.60 (1.34) | 51.20 (2.39) | 51.80 (1.92) | 52.20 (0.84) | 55.40 (1.14) |
| SemgHandMovementCh2 | 81.80 (9.42) | 89.20 (5.85) | 95.60 (9.84) | 61.20 (5.40) | 59.80 (3.70) | 59.20 (6.57) | 56.80 (3.96) | 61.60 (2.07) | 62.00 (2.55) |
| SemgHandSubjectCh2 | 96.60 (0.55) | 91.40 (9.53) | 92.80 (9.88) | 71.40 (7.57) | 75.20 (3.42) | 71.00 (7.97) | 73.60 (2.07) | 64.00 (7.78) | 73.00 (3.94) |
| ShakeGestureWiimoteZ | 89.60 (2.30) | 26.40 (25.44) | 22.20 (19.15) | 85.00 (14.05) | 90.20 (8.41) | 72.40 (16.32) | 82.20 (13.44) | 89.60 (10.81) | 93.00 (6.28) |
| ShapesAll | 81.00 (1.73) | 36.40 (13.11) | 36.20 (6.98) | 85.00 (4.90) | 81.40 (2.97) | 74.40 (6.19) | 68.80 (3.49) | 89.40 (3.36) | 87.20 (2.28) |
| SmallKitchenAppliances | 79.40 (1.34) | 91.60 (2.30) | 95.40 (6.31) | 52.60 (2.41) | 53.20 (2.28) | 52.20 (1.10) | 52.60 (1.52) | 53.20 (2.28) | 56.00 (2.92) |
| SmoothSubspace | 98.80 (0.84) | 77.20 (15.07) | 82.40 (12.16) | 74.00 (4.24) | 72.20 (5.76) | 71.00 (4.00) | 71.40 (5.41) | 72.80 (5.40) | 69.60 (6.43) |
| StarLightCurves | 100.00 (0.00) | 94.20 (5.76) | 100.00 (0.00) | 42.80 (18.75) | 50.80 (0.45) | 47.40 (15.50) | 60.00 (5.83) | 49.40 (12.78) | 74.80 (0.45) |
| SwedishLeaf | 92.40 (0.55) | 67.60 (6.54) | 71.60 (7.20) | 78.00 (2.35) | 75.80 (2.49) | 71.20 (2.59) | 71.60 (2.97) | 78.80 (2.49) | 77.40 (2.88) |
| Symbols | 93.20 (1.10) | 21.60 (20.42) | 27.20 (23.29) | 90.80 (11.21) | 89.40 (12.16) | 86.40 (14.22) | 86.00 (15.51) | 93.20 (8.58) | 91.60 (10.01) |
| SyntheticControl | 100.00 (0.00) | 28.20 (13.90) | 21.00 (8.03) | 92.00 (3.94) | 93.40 (2.07) | 88.80 (5.97) | 89.20 (4.27) | 93.40 (2.97) | 94.80 (1.30) |
| Trace | 100.00 (0.00) | 73.20 (20.19) | 83.40 (20.55) | 45.80 (25.72) | 33.20 (21.99) | 47.40 (13.01) | 41.00 (11.51) | 59.40 (18.30) | 48.80 (18.83) |
| TwoPatterns | 100.00 (0.00) | 80.80 (14.89) | 100.00 (0.00) | 83.80 (15.66) | 81.60 (8.38) | 89.40 (10.95) | 90.60 (4.39) | 75.40 (15.61) | 86.80 (3.96) |
| UMD | 99.40 (0.55) | 90.60 (10.90) | 86.60 (17.90) | 69.20 (11.71) | 73.40 (8.88) | 73.40 (8.96) | 74.60 (8.14) | 62.00 (10.49) | 66.80 (13.18) |
| UWaveGestureLibraryAll | 89.80 (0.45) | 59.00 (16.12) | 71.60 (27.06) | 82.20 (5.26) | 80.20 (3.96) | 79.60 (4.72) | 76.40 (3.36) | 82.80 (6.72) | 82.40 (4.77) |
| UWaveGestureLibraryX | 74.20 (0.45) | 57.80 (8.73) | 57.80 (7.05) | 80.40 (2.30) | 77.60 (2.07) | 76.20 (2.86) | 72.20 (2.39) | 81.60 (3.51) | 80.80 (2.28) |
| UWaveGestureLibraryY | 68.80 (0.84) | 93.40 (2.97) | 100.00 (0.00) | 61.40 (0.55) | 60.80 (0.84) | 59.20 (1.92) | 58.00 (1.87) | 58.00 (1.41) | 63.00 (0.71) |
| UWaveGestureLibraryZ | 75.40 (0.55) | 91.60 (4.93) | 100.00 (0.00) | 51.00 (4.30) | 51.60 (3.58) | 50.40 (2.88) | 50.20 (2.17) | 53.40 (5.46) | 57.60 (2.88) |
| WordSynonyms | 51.40 (2.88) | 51.40 (5.32) | 52.20 (3.42) | 76.60 (1.52) | 76.40 (2.30) | 67.00 (4.06) | 66.80 (4.66) | 82.20 (1.10) | 82.40 (1.14) |
| Worms | 62.40 (4.72) | 75.00 (6.20) | 74.20 (7.69) | 54.60 (7.92) | 55.80 (7.50) | 49.20 (5.72) | 49.80 (5.36) | 65.40 (3.44) | 66.60 (2.07) |
| Average | 76.40 | 69.13 | 70.54 | 64.39 | 65.47 | 62.53 | 63.45 | 68.26 | 70.06 |

Table A12: Results of our method, ALSET, on the UCR datasets.

| Dataset | F1↑ (known) | FPR95↓ Energy | FPR95↓ MSP | AUROC↑ Energy | AUROC↑ MSP | AUPR (known)↑ Energy | AUPR (known)↑ MSP | AUPR (unknown)↑ Energy | AUPR (unknown)↑ MSP |
|---|---|---|---|---|---|---|---|---|---|
| Adiac | 64.43 (4.12) | 49.14 (15.30) | 46.14 (9.42) | 78.43 (7.37) | 82.29 (4.89) | 69.00 (8.19) | 74.43 (8.32) | 83.43 (6.73) | 86.86 (3.18) |
| AllGestureWiimoteX | 76.64 (1.12) | 84.36 (6.58) | 80.09 (6.33) | 61.73 (4.05) | 68.27 (2.10) | 57.45 (4.52) | 64.73 (1.68) | 63.82 (4.85) | 69.09 (2.70) |
| AllGestureWiimoteY | 73.43 (1.13) | 84.33 (13.15) | 82.57 (15.21) | 63.67 (10.67) | 65.14 (7.52) | 63.00 (10.09) | 62.00 (5.66) | 62.78 (11.23) | 64.71 (10.34) |
| AllGestureWiimoteZ | 73.14 (1.21) | 79.57 (7.28) | 79.57 (5.38) | 63.71 (3.30) | 64.14 (3.58) | 59.43 (3.64) | 60.29 (3.68) | 65.57 (4.50) | 66.00 (4.32) |
| ArrowHead | 86.00 (0.71) | 93.60 (2.70) | 91.20 (6.38) | 52.00 (2.00) | 51.80 (4.66) | 55.40 (2.88) | 55.00 (1.41) | 51.20 (3.35) | 51.00 (4.36) |
| BME | 97.60 (0.89) | 49.80 (15.29) | 27.80 (14.89) | 78.80 (5.45) | 86.00 (3.81) | 69.00 (5.79) | 74.80 (4.27) | 83.20 (5.81) | 90.80 (3.27) |
| Beef | 77.27 (9.18) | 79.18 (13.99) | 77.73 (12.88) | 54.00 (15.26) | 49.27 (12.15) | 50.91 (11.29) | 47.09 (8.72) | 61.45 (12.62) | 60.55 (9.96) |
| CBF | 99.40 (0.55) | 74.40 (42.02) | 70.00 (16.17) | 49.80 (40.87) | 80.20 (12.95) | 59.20 (31.50) | 81.20 (8.44) | 57.60 (30.51) | 78.40 (15.57) |
| Car | 97.60 (0.89) | 53.80 (23.06) | 57.60 (23.99) | 85.60 (3.05) | 84.60 (1.52) | 82.80 (5.50) | 81.00 (3.32) | 85.60 (5.13) | 84.40 (4.67) |
| ChlorineConcentration | 82.40 (1.82) | 96.20 (1.30) | 97.20 (0.84) | 55.20 (2.77) | 55.60 (1.67) | 54.80 (2.39) | 55.60 (1.52) | 52.60 (2.88) | 52.40 (1.34) |
| CinCECGTorso | 74.00 (4.30) | 89.69 (10.40) | 80.20 (15.97) | 52.94 (9.69) | 60.60 (11.74) | 51.50 (7.19) | 59.00 (8.92) | 55.31 (10.61) | 63.20 (13.66) |
| CricketX | 80.00 (1.22) | 59.40 (9.45) | 61.40 (7.09) | 77.60 (5.55) | 78.00 (4.00) | 73.20 (3.83) | 80.20 (5.07) | 79.80 (3.90) | |
| CricketY | 77.40 (1.82) | 90.25 (5.76) | 86.80 (8.32) | 61.00 (5.66) | 62.40 (5.77) | 58.58 (5.85) | 59.00 (2.35) | 60.75 (5.85) | 62.60 (6.73) |
| CricketZ | 81.40 (1.52) | 63.60 (8.47) | 66.80 (9.09) | 73.40 (2.61) | 74.00 (2.35) | 67.00 (4.47) | 67.60 (4.39) | 76.80 (1.92) | 76.80 (2.95) |
| Crop | 55.60 (16.52) | 75.30 (23.63) | 91.20 (4.02) | 51.30 (11.56) | 32.70 (13.62) | 48.20 (6.14) | 39.50 (5.19) | 60.90 (14.19) | 45.30 (6.15) |
| DiatomSizeReduction | 94.80 (3.11) | 13.80 (15.94) | 17.40 (19.96) | 90.60 (11.19) | 90.40 (10.99) | 82.00 (14.61) | 82.00 (14.09) | 94.00 (7.42) | 93.60 (8.20) |
| DistalPhalanxOutlineAgeGroup | 78.40 (5.41) | 85.40 (3.44) | 88.40 (6.99) | 43.40 (6.47) | 39.80 (7.76) | 42.60 (2.51) | 41.80 (2.95) | 54.20 (5.02) | 50.40 (7.37) |
| DistalPhalanxTW | 45.00 (1.22) | 92.40 (5.27) | 91.60 (5.22) | 14.80 (7.56) | 15.60 (7.40) | 33.60 (1.95) | 33.60 (1.67) | 40.20 (5.85) | 41.00 (5.52) |
| DodgerLoopDay | 36.00 (10.46) | 88.20 (12.51) | 84.80 (15.99) | 48.60 (17.63) | 41.20 (13.31) | 49.60 (12.40) | 43.80 (7.56) | 56.10 (12.11) | 62.40 (12.30) |
| ECG5000 | 70.00 (0.00) | 59.40 (18.04) | 57.00 (10.25) | 83.00 (2.00) | 83.00 (1.87) | 80.40 (1.67) | 78.20 (2.17) | 81.40 (3.71) | 82.80 (4.09) |
| EOGHorizontalSignal | 58.30 (2.45) | 97.60 (2.12) | 98.40 (1.17) | 24.60 (5.15) | 19.20 (3.99) | 36.80 (2.94) | 34.20 (1.23) | 37.50 (3.47) | 35.00 (1.49) |
| EOGVerticalSignal | 44.40 (1.95) | 91.00 (5.70) | 90.60 (3.78) | 54.60 (10.01) | 55.40 (7.50) | 52.60 (7.64) | 52.40 (5.59) | 56.20 (9.04) | 58.00 (5.15) |
| ElectricDevices | 69.33 (1.15) | 82.00 (2.00) | 88.00 (3.61) | 65.33 (6.03) | 44.67 (2.52) | 63.00 (4.58) | 43.00 (1.00) | 66.33 (4.16) | 55.00 (3.00) |
| EthanolLevel | 17.71 (0.95) | 97.50 (2.97) | 93.29 (3.40) | 39.00 (13.31) | 50.00 (4.76) | 44.83 (8.05) | 49.86 (4.30) | 43.75 (7.77) | 50.71 (4.35) |
| FaceAll | 85.60 (3.65) | 21.60 (6.39) | 22.80 (4.97) | 90.00 (7.07) | 90.60 (3.44) | 85.40 (13.03) | 86.40 (6.91) | 93.00 (4.06) | 93.40 (2.30) |
| FaceFour | 92.00 (5.70) | 61.20 (18.10) | 61.00 (11.02) | 77.80 (11.52) | 84.00 (5.52) | 70.60 (13.67) | 81.20 (9.34) | 81.20 (8.67) | 84.60 (5.27) |
| FacesUCR | 91.20 (1.30) | 34.20 (5.02) | 34.20 (7.79) | 89.00 (1.73) | 88.40 (0.89) | 84.80 (3.70) | 83.40 (1.34) | 91.00 (2.00) | 90.60 (1.67) |
| Fish | 91.80 (6.06) | 60.20 (31.44) | 56.60 (28.01) | 69.20 (19.47) | 70.00 (15.07) | 63.00 (17.22) | 62.60 (13.05) | 75.00 (17.31) | 76.60 (12.54) |
| Fungi | 96.60 (2.07) | 13.40 (8.02) | 12.40 (3.51) | 90.60 (8.02) | 92.40 (2.41) | 81.40 (15.21) | 81.80 (5.02) | 94.80 (4.32) | 95.40 (1.14) |
| GestureMidAirD1 | 60.60 (1.82) | 44.40 (38.68) | 55.20 (25.51) | 80.00 (14.82) | 74.40 (12.22) | 71.20 (16.84) | 64.40 (13.87) | 82.20 (15.88) | 78.60 (11.50) |
| GestureMidAirD2 | 50.67 (2.77) | 56.87 (24.04) | 67.67 (21.66) | 70.13 (15.20) | 65.20 (10.64) | 60.60 (10.80) | 56.33 (9.71) | 75.60 (14.49) | 70.40 (12.05) |
| GestureMidAirD3 | 31.67 (2.19) | 64.53 (17.63) | 69.87 (13.54) | 67.33 (10.42) | 65.00 (10.45) | 55.20 (5.59) | 56.40 (8.67) | 73.13 (9.55) | 70.93 (9.28) |
| GesturePebbleZ1 | 92.20 (1.30) | 95.80 (4.92) | 94.40 (7.57) | 47.80 (14.04) | 50.20 (11.73) | 49.40 (8.91) | 51.00 (7.94) | 49.60 (10.21) | 50.80 (10.33) |
| GesturePebbleZ2 | 87.40 (1.95) | 96.20 (6.83) | 97.80 (3.27) | 38.60 (8.59) | 40.80 (6.53) | 43.00 (4.00) | 44.80 (2.49) | 43.60 (8.68) | 43.20 (4.66) |
| Haptics | 51.40 (2.30) | 84.60 (8.96) | 84.60 (5.18) | 60.20 (6.80) | 60.80 (6.14) | 56.80 (6.46) | 57.80 (4.44) | 62.80 (5.72) | 63.40 (5.32) |
| InlineSkate | 42.80 (2.17) | 95.80 (3.27) | 96.80 (2.68) | 55.00 (5.61) | 52.60 (4.62) | 55.00 (4.47) | 53.20 (2.95) | 52.20 (5.50) | 51.00 (4.53) |
| InsectEPGRegularTrain | 100.00 (0.00) | 3.40 (7.60) | 0.00 (0.00) | 99.60 (0.89) | 100.00 (0.00) | 99.40 (1.52) | 100.00 (0.00) | 99.60 (0.89) | 100.00 (0.00) |
| InsectEPGSmallTrain | 100.00 (0.00) | 43.40 (51.79) | 23.60 (31.54) | 59.40 (54.23) | 78.20 (29.70) | 71.80 (37.25) | 77.20 (30.82) | 71.80 (37.25) | 86.80 (17.80) |
| InsectWingbeatSound | 58.80 (0.84) | 81.80 (5.07) | 82.80 (5.72) | 62.80 (3.35) | 60.80 (4.82) | 58.80 (3.03) | 56.20 (3.56) | 65.00 (3.94) | 63.40 (5.22) |
| LargeKitchenAppliances | 90.20 (0.84) | 89.00 (8.57) | 89.60 (5.41) | 61.60 (5.61) | 63.40 (4.39) | 60.00 (2.35) | 62.80 (3.70) | 59.20 (7.85) | 61.00 (6.12) |
| Lightning7 | 78.10 (2.42) | 88.70 (7.76) | 84.60 (8.86) | 51.10 (9.28) | 52.90 (9.69) | 51.00 (7.69) | 51.60 (8.04) | 56.00 (8.99) | 59.10 (8.67) |
| Mallat | 90.40 (2.30) | 24.80 (15.06) | 28.80 (7.60) | 92.80 (5.02) | 91.20 (3.90) | 91.60 (6.35) | 88.80 (4.97) | 94.00 (4.30) | 93.00 (2.74) |
| Meat | 100.00 (0.00) | 14.20 (3.11) | 11.20 (1.10) | 93.60 (1.67) | 94.80 (2.17) | 88.60 (5.64) | 94.00 (2.92) | 95.40 (1.14) | 96.20 (1.79) |
| MedicalImages | 84.40 (1.14) | 91.00 (2.24) | 92.40 (4.04) | 62.60 (2.19) | 60.60 (2.07) | 63.40 (3.51) | 60.60 (2.30) | 59.60 (2.70) | 57.60 (2.70) |
| MelbournePedestrian | 96.00 (0.00) | 63.80 (5.72) | 58.60 (6.07) | 83.60 (1.34) | 84.60 (1.52) | 79.80 (1.92) | 79.60 (1.22) | 80.00 (2.45) | 84.20 (2.49) |
| MiddlePhalanxOutlineAgeGroup | 57.50 (6.24) | 89.60 (5.56) | 91.80 (3.19) | 47.40 (11.15) | 41.50 (4.20) | 50.00 (9.61) | 44.70 (2.95) | 52.30 (8.27) | 48.00 (4.08) |
| MiddlePhalanxTW | 28.00 (3.00) | 80.40 (13.07) | 85.20 (10.23) | 34.00 (19.87) | 30.00 (14.66) | 40.80 (8.67) | 38.20 (5.76) | 53.20 (14.29) | 48.80 (11.37) |
| MixedShapesRegularTrain | 89.60 (0.55) | 32.80 (11.88) | 30.80 (9.68) | 88.80 (3.1) | 88.40 (2.41) | 84.00 (3.94) | 83.20 (2.59) | 90.60 (3.51) | 91.00 (3.08) |
| MixedShapesSmallTrain | 85.20 (0.84) | 31.40 (5.90) | 32.80 (7.05) | 89.80 (3.03) | 87.20 (3.11) | 85.40 (4.45) | 82.20 (3.77) | 92.00 (2.12) | 90.60 (2.51) |
| NonInvasiveFetalECGThorax1 | 91.20 (0.45) | 48.40 (25.52) | 38.80 (13.72) | 77.70 (15.47) | 81.20 (8.76) | 70.30 (14.83) | 71.60 (11.33) | 81.50 (15.28) | 86.40 (5.90) |
| NonInvasiveFetalECGThorax2 | 93.20 (0.84) | 8.00 (2.83) | 10.00 (3.08) | 97.60 (1.14) | 96.20 (1.30) | 94.40 (1.52) | 93.40 (2.19) | 98.20 (0.84) | 97.40 (0.89) |
| OSULeaf | 82.80 (2.17) | 60.60 (8.02) | 67.40 (12.05) | 79.00 (6.16) | 76.00 (6.96) | 72.20 (6.72) | 69.60 (7.23) | 82.60 (5.41) | 79.80 (6.02) |
| OliveOil | 86.40 (5.22) | 55.20 (23.89) | 67.60 (26.31) | 73.40 (21.78) | 64.40 (20.65) | 72.00 (17.89) | 61.20 (14.77) | 78.00 (19.80) | 69.80 (19.64) |
| PLAID | 49.00 (1.87) | 85.80 (6.50) | 89.40 (1.52) | 54.60 (11.37) | 55.00 (2.92) | 51.40 (8.79) | 50.80 (5.72) | 61.80 (6.80) | 59.60 (0.89) |
| PigAirwayPressure | 43.40 (3.05) | 75.40 (11.10) | 76.60 (9.86) | 44.20 (11.50) | 52.20 (8.81) | 42.40 (4.88) | 47.80 (7.98) | 59.00 (9.41) | 62.00 (6.44) |
| PigArtPressure | 85.60 (3.29) | 26.20 (4.66) | 12.60 (6.31) | 80.80 (4.60) | 93.60 (4.28) | 64.20 (5.26) | 82.80 (8.61) | 88.60 (2.19) | 96.20 (2.39) |
| PigCVP | 68.00 (2.92) | 44.40 (22.24) | 47.20 (7.19) | 77.60 (8.02) | 76.00 (5.79) | 64.60 (9.10) | 62.20 (5.54) | 83.40 (8.35) | 82.20 (4.21) |
| Plane | 100.00 (0.00) | 1.20 (0.45) | 2.20 (0.84) | 99.60 (0.89) | 99.00 (1.22) | 98.20 (4.02) | 97.20 (4.15) | 99.80 (0.45) | 99.40 (0.55) |
| ProximalPhalanxOutlineAgeGroup | 71.10 (4.01) | 90.27 (7.91) | 92.00 (8.77) | 32.47 (13.99) | 32.10 (12.59) | 39.13 (5.90) | 38.70 (4.00) | 46.13 (9.96) | 45.60 (8.47) |
| ProximalPhalanxTW | 43.70 (13.78) | 76.90 (28.95) | 74.50 (28.53) | 37.10 (28.10) | 39.10 (27.72) | 44.70 (19.41) | 45.50 (19.83) | 53.70 (21.77) | 55.70 (20.92) |
| RefrigerationDevices | 77.53 (1.92) | 94.67 (4.06) | 94.67 (4.91) | 51.27 (3.61) | 52.40 (3.25) | 50.13 (2.90) | 51.07 (2.55) | 52.07 (4.38) | 52.73 (4.53) |
| Rock | 77.80 (3.65) | 71.10 (18.48) | 72.20 (15.92) | 69.20 (8.59) | 70.80 (7.67) | 60.80 (7.76) | 66.20 (8.90) | 74.40 (8.11) | 74.00 (7.38) |
| ScreenType | 57.10 (2.51) | 94.10 (2.69) | 94.50 (2.72) | 51.40 (2.32) | 51.50 (3.27) | 50.30 (1.83) | 50.10 (2.08) | 52.10 (2.60) | 52.10 (3.00) |
| SemgHandMovementCh2 | 85.87 (1.81) | 92.27 (5.50) | 91.93 (4.96) | 63.87 (3.78) | 63.60 (2.75) | 62.93 (4.38) | 62.47 (3.89) | 60.07 (4.92) | 60.80 (4.44) |
| SemgHandSubjectCh2 | 96.63 (0.83) | 95.00 (10.41) | 90.84 (14.87) | 68.26 (7.99) | 71.74 (6.65) | 69.63 (5.87) | 72.79 (4.34) | 60.47 (9.71) | 64.11 (9.22) |
| ShakeGestureWiimoteZ | 89.10 (3.48) | 15.70 (11.89) | 12.70 (8.13) | 93.30 (6.06) | 94.50 (3.47) | 89.20 (10.98) | 90.50 (7.21) | 95.30 (3.95) | 96.10 (2.60) |
| ShapesAll | 80.00 (1.22) | 41.50 (5.15) | 34.00 (13.13) | 81.30 (5.70) | 83.80 (7.05) | 71.60 (9.11) | 72.60 (8.14) | 86.40 (3.20) | 88.40 (5.03) |
| SmallKitchenAppliances | 81.60 (0.89) | 85.60 (3.78) | 85.00 (4.24) | 57.60 (2.51) | 57.80 (1.48) | 54.80 (2.68) | 54.20 (0.84) | 60.20 (3.96) | 60.40 (3.13) |
| SmoothSubspace | 98.60 (0.89) | 86.20 (6.06) | 79.20 (11.69) | 73.80 (3.77) | 76.40 (3.78) | 72.40 (4.77) | 74.20 (4.55) | 70.40 (3.58) | 74.00 (5.15) |
| StarLightCurves | 100.00 (0.00) | 88.40 (21.08) | 92.40 (6.17) | 37.80 (21.12) | 40.70 (14.61) | 44.30 (16.00) | 44.80 (10.18) | 48.80 (17.66) | 49.90 (10.19) |
| SwedishLeaf | 93.10 (0.57) | 70.05 (12.55) | 68.90 (13.12) | 75.40 (4.95) | 76.00 (3.13) | 69.40 (4.18) | 70.80 (4.18) | 76.85 (5.61) | 77.50 (5.32) |
| Symbols | 95.20 (1.30) | 10.40 (10.60) | 7.20 (5.45) | 94.20 (7.09) | 97.80 (1.92) | 89.40 (12.82) | 96.20 (3.03) | 96.40 (4.34) | 97.80 (1.64) |
| SyntheticControl | 100.00 (0.00) | 24.42 (15.47) | 26.40 (4.93) | 91.42 (8.46) | 91.80 (2.28) | 87.83 (10.84) | 87.40 (3.36) | 93.33 (6.37) | 93.20 (1.92) |
| Trace | 100.00 (0.00) | 79.70 (18.77) | 86.30 (20.83) | 39.00 (19.91) | 33.70 (19.34) | 42.90 (8.74) | 40.70 (9.15) | 54.20 (17.20) | 48.80 (17.07) |
| TwoPatterns | 100.00 (0.00) | 45.60 (33.20) | 36.20 (33.80) | 93.40 (4.56) | 96.00 (3.32) | 95.60 (2.97) | 97.20 (2.17) | 88.80 (7.79) | 91.60 (7.30) |
| UMD | 99.30 (0.67) | 87.00 (21.36) | 90.30 (18.07) | 60.50 (14.73) | 62.50 (14.48) | 65.40 (11.92) | 67.20 (10.81) | 57.50 (15.90) | 57.70 (15.11) |
| UWaveGestureLibraryAll | 90.20 (0.45) | 42.20 (8.70) | 44.00 (9.75) | 86.00 (2.92) | 84.80 (2.49) | 82.40 (3.21) | 80.00 (2.92) | 88.60 (2.70) | 87.60 (2.88) |
| UWaveGestureLibraryX | 75.25 (1.07) | 58.45 (7.94) | 58.90 (6.70) | 79.95 (3.89) | 78.00 (2.97) | 75.60 (4.43) | 73.50 (3.65) | 81.50 (4.27) | 80.55 (3.14) |
| UWaveGestureLibraryY | 68.62 (0.74) | 91.40 (6.43) | 91.25 (4.71) | 61.20 (3.70) | 60.50 (2.39) | 59.80 (2.86) | 58.75 (1.98) | 59.00 (5.70) | 59.00 (4.07) |
| UWaveGestureLibraryZ | 75.20 (0.45) | 92.20 (2.95) | 91.60 (2.70) | 52.40 (3.13) | 51.60 (1.52) | 52.00 (1.87) | 50.80 (1.30) | 54.00 (3.39) | 54.20 (2.28) |
| WordSynonyms | 50.20 (2.05) | 48.80 (11.82) | 52.00 (12.06) | 78.80 (5.89) | 78.40 (4.99) | 69.40 (8.02) | 68.80 (5.81) | 84.40 (4.67) | 83.20 (4.49) |
| Worms | 64.80 (3.11) | 88.60 (9.02) | 75.80 (7.95) | 44.00 (19.22) | 57.60 (13.76) | 45.20 (9.86) | 52.20 (10.13) | 53.40 (13.01) | 67.60 (7.77) |
| Average | 77.10 | 65.51 | 64.90 | 66.49 | 67.21 | 64.42 | 64.89 | 70.28 | 70.95 |

