# OpenReview forum: "Time-Series Open-set Recognition with Adaptive Local Outlier Synthesis and Exposure"
_ICLR.cc/2026/Conference — ICLR 2026 Conference Withdrawn Submission_

### Official Review · Reviewer_NaMX · 2025-10-27

**Soundness:** 2
**Presentation:** 2
**Contribution:** 2
**Rating:** 2
**Confidence:** 3

**Summary:**

This paper introduces ALSET, a method for open-set recognition (OSR) in time-series data. ALSET synthesizes outliers in the latent space by sampling from local Gaussian distributions with learnable, per-dimension standard deviations, regulated by a feedback mechanism from the classifier.

**Strengths:**

1. **Application to Time-Series:** The work addresses a relatively underexplored area of OSR for time-series data, which is a valuable contribution.
2. **Comprehensive Empirical Evaluation:** The authors perform extensive experiments across multiple datasets and encoders, providing a solid empirical basis for their claims.

**Weaknesses:**

1. **Limited Methodological Novelty:** The core idea of generating synthetic outliers in the latent space using Gaussian distributions is an established technique in vision OSR (e.g., VOS, NPOS). The introduction of adaptive, per-dimension standard deviations and a classifier feedback loop provides a degree of innovation, but the overall conceptual advance over the non-parametric NPOS feels incremental.
2. **Lack of Theoretical or Deep Analytical Justification:** The proposed adaptive uncertainty mechanism and the feedback loop are heuristic. The paper would be significantly strengthened by a more rigorous analysis or theoretical intuition explaining why this adaptive mechanism is necessary and how it better captures the OSR space compared to a well-tuned fixed variance.
3. **Experimental Setup Concerns:**The strategy of designating the "last class" as unknown for UCR/UEA is arbitrary and may not reflect realistic OSR scenarios where unknown classes can be semantically or distributionally "near" or "far."
4. **Baselines Lack Timeliness:** The empirical comparison is not state-of-the-art. The primary baselines (VOS, NPOS, ASH, ReAct, Scale) were published in 2023 or earlier. The OSR field is rapidly evolving, yet the paper does not compare against any major OSR methods from 2024.
5. **Marginal Gains**: The improvements over strong baselines like NPOS and VOS are often modest and not consistently significant across all datasets and encoders.
6. The paper lacks sufficient experimental analysis, including thorough ablations, hyperparameter sensitivity studies, and resource cost evaluation, which is critical for assessing the method's true contribution and practicality.

**Questions:**

See above.

---

### Official Review · Reviewer_Amhx · 2025-10-31

**Soundness:** 1
**Presentation:** 2
**Contribution:** 1
**Rating:** 2
**Confidence:** 4

**Summary:**

The paper tackles open-set recognition (OSR) for time-series data, classifying known classes while rejecting unknowns. It introduces ALSET, which synthesizes unknown examples in latent space by learning per-dimension standard deviations around each known sample via a neighborhood-based estimator. The classifier is trained with true labels for knowns and a uniform target over classes for synthesized unknowns to curb overconfidence. A feedback loop between the classifier and the estimator regulates distribution spread. If generated outliers begin to overlap with known samples, the classifier penalizes further variance growth, preserving separation. To establish a strong baseline, the authors adapt computer-vision OSR methods for time-series evaluation. Overall, ALSET offers an outlier synthesis-and-exposure framework for time-series OSR benchmarks.

**Strengths:**

The paper is well written and clear in its problem definition.



The evaluation include different and several metrics for performance measurement.

**Weaknesses:**

W1) The authors aim to define the OSR task for time-series datasets; however, the paper should better motivate concrete applications and explain why OSR is crucial in those settings.

W2) Several recently accepted OE methods are not cited [D,E]

W3) As this direction is  mature in computer vision, the authors should clarify their technical novelty relative to vision methods. They needed to  demonstrate the approach on an image benchmark and compare against  vision baselines to validate that the pipeline  generates informative synthesized outliers.

W4) In Tables 1 and 2, I do not see a significant AUROC gap in favor of ALSET, and variance over multiple runs is missing.


W5) The method sounds like a combination of prior ideas (A, B, C) adapted to time series; the paper should clarify what is  new and why it matters beyond adaptation.

W6) The open-set protocol using “last class as unknown” can be arbitrary and dataset-specific. I expect random unknown-class selections and results averaged over multiple splits, as is common in the image domain.

W7) Important hyperparameter details are missing. How are α, λ, and β chosen in practice? There is no ranges, the selection criterion and a sensitivity analysis to show how robust the pipeline is to these choices.



[A] A Baseline for Detecting Misclassified and Out-of-Distribution Examples in Neural Networks


[B] Deep Anomaly Detection with Outlier Exposure


[C] VOS: Learning What You Don't Know by Virtual Outlier Synthesis

[D] RODEO: Robust Outlier Detection via Exposing Adaptive Out-of-Distribution Samples

[E] Adversarially Robust Out-of-Distribution Detection Using Lyapunov-Stabilized Embeddings

**Questions:**

See Weaknesses.

---

### Official Review · Reviewer_1iVe · 2025-11-10

**Soundness:** 2
**Presentation:** 3
**Contribution:** 2
**Rating:** 2
**Confidence:** 3

**Summary:**

The authors introduce ALSET, a novel model designed for open set recognition in time series classification, aiming to classify known classes while simultaneously identifying unknown ones accurately. ALSET learns a dimension-wise Gaussian distribution over the latent embeddings of samples from known classes, and sampled points from this distribution are represented as unknown classes. The model is jointly trained for classification loss, L2 regularization on the standard deviations of the Gaussian distribution and the embeddings. Evaluated against various computer vision-based open set techniques across the UCR (83 datasets), UAE (28 datasets), and Human Activity Recognition benchmarks, ALSET consistently demonstrated superior performance.

**Strengths:**

1. While Open Set Recognition (OSR) has been reasonably well-researched for computer vision applications, its investigation for time series remains limited. The authors have done a good job developing a model specifically for Open Set Recognition in time series classification.

2. The paper is very well written and easy to read

3. Creating a per-dimension Gaussian distribution around the latent embedding and then sampling from this space to represent unknown classes is a new approach. The loss function is useful and well-argued.

4. By using more than 100 datasets, the authors demonstrate that, on average, the proposed technique consistently outperforms many other OSR techniques.

**Weaknesses:**

1. The proposed ALSET framework operates entirely in latent space and does not exploit any temporal or sequential characteristics of time-series data. The approach is modality-agnostic and could be directly applied to image embeddings. It is therefore unclear what makes ALSET a time-series-specific OSR method beyond the dataset choice and encoder architecture

2. ALSET generates synthetic unknown samples by drawing from a Gaussian distribution centered on each known embedding. This implies that the generated unknown samples are close to known data, whereas true unknowns can lie anywhere (potentially on the tails of the Gaussian). Although the adaptive variance mechanism may expand the space, there is a high probability that the synthetic unknowns do not represent truly out-of-distribution examples.

3. Following weakness 1, the evaluation only includes baselines originally designed for computer vision, adapted to time-series (l. 345,356). Existing time-series-specific OSR models (eg, [1,2]) are not considered.

Minor:

Please use the inline citation commands (\citet and \citep) correctly. It is slightly inconvenient to read with inconsistent citations.

References:

[1] Akar, Tolga, et al. "Open set recognition for time series classification." PAKDD 2022.

[2] Oh, Hyeryeong, and Seoung Bum Kim. "Multivariate time series open-set recognition using multi-feature extraction and reconstruction." IEEE Access 2022.

**Questions:**

1. Consider a simple hypothetical scenario, where we train the model for a particular dataset and give samples from different datasets. Do these samples fall within the high density region of the Gaussian distribution?

2. Since latent embeddings are of high dimension, one requires a large number of samples to cover the region (curse of dimensionality). How does the number of samples used impact the performance of the model?

3. Since ALSET is modality-agnostic, how does it perform on computer vision OSR benchmarks compared to CV-based baselines?

4. It is not clear to me how ALSET decides whether a sample is from In-distribution or Out-of-distribution. Does it also use eq. 1?

---

### Note · Authors · 2025-11-18

I have read and agree with the venue's withdrawal policy on behalf of myself and my co-authors.